# Single-cell analysis reveals altered tumor microenvironments of relapse- and remission-associated pediatric acute myeloid leukemia

Hope Mumme [1,7], Beena E. Thomas[2,3,7], Swati S. Bhasin [2,3], Upaasana Krishnan[1,4], Bhakti Dwivedi[5], Pruthvi Perumalla[4], Debasree Sarkar [1,3], Gulay B. Ulukaya[1], Himalee S. Sabnis[2,3], Sunita I. Park[6], Deborah DeRyckere [2,3], Sunil S. Raikar [2,3], Melinda Pauly[2,3], Ryan J. Summers [2,3], Sharon M. Castellino[2,3], Daniel S. Wechsler[2,3], Christopher C. Porter [2,3], Douglas K. Graham[2,3] & Manoj Bhasin [1,2,3,4] ✉

Acute myeloid leukemia (AML) microenvironment exhibits cellular and molecular differences among various subtypes. Here, we utilize single-cell RNA sequencing (scRNA-seq) to analyze pediatric AML bone marrow (BM) samples from diagnosis (Dx), end of induction (EOI), and relapse timepoints. Analysis of Dx, EOI scRNA-seq, and TARGET AML RNA-seq datasets reveals an AML blasts-associated 7-gene signature (*CLEC11A, PRAME, AZU1, NREP, ARMH1, C1QBP, TRH*), which we validate on independent datasets. The analysis reveals distinct clusters of Dx relapse- and continuous complete remission (CCR)-associated AML-blasts with differential expression of genes associated with survival. At Dx, relapse-associated samples have more exhausted T cells while CCR-associated samples have more inflammatory M1 macrophages. Post-therapy EOI residual blasts overexpress fatty acid oxidation, tumor growth, and stemness genes. Also, a post-therapy T-cell cluster associated with relapse samples exhibits downregulation of MHC Class I and T-cell regulatory genes. Altogether, this study deeply characterizes pediatric AML relapse- and CCR-associated samples to provide insights into the BM microenvironment landscape.

Acute myeloid leukemia (AML) accounts for about 20% of childhood leukemia[1]. Improvements in outcomes for children with AML, attributed to intensification of frontline therapy, hematopoietic stem cell (HSC) transplant, and better supportive care, have not kept pace with those for children with acute lymphoblastic leukemia (ALL). The prognosis and treatment of pediatric AML remain challenging because of disease heterogeneity, high relapse rates, and therapy-associated toxicity[2]. While measurable/minimal residual disease (MRD) status is

[1]Department of Biomedical Informatics, Emory University School of Medicine, Atlanta, GA, USA. [2]Aflac Cancer and Blood Disorders Center, Children's Healthcare of Atlanta, Atlanta, GA, USA. [3]Department of Pediatrics, Emory University School of Medicine, Atlanta, GA, USA. [4]Department of Biomedical Engineering, Georgia Institute of Technology, Atlanta, GA, USA. [5]Department of Biostatistics and Bioinformatics Shared Resource, Winship Cancer Institute, Emory University, Atlanta, GA, USA. [6]Department of Pathology, Children's Healthcare of Atlanta, Department of Pathology and Laboratory Medicine, Emory University School of Medicine, Atlanta, GA, USA. [7]These authors contributed equally: Hope Mumme, Beena E. Thomas. ✉e-mail: manoj.bhasin@emory.edu

critical for post-therapy patient stratification and clinical assessment, it is not infallible, with relapse occurring even after intense end of consolidation therapy. This may be partly due to inefficient determination of MRD positivity using current diagnostic techniques including bulk gene expression analysis and imaging. It is important to identify risk stratification strategies that can be used to better guide therapy decisions and improve clinical outcomes for pediatric AML patients. Deeper characterization of the disease is essential to overcome the above-mentioned limitations in AML prognosis and post-treatment assessment of response to therapy.

The Therapeutically Applicable Research to Generate Effective Treatments (TARGET) AML initiative and other large-scale genetic profiling studies have revealed that pediatric AML is a genetically diverse disease having various gene mutations, insertions, and fusions associated with poor prognosis[3]. Molecular characterization of the bone marrow (BM) tumor microenvironment (TME) along with the identification of biomarkers/targets that distinguish AML blast cells (AML-blasts) from non-blast cells are critical for defining malignant cells clusters and/or changes in the transcriptional landscape. The cellular composition of the TME and the transcriptional states thereof may contribute to and be affected by the leukemic process at the time of diagnosis (Dx) and the presence of residual blast cells at the end of induction (EOI) chemotherapy. Recent studies using single-cell RNA sequencing (scRNA-seq) assays have revealed the molecular diversity in blast and non-blast cells in AML patients[4–8]. These studies underline the critical role of scRNA-seq analysis for revealing cellular and molecular changes in BM microenvironment as well as in peripheral sites of AML patients.

In this work, we perform comparative single-cell transcriptome analysis of baseline Dx as well as post-therapy EOI samples critical for characterizing the role of TME determinants in pediatric AML relapse or continuous complete remission (CCR). Utilizing integrated scRNA-seq data analysis of paired Dx, EOI samples as well as bulk RNA sequencing dataset 1 of TARGET AML initiative (TARGET AML-1), we identify a 7-gene signature that distinguishes AML-blasts from non-transformed cells. Next, we validate the ability of the 7-gene signature to distinguish AML blasts and non-blast cells in AML samples, i.e., to identify AML-blasts and non-blasts in independent AML BM aspirate samples. Further analysis of AML Dx samples reveals specific transcriptomic differences between relapse- and CCR-associated AML-blasts. Furthermore, distinct baseline characteristics of innate and adaptive immune cells at Dx correlate with either CCR (monocytes/macrophages augmentation) or relapse (exhausted T cells augmentation). We also identify post-therapy residual blasts in EOI samples and clinical outcome-associated differences in immune cells. With the identification of specific cell types and incidental genes and pathways at both Dx and EOI, associated with pediatric AML relapse/CCR, this study opens avenues for the development of better diagnostic and therapeutic targets in pediatric AML.

## Results

### Clinical samples and study design

Frozen BM samples obtained from the Children's Healthcare of Atlanta (CHOA) pediatric biorepository were viably thawed and processed using 10x Genomics platform for single-cell profiling. A total of 31 BM samples were processed from 20 patients (six with relapse; patients 1–6 and 14 with CCR; patients 7–20) (Supplementary Tables 1, 2). Of the 31 BM samples, 19 were collected at Dx (1D–14D, 16D–20D), 10 were collected at EOI (3E, 5E, 6E, 14E–20E), and two were collected at relapse (Rel) (5R, 6R) (Supplementary Fig. 1, Supplementary Tables 1). Therefore, there were nine paired Dx and EOI samples, of which four were used for developing the 7-gene signature (patients 3, 5, 6, and 14), and five were used to validate the 7-gene signature (patients 16–20). Two patients (5 and 6) had samples collected at Dx, EOI, and Rel (Supplementary Fig. 1, Supplementary Table 1).

## Single-cell analysis identified differentiated cells and heterogeneous patient-specific AML-blast clusters

To develop a signature for identifying AML-blast cells, we initially performed a comparative analysis on scRNA-seq data from four paired Dx, EOI samples (patients 3, 5, 6, 14). After preprocessing, alignment, and quality control steps including removal of low-quality reads and cells, the remaining 19,350 cells were analyzed using unsupervised and supervised approaches. The unsupervised analysis of preprocessed and normalized scRNA-seq data using the Uniform Manifold Approximation and Projection (UMAP) approach[9] revealed 14 cell clusters (Fig. 1a). Eleven of these clusters were manually annotated based on the expression of well-established gene markers as differentiated immune and stromal cells (Fig. 1b). The remaining clusters were putatively considered as blast cell clusters and annotated based on top cluster-specific overexpressed genes (MPO+, CD34+, AZU+) (Fig. 1b, Supplementary Fig. 2). The differentiated immune cells included T cells (CD3D+, IL32+), B cells (CD19+, CD79A+, MS4A1+), monocytes (mono; CD14+), macrophages (macro; CD68+), monocytes/macrophages (Mono/mac; CD14+, CD68+), plasmacytoid dendritic cells (pDC; GZMB+, IL3RA+, IRF8+), pre-B/plasma cells (CD38+, CD79A+, JCHAIN+) and neutrophils (CD63+) (Fig. 1b). The three undifferentiated blast cell clusters exhibited overexpression of canonical blasts associated marker genes (MPO+, CD33+, CD34+) and/or multiple lineage genes (CD3D+, CD19+, ELANE+, etc.) (Fig. 1b, Supplementary Fig. 2). The blast cell clusters were patient-specific, reflecting heterogeneity among the blast cells (Fig. 1c). The split UMAP analysis showed that these three canonical blast cell clusters were over-represented in the Dx samples (Dx-enriched) compared to EOI samples which had more differentiated cells (EOI-enriched) (Fig. 1d). In summary, paired Dx, EOI scRNA-seq data analysis revealed heterogeneous AML-blast cells as well as immune cells and other differentiated cells.

### Identification of AML-blasts progenitor signature

After identifying the AML-blast and non-blast clusters, the integrated scRNA-seq data of the paired Dx, and EOI samples from four patients (3, 5, 6, and 14) were used to develop an AML-blast-specific gene signature. Differential expression of genes (DEG) analysis using Wilcoxon rank test, performed between the Dx enriched AML-blast cell clusters (MPO+, CD34+, AZU+) and EOI enriched non-blast cell clusters from Dx and EOI time-points, respectively (Fig. 1d), identified a set of 232 significantly DEGs (Fold Change (FC) > 1.2, adjusted $P < 0.01$). To further refine the gene list, we performed external analysis using the TARGET AML bulk transcriptome dataset 1 (TARGET AML-1)[3] that contains Dx BM samples with different proportions of blast cells and post-treatment EOI BM samples. The TARGET AML-1 Dx samples were partitioned into 3 bins based on the extent of disease burden (>60%, 30–60%, <30% blast cells) to determine the blast percentage associated expression patterns of the identified 232 scRNA-seq data blast-related genes. 44 genes were significantly overexpressed in high blasts samples (>60%) compared to EOI samples (FC > 1.8 and $P < 0.03$) as well as low blasts (<30%) samples (FC > 1.8 and $P < 0.04$) (Supplementary Fig. 3, Supplementary Table 3). The AML-blast specificity of these 44 genes is evident upon gene expression analysis in scRNA-seq data from patients 3, 5, 6, 14, with higher expression in the Dx AML-blast cells compared to post-treatment EOI non-blast cells (Fig. 1e, Supplementary Table 4). Of the 44 genes, 20 genes that showed specific overexpression in the Dx AML-blasts compared to post-treatment non-blast cells were selected for further evaluation (Supplementary Fig. 4, Supplementary Table 5). The expression of the selected 20 genes was higher in >60% and 30–60% blasts samples compared to <30% blasts and EOI samples in the TARGET AML-1 dataset (Fig. 2a). These overexpressed genes in blast cells include genes related to anti-apoptosis (PRAME, MSLN, CITED4)[10–12], growth factor for progenitor hematopoietic cells (CLEC11A)[13], cell proliferation (CAPRIN1)[14], and PPARα-induced proliferation and tumor growth (FABP5)[15]. Feature

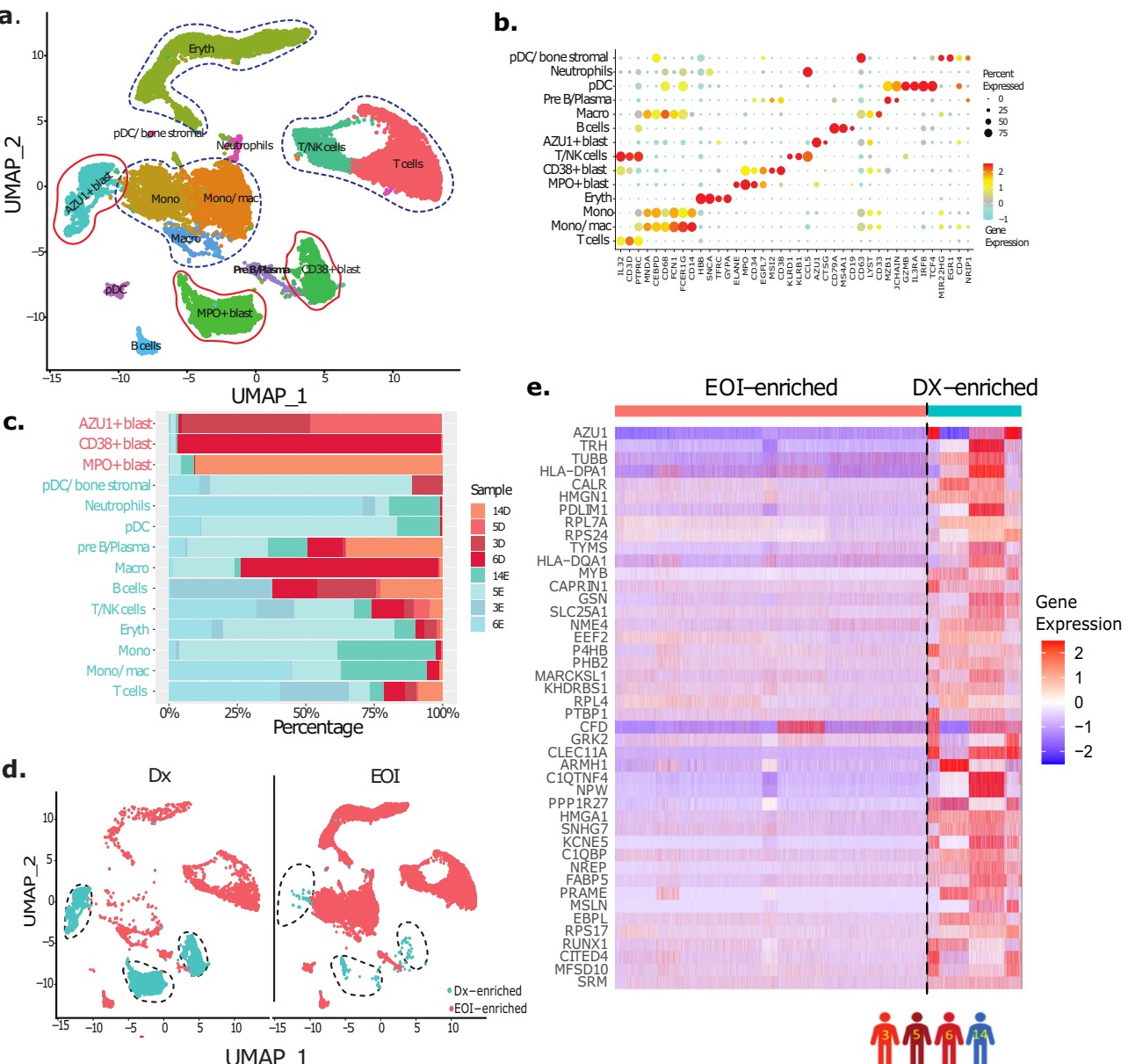

**Fig. 1 | Single-cell transcriptional profiling identifies heterogenous putative AML blast cell clusters. a** UMAP embedding of the four paired Dx and EOI biologically independent samples (three with relapse; patients 3, 5, 6, and one with CCR; patient 14) consisting of 19,350 high quality single cells partitioned into fourteen clusters. These clusters are colored based on expression of canonical cell type(s) gene markers (Fig. 1b). **b** Dot plot showing expression of different cell type specific marker genes that were used to annotate the cell types. The color intensity represents gene expression with red: high, yellow: medium, cyan: low, and size of dot represents percentage of cells expressing each gene in individual cell types. **c** Relative proportion of different patient samples in each cell type cluster. Most of the blast cell clusters are made up of cells from one or two patient samples indicating AML blast heterogeneity. On the other hand, most of the immune cell clusters contain cells from multiple patient samples. D denotes Dx samples, E denotes EOI samples. **d** Split UMAP showing the putative blast cells that are significantly over-represented in Dx samples (Dx enriched: ~95% of total blast cell clusters are made up of Dx samples with remaining ~5% made up of EOI samples; EOI enriched: non-blast cells making ~95% of EOI samples). **e** Heatmap of the selected set of 44 genes showing significant overexpression (Wilcoxon rank sum test, two-sided; Fold Change >1.2, *P* < 0.01) in Dx enriched blast cells in comparison to EOI enriched differentiated cells. Relative gene expression is shown with blue and red colors representing low and high expressing genes, respectively. Columns and rows represent the cells and genes, respectively. Patient samples used are shown with colored human icons at the bottom of the figure.

plots of two of the identified blast-associated genes (*NREP* and *CLEC11A*) displayed high expression in the three blast clusters identified in the paired Dx, EOI samples (Fig. 2b) that are equivalent to or better than the genes used for clinical identification of AML-blasts (*CD34, MPO, CD33,* and *CD56/NCAM1*) (Supplementary Fig. 5). While not much difference was seen in the expression of the 20 genes between >60% and 30–60% Dx blast samples (many of these genes are expressed highly in AML-blasts), the differences in expression were obvious when compared with low blasts (<30%) and post-treatment

EOI samples (Fig. 2a, c). Also, the majority of the 20 blast-associated genes (14 out of 20) exhibited significant association with overall survival (OS) in the TARGET AML-1 dataset (Fig. 2d, Supplementary Table 6) indicating their involvement in AML pathology. To generate a more concise AML-blast-specific signature and evaluate performance in discriminating AML-blast cells from other cells, we implemented a supervised machine learning approach, Support Vector Machine (SVM) and calculated its performance using cross-validation approach. SVM is a supervised machine learning technique for identifying

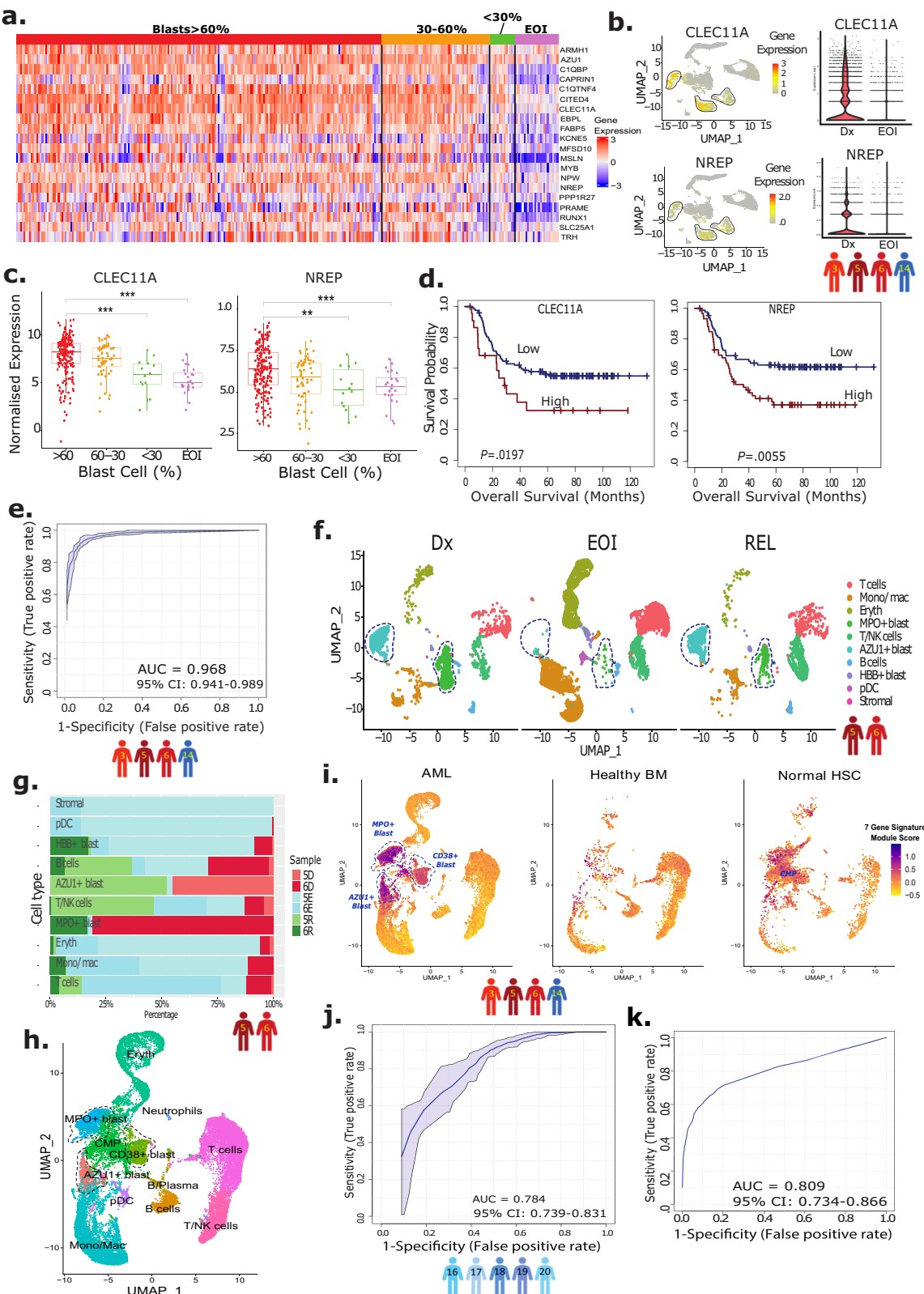

multigene biomarker panels from complex datasets[16,17]. The analysis identified a set of seven genes (*CLEC11A, PRAME, AZU1, NREP, ARMH1, C1QBP, TRH*) that discriminated AML-associated blast cells from non-blast cells with an AUC of 0.968 (Fig. 2e). The expression of the individual genes depicted variability across the patients, attributable to AML heterogeneity (Supplementary Fig. 6a). The expression levels of most genes, except for *TRH*, were significantly higher in Dx-AML blast

cells across multiple patients. These genes exhibited either no or very low expression in non-blast EOI-enriched cells from all patients. Some of these genes, have been previously associated with AML; for example *CLEC11A* is a growth factor of hematopoietic cells[13], *PRAME* expression is enhanced in AML[18], and *AZU1* expression is modulated in AML[19]. *NREP* is known to affect cell development[20], while PRAME may alter metabolic and immune pathways[21]. The analysis also identified the

**Fig. 2 | Development of blast progenitor signature from heterogeneous AML-blasts. a** Heatmap of 20 genes that show significant overexpression in the diagnosis (Dx) TARGET AML-1 samples with high blast enrichment (> 60% blasts and 30–60% blasts) as compared to Dx low (<30%) blasts and end of induction (EOI) samples. The Y-axis represents gene names and X-axis represents the patient's ID. Color scale: blue and red colors represent low and high expression of genes, respectively. **b** Feature plots of select genes (*CLEC11A, NREP*) from 20-genes set showing uniform overexpression in the lassoed blast clusters (UMAPs of Dx, EOI samples from patients 3, 5, 6, 14). Color scale represents high (red), medium (yellow), and low (gray) expression of the select genes. Violin plots of select genes (*CLEC11A, NREP*) expression in Dx (*n* = 4) and EOI (*n* = 4) samples, showing overexpression in the former. **c** Representative expression profiles of select genes (*CLEC11A, NREP*) show a progressive downregulation pattern from high blast % to low blast % to EOI samples in TARGET AML-1 dataset (biologically independent samples in >60% blasts group, *n* = 203; 30–60% blasts group, *n* = 60; <30% blasts group, *n* = 14; EOI group, *n* = 24, two-sided Wilcoxon rank sum test, **P < 0.01, ***P < 0.001). Comparisons of normalized *CLEC11A* expression in >60% blasts vs. <30% blasts (*P* = 9.6e −05) and >60% blasts vs. EOI (*P* = 1.1e−08) groups and *NREP* expression in >60% blasts vs. <30% blasts (*P* = 0.0052) and >60% blasts vs EOI (*P* = 0.00018) groups had significant results. Boxplots show the distribution of expression with the center of the box representing the median, upper and lower bounds representing 75% and 25% percentiles, and upper and lower whiskers extending to the largest value no further than 1.5 times interquartile range from bounds of box. Source data are provided as a Source data file. **d** Survival correlations showed the higher expression of *CLEC11A* (biologically independent samples: High: *n* = 23, Low: *n* = 122; HR = 1.96,

95% Confidence Interval (CI): 1.10–3.48, Cox HR *P* = 0.022, one-sided log-rank *P* = 0.0197) and *NREP* (biologically independent samples: High: *n* = 60, Low: *n* = 85; HR = 1.95, 95% CI: 1.21–3.14, Cox HR *P* = 0.0064, one-sided log-rank *P* = 0.0055) is associated with poor survival. One-sided log-rank tests were used to compare survival curves of high and low expression groups. Cox proportional hazards models were used to calculate hazard ratios and Wald tests were used to determine significance. **e** Receiver operative Curve (ROC) depicting performance of Support Vector Machine (SVM) approach based 7-gene signature (*CLEC11A, PRAME, AZU1, NREP, ARMH1, C1QBP, TRH*) discriminated AML-blasts from other cells. AUC (Area under curve) is 0.968, and gray shaded area around curve represents 95% CI (0.941–0.989). **f** UMAP plot depicting the blast cells (lassoed) of the longitudinal samples (Dx, EOI, Rel) from two patients (patient 5 and 6). **g** Bar plot showing fraction of Dx, EOI, and Rel samples of patients 5 and 6 in each cell type. D denotes Dx samples, E denotes EOI samples, R denotes relapse samples. **h** UMAP of integrated Dx, EOI AML samples (patients 3, 5, 6, 14), and healthy BM and normal HSCs from external datasets. **i.** Feature plot of the 7-gene signature module score in UMAPs split by dataset (AML, healthy BMs, normal HSCs). Color scale: purple represents a high module score, i.e., the 7-gene signature is highly expressed by the cells, and yellow represents a low module score. **j** Performance of SVM classifier as AUC using ROC analysis on independent validation set of Dx, EOI samples from five patients (patients: 16–20). AUC is 0.784, and gray shaded area around curve represents 95% CI (0.739–0.831). **k** Performance of SVM classifier as AUC on prospective publicly available validation dataset of pediatric AML, healthy BM, and normal HSC samples. AUC is 0.809 (95% CI: 0.734–0.866).

*ARMH1/C1orf228* gene, which has not been previously reported to be associated with pediatric or adult AML. Therefore, this set of seven genes overexpressed in AML-blasts is associated with cell functions that may promote tumor growth and spread (Supplementary Table 7).

The module score based on the 7-gene signature effectively distinguished potential AML-blast cells from non-blast cells with only cluster 3 (*MPO*[+]) and cluster 5 (*AZU1*[+]) displaying positive module scores (Supplementary Fig. 6b, c) upon integrated analysis of Dx, EOI and Rel samples (patients 5 and 6). Split UMAP revealed the two transcriptomically distinct potential blast cell clusters (*MPO*[+] and *AZU1*[+]) to be predominant in Dx (5D, 6D) and Rel (5R, 6R) samples, and minimally represented in EOI (5E, 6E) samples (Fig. 2f, g). Interestingly, the analysis also revealed differences in the immune cell's percentages (B and T cells) of patient 5 (MLL+) and patient 6 (7/7(del)/7(add)) (Fig. 2g). These blasts and immune cell type differences may be attributed to AML subtypes differences and may contribute to differences in disease progression/therapy response by each patient.

Next, we assessed the specificity of the AML-blast-associated 7-gene signature in differentiating blast cells from normal HSCs and healthy BM samples. The analysis was performed by integrating four paired Dx, EOI AML samples (patients 3, 5, 6, 14), a healthy BM scRNA-seq dataset from young adults[7] and a publicly available HSC dataset (https://data.humancellatlas.org/explore/projects/cc95ff89-2e68-4a08-a234-480eca21ce79). The clusters were annotated using top genes for AML-blasts (*MPO*[+], *AZU*[+], *CD38*[+]), canonical immune and stromal cell marker genes, and Single R automatic labels (Fig. 2h, Supplementary Fig. 7a, b). The split UMAP and bar plots showed AML blast cell clusters (*MPO*[+], *AZU*[+], *CD38*[+]) to be made up of mostly cells from AML samples (Supplementary Fig. 7a, b). Interestingly, the HSC dataset we retrieved from the Human Cell Atlas Census of Immune Cells had cells present in non-progenitor, mature clusters in our combined analysis; however, the majority (~60%) of the cells from the HSC dataset were located in the common myeloid progenitor (CMP) clusters. Feature plots of the module score (calculated using "AddModuleScore" function from Seurat package) based on the 7-gene signature demonstrated significantly higher expression in the AML-blasts compared to non-blast cells as well as to healthy BM and normal HSC samples (Fig. 2I). In addition, significant overexpression (*P* < 0.0001) of the 7-gene signature module score was observed in the

AML-blast cell clusters as compared to the CMP cells from the normal HSCs (Supplementary Fig. 7c).

To further validate the performance of the 7-gene signature in differentiating blast cells from non-blast cells, we analyzed two independent scRNA-seq datasets. Dataset one comprised of paired Dx, and EOI samples from additional five AML patients (patients 16–20). The split UMAP of paired Dx, EOI samples revealed Dx enriched clusters (0, 2, 5, 6, 10, 11, 13, 14, 17), which are reduced drastically in post-therapy EOI samples (Supplementary Fig. 8a). The non-blast cell clusters were annotated using SingleR[22] (v2.20) and established canonical immune and stromal cells markers (Supplementary Fig. 8b, c). The SVM-based analysis on 7-gene signature for distinguishing AML-blasts from non-blast cells resulted in an AUC of 0.78 (Fig. 2j). Dataset two comprised of an external AML dataset containing Dx samples from eight pediatric AML patients[7] (Supplementary Fig. 9). Integrated analysis of scRNA-seq dataset two along with healthy BMs[7] and normal HSCs from the human cell atlas (https://data.humancellatlas.org/explore/projects/cc95ff89-2e68-4a08-a234-480eca21ce79) was used to identify blast cells and normal BM cells. Clusters 4, 10, 11, 13–16, 18, 20, and 21 were labeled as AML blast clusters based on the following factors: (i) substantial proportion cells in the clusters from AML samples (Supplementary Fig. 9a, c) (ii) lack of established normal cell marker expression like *CD3D, CD14*, etc. (Supplementary Fig. 9b), (iii) expression of known AML blast markers, i.e., *CD38, MPO*, and *AZU1* (Supplementary Fig. 9b), and (iv) SingleR annotation of cell clusters as myeloid progenitors (GMP, CMP) (Supplementary Fig. 9c). To further assess the blast cluster annotation, we performed additional CNA analysis using the InferCNV tool[23]. For this analysis, the healthy BM and HSCs were utilized as the "reference" cells and AML clusters were input as the "observation" cells. The chromosome count metric was calculated for each cell, and these values were plotted on a feature plot (Supplementary Fig. 9d) showing high CNAs for multiple blast clusters (e.g., 11, 13, 14). Further blast clusters depicted a higher distribution of CNA chromosome counts than non-blast clusters (Supplementary Fig. 9e). The SVM model based on the 7-gene signature expression in this external dataset achieved an AUC of 0.809 in correctly identifying blast and non-blast cells (Fig. 2k). These high AUCs values obtained with the SVM models in two different independent AML scRNA-seq datasets further validates the AML-blasts specificity of the 7-gene signature.

Finally, we also validated the association between AML-blast cells and the 7-gene signature by evaluating their expression in an independent TARGET AML dataset 2 (TARGET AML-2), which contains both AML and normal BM samples (Supplementary Fig. 10). This dataset 2, released recently by the TARGET AML initiative, was not used during the development of the 7-gene signature. The TARGET AML-2 primary AML samples (Dx) with clinical BM blast percentage >5% ($n$ = 1320) were included in our correlation analysis. We observed a significant association between the Z-score enrichment value for the 7-gene signature and the BM blast percentage (Supplementary Fig. 10a, Pearson's correlation, R = 0.16, $P$ < 0.0001). The significant association of the 7-gene signature expression with clinically reported blast cells percentages further validates the AML-blasts specificity of the signature. Some of the individual genes in the 7-gene signature (CLEC11A, NREP, ARMH1, C1QBP) also exhibited significant associations with BM blast percentage (Supplementary Fig. 10b–h, Pearson's correlation, R > 0, $P$ < 0.01). The development and validation process of the 7-gene AML-blast signature is summarized in the flow chart presented in Supplementary Fig. 11. The rigorous validation across multiple datasets and observing reduced expression levels in healthy BM and normal HSCs datasets, supports association of 7-gene signature with pediatric AML-blasts.

Chromosome disruptions are common in AML[24]. Therefore, for confirming the malignant nature of the blast clusters identified in the integrated Dx, EOI scRNA-seq data from patients 3, 5, 6, 14, we performed copy number alteration (CNA) analysis of the cells from the blast clusters (AZU1+, MPO+, CD38+) using the InferCNV tool[23,25]. Comparison of cells from blast clusters to cells from healthy BM and normal HSCs revealed CNAs (amplifications and deletions) in the blast clusters (Supplementary Fig. 12). This is in accordance with recurring CNAs reported in AML, especially in genes located on chromosome 1q, 8, 11q, 13q, 17q, 19, and 21q[24]. AML subtype specific CNAs were also observed, for e.g., CD38+ blasts from patient 6 with 7q (7/add(7q)/del(7q)) displayed gain and loss of chromosome 7, AZU1+ blasts from patients 3 and 5 displayed deletions in chromosome 15. Chromosome 15 is known to be associated with myeloid malignancies. AZU1+ blasts displayed amplifications in chromosome 19 which is the chromosomal location of the AZU1 gene (Supplementary Fig. 12).

The clinical significance of this 7-gene signature was assessed by evaluating correlation with survival in the TARGET AML-1 data. Survival analysis using the cutp method in Survival genie web tool[26], revealed that patients having high expression of the 7-gene signature had a significant correlation with poor overall survival compared to patients with low expression of these genes (HR = 2.3, log-rank $P$ = 0.007) (Supplementary Table 8).

### Comparative analysis of Dx relapse- and CCR-associated AML-blasts

To characterize the Dx blast and immune cells, we analyzed 14 of the BM samples collected at Dx, 6 with relapse (1D–6D) and 8 with CCR (7D–14D) (Supplementary Fig. 1, Supplementary Table 1). After quality control, processing, and normalization, 28,181 cells clustered into 15 distinct clusters (Supplementary Fig. 13a). Based on the 7-gene blast signature module score, 14,166 cells were classified as AML-blast cells (clusters 1, 3, 5, 9, 10, 12, 13) and the remaining 14,015 cells were classified as non-blast cell clusters (Supplementary Fig. 13b).

To perform focused analysis on the AML-blast cells, we subsetted out blast cells and performed reclustering. The split UMAP revealed that the CCR-associated blast clusters (1, 3, 4, 6, 8, 10, 14) clustered next to each other and relapse-associated blast clusters (0, 2, 5, 7, 11, 12, 13) clustered together (except cluster 9) suggesting more similarity in transcriptome between the blast cells from similar clinical outcome i.e., relapse or CCR (Fig. 3a). Distinct patient-specific blast clusters indicative of inter-patient variation was observed upon plotting patient samples contribution for each cluster (Fig. 3b). CCR-associated

clusters (1, 3, 4) and relapse-associated clusters (0, 2, 7, 13) clustering near each other and representing multiple patients were selected for further analysis. DEG analysis of these clusters revealed significant transcriptome differences with the CCR-associated clusters exhibiting high expression of genes like MPO, IFITM3, TRH, PRTN3, and HLA-DPA1 and relapse-associated clusters expressing elevated levels of genes like CRIP1, FLNA, and RFLNB/FAM101B (Fig. 3c). Decreased expression of antigen-presenting HLA molecules has been reported in relapsed AML samples[27] with methylation changes being shown to suppress HLA expression in certain AML genotypes[28]. Survival analysis using the Survival Genie[26] TARGET AML dataset indicated that FLNA (cytoskeleton component, signaling scaffold and Pol I transcription regulator) and RFLNB/FAM101B (formation of cartilaginous skeletal elements) genes overexpressed in relapse-associated cell clusters were significantly associated with poor OS (log-rank $P$ = 0.0033 and 0.015, respectively) (Fig. 3d, Supplementary Table 9). On the other hand, genes overexpressed in the CCR-associated clusters such as MPO (microbicidal activity of neutrophils) and TRH (controls the secretion of thyroid-stimulating hormone) were associated with significantly better OS (log-lank $P$ = 0.001 and 0.000396 for MPO and TRH, respectively) (Fig. 3d, Supplementary Table 9). In addition, higher expression of RFLNB/FAM101B genes showed a significant correlation with shorter EFS while lower expression was associated with intermediate EFS (Supplementary Fig. 14a). The interactive string network analysis based on known and predicted protein-protein interactions[29] identified FLNA as a key RFLNB/FAM101B interacting protein. Another identified interactor of RFLNB/FAM101B was WDFY4, which plays a critical role in the regulation of classical dendritic cells mediated cross-presentation of viral and tumor antigens (Supplementary Fig. S14b). The Kaplan–Meier plots showed a significant correlation of high expression of combined RFLNB/FAM101B and WDFY4 genes with poor OS (log-rank $P$ = 0.017) and shorter EFS (log-rank $P$ = 0.00094) (Supplementary Fig. 14c, d; Supplementary Table 8). Furthermore, pathways analysis on relapse-associated Dx blast cells DEGs depicted significant activation ($P$ < 0.01) of multiple pathways including "RhoGDI signaling", "eNOS signaling", "Androgen Signaling", and "Protein Kinase A signaling" (Fig. 3e). These relapse-associated blast cells also depicted significant inhibition of pathways related to JAK/STAT signaling, HIF1-α, Interferon, and neuroinflammation signaling (Fig. 3e). Further upstream regulator-based systems biology analysis identified activation of a highly connected cohesive network of cell growth and proliferation-related master regulators (MTA1, NKX2.3, TCF3) in relapse-associated blasts (Fig. 3f). Significant inhibition of multiple immune and metabolism-related key molecules including HIF1A, IRF7, and STAT1 was also observed in the Dx relapse-associated samples enriched AML-blasts (Supplementary Fig. 14e).

Additional AML genetic subtype-associated transcriptome changes were revealed when we conducted DEGs analysis among the Dx AML-associated blasts (1D-14D, 16D-20D samples). Supplementary Fig. 15 shows the top DEGs (FC > 1.2, $P$ < 0.05) in the blasts of samples split into groups based on mutations or chromosomal alterations. Three of the Dx relapse-associated samples in this study had MLL rearrangements. MLL (KMT2A) rearrangement is one of the most common recurrent cytogenetic aberrations in pediatric AML[30]. We observed increased expression of a specific set of genes in MLL rearrangement patients including GRK2, KCNE5, AZU1, TRPM4, and C4orf48. G protein-coupled receptor kinase 2 (GRK2) has been shown to associate with HSP90 in AML cells[31]. GRK2/HDAC6 mediates tubulin acetylation regulation that promotes aberrant cellular adhesion, motility, and transformation[31]. Daunorubicin resistance has been reported to be induced by miRNA-125 by decreasing GRK2[32]. C4orf48 is a protein secreting gene that has not yet been associated with AML. It is a neuropeptide that has been suggested to play a role in the differentiation of cells[33]. Dysregulation of myeloid differentiation associated AZU1 has been reported in t(8;21) AML[34]. Transient Receptor Potential

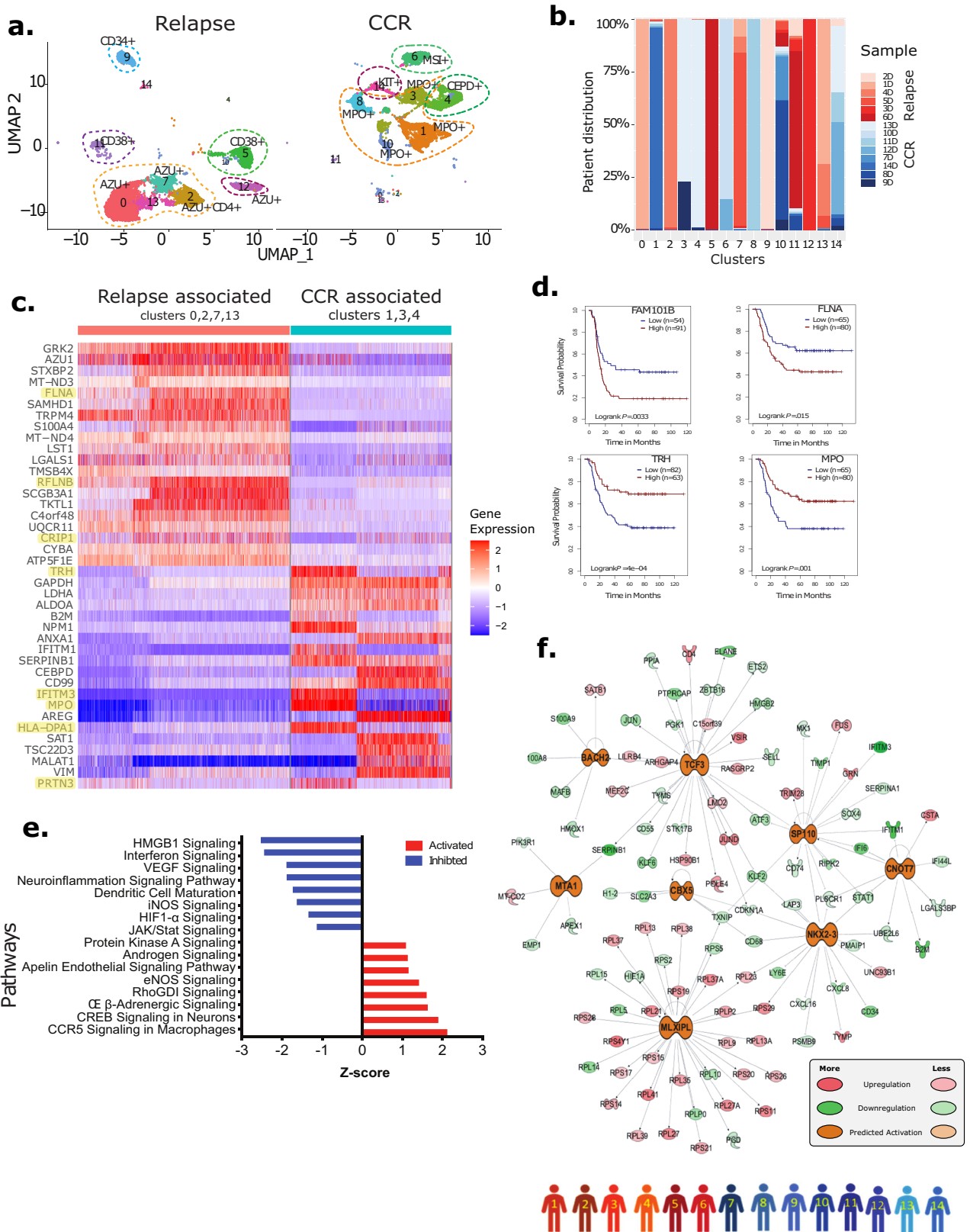

cation channel subfamily M member 4 (*TRPM4*) gene codes for a calcium-activated ion channel has been shown to be overexpressed in MLL-rearranged AML and therapeutic blockade suggested as an alternative approach to treat patients with high TRPM4[35]. The RUNX+ (t(8;21)(q22;q22) RUNX1-RUNX1T1) patients (11, 12, 14) achieved CCR outcome. These RUNX+ patients displayed overexpression of a specific set of genes as well as genes expressed in other AML genetic

subtypes. Overexpressed genes in the RUNX+ patient's Dx blast cells included *IFITM1, IFITM3, TRH, MPO, TUBB,* and *CD74*. While interferon-inducible transmembrane proteins member, *IFITM3*, has been reported to be associated with adverse AML prognosis in adults when only chemotherapy is given[36], it may have a different role in pediatric AML-blasts. CD74 reported to be expressed in AML, plays a critical role in MHC class II processing[37]. CD74 is also a transcriptional regulator

**Fig. 3 | AML-blast cells from samples with relapse depicted differences in transcriptome profile in comparison to samples with CCR. a** Split UMAP plot of Dx AML blast cells of relapse- and CCR-associated biologically independent samples from patients' 1–14 (shown with colored human icons at bottom of the figure). Blast cells formed fifteen clusters based on transcriptome profiles and were labeled manually based on top genes. **b** Stacked bar plots showing the proportion of cells from each patient in different clusters (Red: Relapse-associated, Blue: CCR-associated). D denotes Dx samples. **c** Heatmap of top differentially expressed genes between select relapse-associated (0, 2, 7, 13) and CCR-associated (1, 3, 4) samples clusters. Color scale: blue and red colors representing low and high relative expression of genes, respectively. The highlighted genes have been discussed in the main text for association with leukemia development and progression by regulating key pathways. **d** Kaplan–Meier plots show high expression of DEGs such as *FLNA*

and *RFLNB/FAM101B* in AML-blasts, having more expression in relapse-associated samples, were associated with poorer OS in TARGET AML dataset. Similar analysis on genes (*MPO, TRH*) with greater expression in the CCR-associated clusters depicted significant association of high expression with better OS. Number of samples in high and low expression groups are shown on plots. One-sided log-rank tests were used to compare survival curves of high and low expression groups. Cox proportional hazards models were used to calculate hazard ratios and Wald tests were used to determine significance. **e** Pathways that were significantly (*P* < 0.01) activated (Z-score >1.5)/inhibited (Z score < −1.5) in samples with relapse vs. samples with CCR. Activation and inhibition of pathways was determined using one-tailed Fisher's Exact tests. **f** Upstream regulatory molecules significantly activated (orange) in the blasts enriched with relapse-associated samples.

controlling immune regulation pathways[38] as well as HSC maintenance[39]. Other highly expressed genes like *MPO and TRH* were shown to be associated with better OS in this study (Fig. 3d). One patient with AMKL (patient 17) who died, exhibited overexpression of genes like *S100A11, LST1, CEBP, SERPINA1, CTSS, FCER1G*, and *IFITM2*. *IFITM2* expression is associated with TGFβ mediated epithelial–mesenchymal transition in gastric cancer[40] while *SERPINA1* which encodes acute phase protein, alpha 1 antitrypsin (AAT) is associated with tumor cell migration, colony formation, and resistance to apoptosis[41]. Of the two 7q (7/add(7q)/del(7q) cases in this study, one was relapse-associated (patient 6) and the other remission-associated (patient 16). A distinct set of genes were overexpressed in these two 7q(del/add) cases with relapse patient overexpressing genes like *IFITM1, IFI6*, and *ALDOA* while CCR patient overexpressed *SNHG29, FTH1*, and *IFITM3* genes. Along with subtype-specific gene over-expression, we observed that many genes (e.g., *S100A10, SNHG29, C1QTNF4, RGCC, HLA-DRB5, FCS, SAT1*) were overexpressed in multiple subtypes (Supplementary Fig. 15).

CNA analysis of Dx AML-blast cells and normal T cells compared to healthy BM/normal HSCs (Supplementary Fig. 16a) as well as matched patient normal cells (Supplementary Fig. 17) using the inferCNV tool[23] revealed patient-specific CNAs. Amplifications and deletions were observed in many chromosomes with prominent CNAs in chromosomes 1, 4, 6, 11, 13, 14, 16, 19, and 21 in AML-blast cells. To demonstrate the location of cells with CNAs on a UMAP in relation to the blast and T-cells from each patient, we generated a feature plot with the CNA chromosome count metric (excluding chromosome 6) and a UMAP with patient blast and T-cell locations (Supplementary Fig. 16b, c). The chromosome 6 alterations are likely attributed to differential HLA gene expression rather than true CNAs and therefore were removed from CNA analysis. Multiple reports have shown CNAs in AML malignant cells[24,42,43]. While chromosome 19 abnormalities have been reported for AMKL[44], we observed prominent amplifications in chromosome 19 for blast cells from MLL+ patients (i.e., patients 1, 4, and 5). A large study on 466 pediatric AML samples depicted CNAs to be associated with survival in standard-risk group. Additional studies with larger patient cohorts are needed to determine role of CNAs in pediatric AML as well as post-therapy clinical outcomes.

To summarize, scRNA-seq analysis revealed global as well genetic subtype-specific DEGs with clinical outcome associations in the Dx pediatric AML-blasts.

**Dx relapse-associated non-blast clusters depicted enrichment of exhausted T cells and diminution of M1 macrophages**

Next, we performed focused analysis on the non-blast cells from the Dx BM samples. The non-blasts clusters were selectively re-clustered and annotated based on the expression of established gene markers and cell annotation tool, singleR, which predicts cell types based on the expression of canonical cell lineage or type defining markers using reference primary cell datasets[22] (Fig. 4a, b;

Supplementary Fig. 18a, b). Split UMAP revealed the differential distribution of cell types, especially T cells and monocytes/macrophages in relapse- and CCR-associated samples (Fig. 4a). These results suggest differences in the type of immune cells populating the AML BM microenvironment at the time of Dx in those who go on to achieve CCR compared to those who eventually suffered relapse. To understand if a particular subtype of T cells is associated with AML relapse/CCR, we subclustered the Dx T cell cluster using Monocle package[45] which revealed 11 distinct subclusters of T cells (Fig. 4c, Supplementary Fig. 19). The observed patient-specific transcriptomically distinct subclusters for naive and activated T cells indicate differential response generated by AML-blasts (Fig. 4d, Supplementary Fig. 20). Naive T cell (*CCR7+, LEF1+, TCF7+*) subclusters 2, 4, and 7 have significant expression of cytotoxic T cell marker (i.e., *CD8*) (Supplementary Fig. 19b). Clusters 5, 6, 9, 10, and 11 were activated T cell clusters expressing *CCL5, KLRB1, KLRD1, GZMH, CD69*, and *CD44* (Fig. 4C, Supplementary Fig. 19b). The unequal distribution of CCR- and relapse- associated samples in different clusters in the T cell subsets is suggestive of relapse- and CCR-associated transcriptome variations in T cell sub-populations (Fig. 4d). Comparative analysis between relapse- and CCR-associated samples revealed that the former had a significantly higher percentage of T cells (Fig. 4e). Also, T cell exhaustion estimation based on ssGSEA score[46,47] revealed significantly higher exhausted T cells (*P* < 0.0001) in relapse-associated samples compared to CCR-associated samples (Fig. 4e). The relapse enriched T cell subclusters (5 and 6) depicted overexpression of T cell exhaustion marker genes including *HAVCR2, LAG3, PDCD1, NFATC1, TIGIT*, and *TOX* (Supplementary Fig. 19b). Also, the relapse-associated dominant clusters 1 and 11 depicted overexpression of *CD69*, a type II glycoprotein that is known to regulate inflammation, exhaustion of tissue-resident T cells, and promoting tumor growth/relapse[48]. On the other hand, significantly more naive T cells (*CCR7, LEF1, TCF7*) were observed in the CCR-associated T cell subsets (Fig. 4e). Comparative analysis of select clusters composed of multiple samples from relapse-associated subclusters (3, 5, 6) and CCR-associated subclusters (7, 8, 10) revealed significant DEGs (Fig. 4f). Relapse-associated subclusters showed enhanced expression of genes associated with effector T cells (*CCL5, NKG7, GNLY, GZMA, GZMK, NFATC1*), ubiquitin gene; *UBC* associated with cell differentiation and certain immune cell functions[49], inflammation associated and mitogen inducible gene, *DUSP2*[50]. On the other hand, CCR-associated clusters showed higher expression of naive T cell marker genes (*LEF1, TCF7, CCR7, SELL*), microtubule-associated gene; *RPL41*[51], scaffold protein gene; *ABLIM1* (Fig. 4f), The pathway analysis of relapse-associated cluster's upregulated DEGs revealed significant upregulation (*P* < 0.05) of multiple immune regulatory pathways including Th1 pathway, calcium-induced T lymphocyte apoptosis, and to a lesser extent, upregulation of T cell exhaustion pathway with downregulation of PD-1 and PD-L1 cancer immunotherapy pathways (Fig. 4g). These results suggest relapse- and

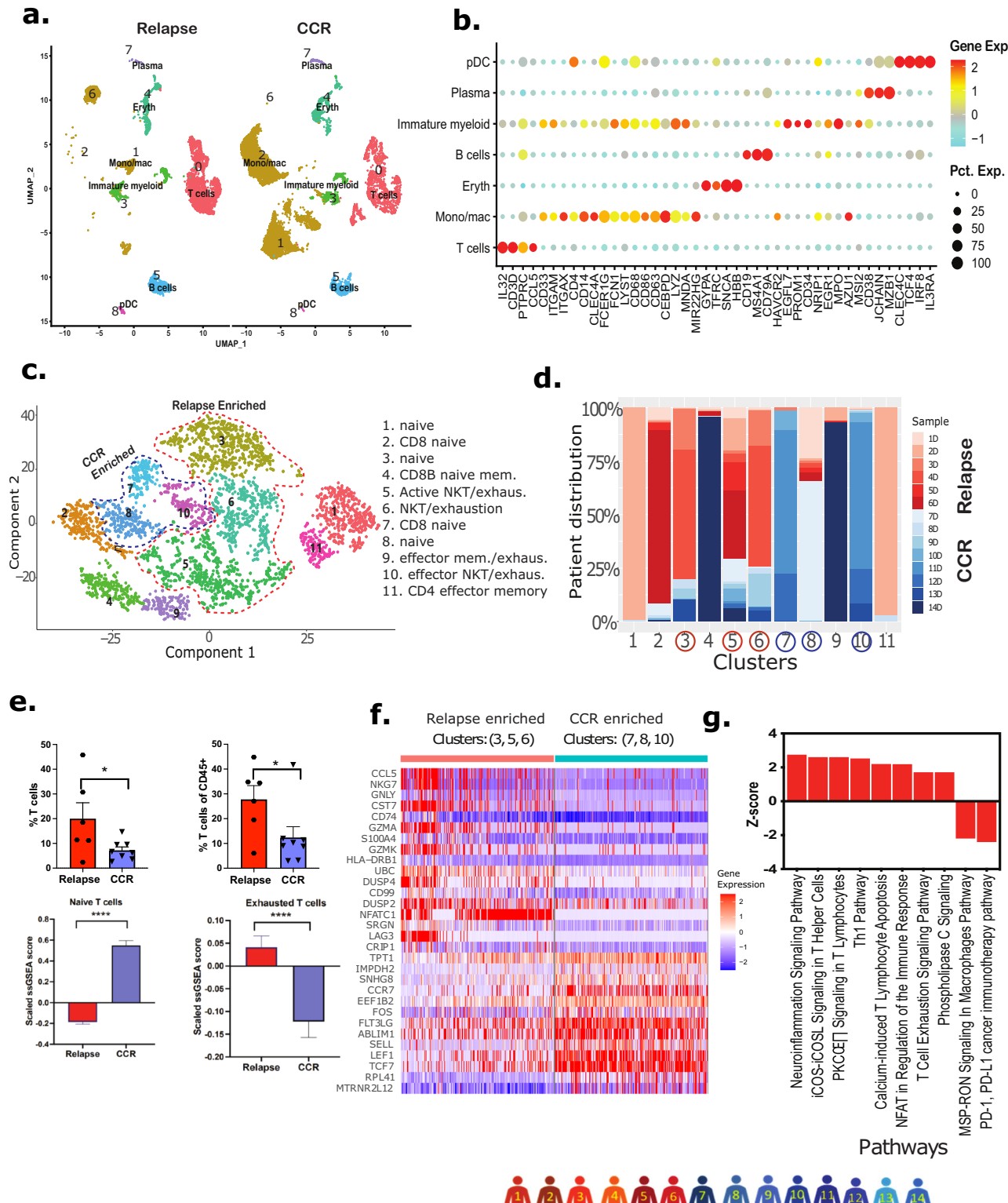

CCR-associated specific enrichment of T cell subtypes that may play a role in the post-therapy outcome.

## Inflammatory monocytes/macrophages enriched in CCR-associated samples

Focused analysis of Dx non-blast cells (from patient samples 1–14) identified three monocyte/macrophage cell clusters with differential proportions of samples from those with CCR and those with relapse

(Fig. 5a). Clusters 1 and 2 were enriched with CCR-associated samples while cluster 6 was exclusively comprised of relapse-associated samples (Fig. 5a, Supplementary Fig. 21a). The non-leukemic nature of the three monocyte/macrophage clusters was confirmed by low expression of leukemia myeloid markers (*ITGAM, ANPEP, CD33*) (Supplementary Fig. 21b). A dot plot of the gene markers used for annotations in this study showed that clusters 1 and 2 express multiple monocyte/macrophage markers (*FCER1G, CD14, CD68*), while cluster 6 expresses

**Fig. 4 | Immune microenvironment analysis of Dx samples.** Non-blast cells from biologically independent samples of patients 1–14 (shown with colored human icons at bottom of the figure) were selected to perform a focused analysis. **a** Split UMAP plot of BME cells based on relapse- and CCR-associated status of samples. **b** Expression-based dot plot of canonical cell-specific markers used for the annotation of lymphoid, myeloid, and erythroid lineages. The color intensity represents gene expression with red: high, yellow: medium, cyan: low, and size of dot represents percentage of cells expressing each gene in individual cell types. **c** t-distributed Stochastic Neighbor Embedding (t-SNE) plot depicting fifteen subclusters of T-lymphocytes. The subclusters were annotated based on the canonical cell type markers: Naive T cells (*CCR7*+, *LEF1*+, *TCF7*+), CD4+ effector T cells (*CD4*+, *CCR6*+, *CXCR6*+, *CCL5*+), CD8+ cytotoxic T cells (*CD8A*+, *CD8B*+, *GZMB*+, *GNLY*+, *PRF1*+) and exhausted T cells (*HAVCR2*+, *LAG3*+, *PDCD1*+, *NFATC1*+, *TIGIT*+, *TOX*+). Clusters enriched with cells from CCR, and cells from relapse samples are lassoed. **d** Stacked bar plots showing the cluster-wise proportion of cells from each patient (Red: relapse, Blue: CCR). **e** Proportion of T cells in relapse- and CCR-associated patients (top panel; CCR: $n = 8$ biologically independent samples, relapse: $n = 6$ biologically independent samples, two-tailed unpaired t-test, *$P < 0.05$. % T cells: $P = 0.0452$, % T cells of CD45+ : $P = 0.0470$). Overall enrichment of naive and exhausted T cell signatures across relapse- and CCR-associated samples based on ssGSEA score (bottom panel; $n = 599$ cells from $n = 8$ biologically independent CCR samples; $n = 1773$ cells from $n = 6$ biologically independent relapse samples; two-tailed unpaired t-test, ****$P < 0.0001$). Bar plots show the mean value, with error bars representing mean + standard error of mean (SEM). Source data are provided as a Source data file. **f** Heatmap of top DEGs in clusters 3, 5, 6 having more relapse-associated samples and clusters 7, 8, 10 having more CCR-associated samples (the T cell subclusters that were made up of multiple samples were selected for this analysis). Relative gene expression is shown with blue and red colors representing low and high expressing genes. **g** Pathways that are significantly activated (Z-score >1.5) and inhibited (Z-score < −1.5) with genes differentially expressed in the relapse-associated T cells. Pathways achieved $P < 0.01$ based on one-tailed Fisher's Exact test.

more premature genes (*MEIS1, CD34*) (Supplementary Fig. 21c, d). CNA analysis using inferCNV tool[23] confirmed that cluster 6 monocytes/macrophages are normal with no significant deletions or amplifications compared to healthy BM monocytes (Supplementary Fig. 22). Gene set enrichment analysis using M1 macrophages genes (*S100A8, S100A9, S100A12, TYROBP, VCAN, CD68, MNDA, CYBB, STAT1*) revealed that CCR-associated samples had significantly ($P < 0.0001$) more M1 macrophages compared to relapse-associated samples (Fig. 5b). DEGs analysis between clusters with more CCR-associated samples (clusters 1, 2) and relapse-associated samples (cluster 6) revealed upregulation of *S100A* inflammatory genes family, protease inhibitor, *CST3* and innate immune system regulator, *CTSS* in the former (Fig. 5c). Systems biology oriented analysis was used to identify modulated upstream regulators that could account for the DEGs observed in the clusters enriched with CCR-associated samples. Gene regulator analysis using Dx CCR-associated samples DEGs revealed upregulation of multiple M1 macrophage polarization-related key regulators including FOS, TREM1, CD44, and IL1B along with activation of the inflammatory response associated NFKB2 (Fig. 5d). The analysis also identified the key regulators that are downregulated including SAMHD1 and SATB1 which are associated with AML progression (Fig. 5d). SAMHD1 has been shown to play a critical role in inhibiting innate immunity through inhibition of NF-kβ and interferon pathways[52]. Evaluation of gene expression levels for key regulators FOS, TREM1, NFKB2, and NFKBIA depicted high expression in CCR-associated samples while *CAT* and *SAMHD1* were higher in the relapse-associated samples (Fig. 5e). To summarize, Dx CCR-associated samples are enriched with inflammatory M1 macrophages, whereas relapse-associated samples depicted enrichment of an immature monocyte population of cells.

## Characterization of post-therapy residual blast cells population at EOI

Characterization of post-therapy EOI samples BM microenvironment is critical for revealing the blasts and non-blasts landscape that determines clinical outcomes. To identify EOI residual blast cells, we performed an integrated scRNA-seq data analysis of four paired Dx and EOI samples (patients 3, 5, 6, 14) and one EOI sample (patient 15 whose Dx sample was not available). Dx and EOI samples integrated well with clusters 4, 6, and 8 enriched in Dx samples (Fig. 6a, Supplementary Fig. 23a). Further analysis revealed that these three Dx enriched clusters are made up of cells from individual patients (Supplementary Fig. 23b). The clusters were annotated using well-established gene markers (Supplementary Fig. 23c). These Dx enriched clusters (4, 6, 8) exhibiting overexpression of the genes from 7-gene signature (Supplementary Fig. 23d) were designated as AML cancerous blasts cell clusters. A small percentage of cells in blast clusters were observed to be from EOI samples (~6% of the Dx samples) (Supplementary Fig. 23e). To determine the transcriptome landscape of post-therapy residual

tumor cells, we performed DEGs analysis of Dx blasts and post-therapy EOI residual blast cells (Fig. 6b). The EOI residual blast cells depicted overexpression of multiple genes associated with tumor growth and poor outcome including *HOPX, SELENOP/SEPP1*, and *FAM30A/C1orf110*[53–55]. Survival analysis on these post-therapy residual tumor cells overexpressed genes in the TARGET AML-1 dataset showed significant association with poor OS as well as lower EFS (Supplementary Fig. 24, Supplementary Table 10). Further gene set enrichment-based pathways analysis depicted significant upregulation of multiple pathways including muscle contraction, fatty acid omega oxidation and PPARα network in the post-therapy EOI residual blast cells (Fig. 6c, Supplementary Table 11). Survival analysis on gene sets associated with upregulated pathways in EOI residual blasts (obtained from Molecular Signatures Database v7.5), in the TARGET AML-1 datasets, revealed significant association with poor survival indicating their (i.e., genes/pathways) role in disease progression (Supplementary Table 12). In addition, upstream regulator analysis on DEGs in EOI residual blast cells depicted activation of multiple key regulators associated with epithelial−mesenchymal transitions including SOX4, STAT3, and TGFB1 (Fig. 6d). To summarize, differential regulation of genes and pathways associated with modulation of BM microenvironment in post-therapy EOI residual blasts may result in poor OS and lower EFS in AML patients.

## Relapse-associated EOI samples depicted enrichment of T cells and diminution of monocytes/macrophages

To examine non-blast cells in the EOI samples, we selected cells from EOI alone (15,070 single cells) from the above Dx, EOI dataset and re-clustered them to obtain distinct cell clusters that were grouped and annotated based on expression of canonical marker genes (Fig. 7a, Supplementary Fig. 25). We focused on cell clusters demonstrating non-blast signatures that are differentially present in CCR- and relapse-associated samples. The split UMAP based on clinical groups (CCR and relapse) revealed the altered distribution of the transcriptionally distinct canonical immune cells (T cells, NK cells, and macrophages/monocytes) (Fig. 7a). Relapse- and CCR-associated non-blasts clusters at EOI revealed a similar pattern to that seen at Dx, i.e., higher enrichment of T cells in relapse-associated samples and a higher proportion of monocytes and macrophages in the CCR-associated samples enriched clusters (Fig. 7b, c). Further M1 and M2 macrophages enrichment analysis revealed a highly significant inflammatory M1 macrophages predominance in CCR-associated EOI samples ($P < 0.001$) and significant M2 macrophages enrichment in relapse-associated EOI samples ($P < 0.05$) (Fig. 7d). The T cells formed three distinct clusters i.e., naive T cell-1, naive T cell-2, and NK/T cells clusters (Fig. 7a, Supplementary Fig. 26a). Naive T cells-2 cluster depicted significantly ($P < 0.05$) higher enrichment in the relapse-associated EOI samples (Fig. 7e). The comparative analysis of the two naive T cell

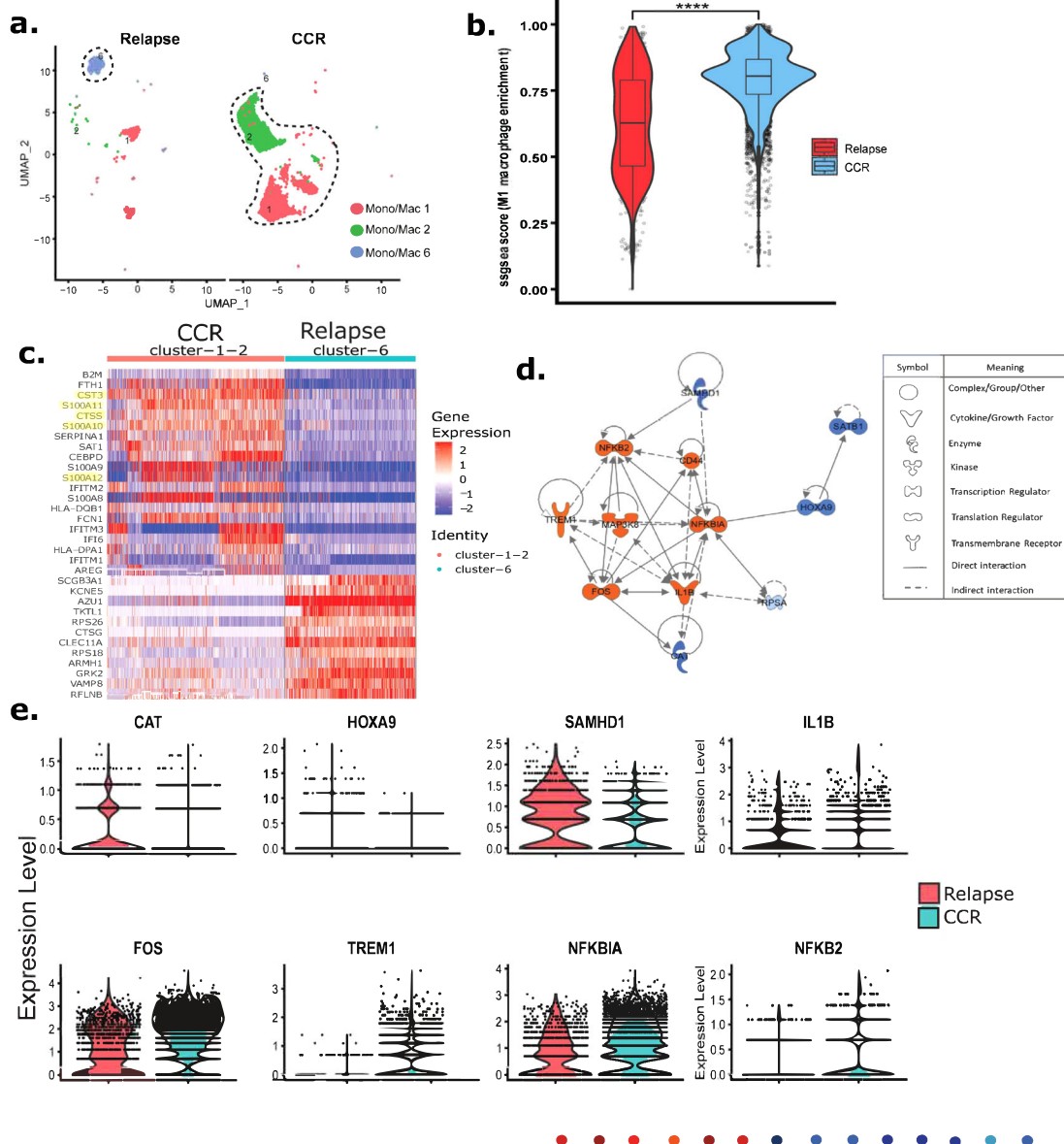

**Fig. 5 | CCR-associated samples at diagnosis depicted enrichment of inflammatory monocytes/macrophages.** We conducted the focused analysis of monocyte/macrophages clusters from biologically independent Dx samples (patients 1–14, shown with colored human icons at bottom of the figure). **a** UMAP plot depicting monocytes/macrophages clusters (lassoed) in Dx samples. Cluster 1 (Mono/mac 1) and 2 (Mono/mac 2) are enriched with cells from CCR-associated patient samples whereas cluster 6 (mono/mac 6) is enriched with cells from relapse-associated samples. **b** SsGSEA analysis shows significantly increased expression of M1 macrophages genes (*S100A8, S100A9, S100A12, TYROBP, VCAN, CD68, MNDA, CYBB, STAT1*) in CCR-associated samples compared to relapse-associated samples (clusters 1, 2, and 6). *n* = 5810 cells from *n* = 8 biologically independent CCR samples, and *n* = 1223 cells from *n* = 6 biologically independent relapse samples, two-tailed Wilcoxon rank sum test *(P* = 1.80e−177), ****P < 0.0001. Boxplots show the distribution of enrichment scores with the center of the box representing the median, upper and lower bounds representing 75% and 25% percentiles, and upper and lower whiskers extending to the largest value no further than 1.5 times interquartile range from bounds of box. Source data are provided as a Source data file. **c** Heatmap of top DEGs from the comparison of relapse- (cluster 6) and CCR-associated dominant (clusters 1, 2) clusters. Relative gene expression is shown with blue and red colors representing low and high expressing genes. The highlighted genes are the ones mentioned in the main text. **d** Upstream regulatory molecules significantly inhibited (blue) and activated (orange) in the CCR-associated samples enriched monocytes/macrophages clusters in comparison to the relapse-associated cluster. **e** Split violin plots showing expression levels of genes encoding specific regulators in CCR- (clusters 1, 2) and relapse-associated (cluster 6) samples.

clusters revealed that relapse-enriched naive T cell-2 has lower expression of MHC class I genes including HLA-A and HLA-B (Supplementary Fig. 26b). Pathway analysis of naive T cell-2 depicted downregulation of pathways related to Th1, Th2, calcium-induced T cell apoptosis, CD28 signaling in T helper cells, and integrin pathways (Fig. 7f). In summary, we observed clinical outcome i.e., CCR or relapse, associated differences in EOI non-blasts BM landscape.

## Independent validation of exhausted T cells and M1 macrophages association with pediatric AML outcomes using bulk RNA-seq gene set enrichment analyses

Immune cell-focused analysis of scRNA-seq data reveals a link between exhausted T-cells and poor pediatric AML outcome (Fig. 4e), while enrichment of M1 macrophages (Fig. 5b) and naive T-cells (Fig. 4e) correlates with improved outcome. We evaluated association of these

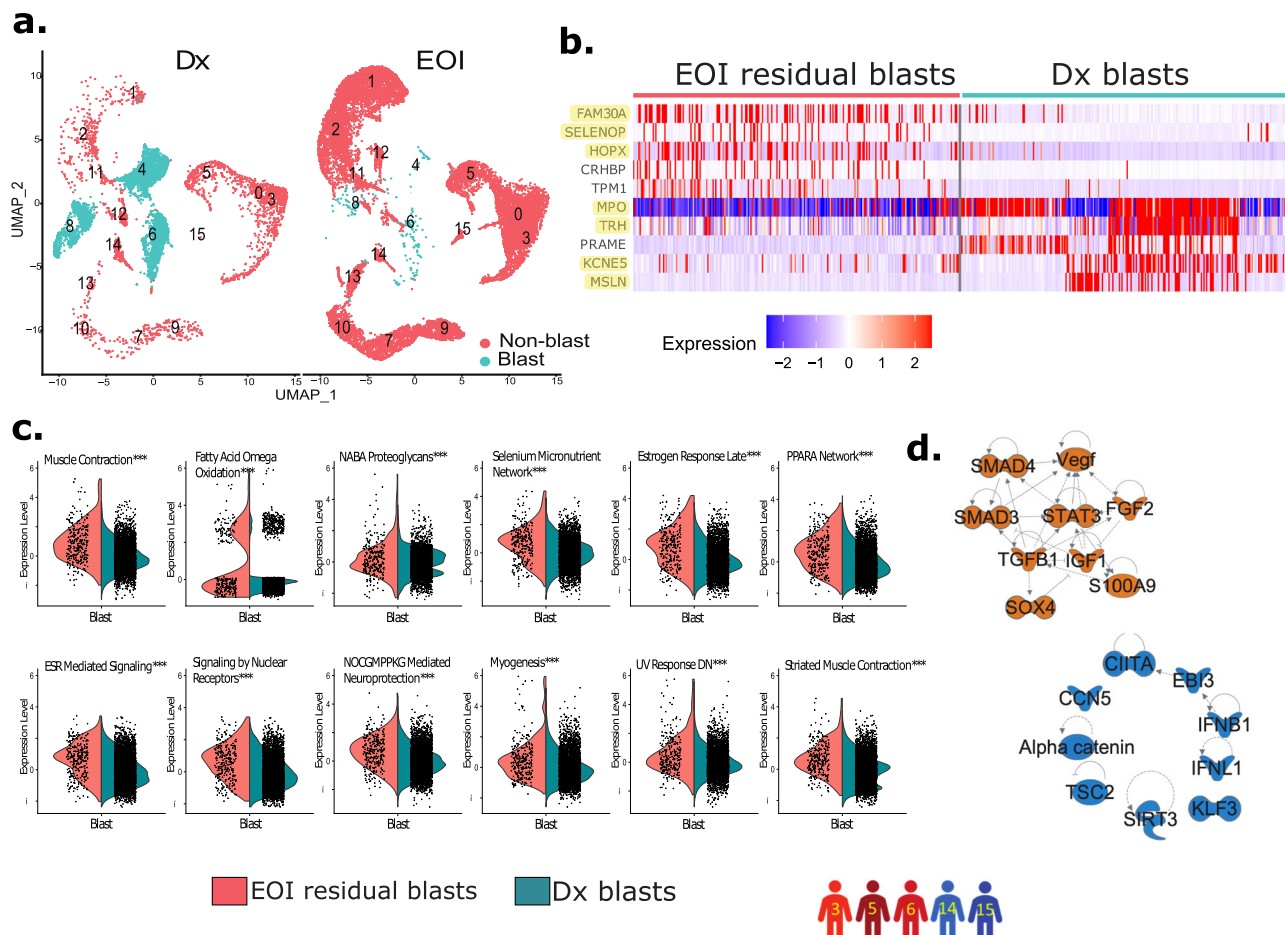

**Fig. 6 | ScRNA-seq analysis of EOI samples identified post-therapy residual blast cells with distinct transcriptome landscapes. a** Split UMAP plot shows the putative blast cells (colored in cyan) that are over-represented in Dx samples (patients 3, 5, 6, 14) and reduced in the EOI samples (patients 3, 5, 6, 14, 15). **b** Heatmap of top markers that are differentially expressed between EOI post-therapy residual and Dx blast cells. Relative gene expression is shown with blue and red colors representing low and high expressing genes. The highlighted genes are the ones mentioned in the main text. **c** Split violin plots of pathways significantly enriched in the EOI residual blast cells compared to Dx blast cells (Dx: *n* = 4173 cells

from *n* = 4 biologically independent samples, EOI: *n* = 264 cells from *n* = 5 biologically independent samples, linear model comparison of means[92], ****P* < 0.001, exact *P*-values are located in Supplementary Table 11). **d** Regulators that are significantly activated (orange color) and inhibited (blue color) in EOI residual blasts (*n* = 264 cells) compared to treatment responsive Dx blasts (*n* = 4173 cells). The significance of transcriptional regulators was determined using one-tailed Fisher's exact test. The regulators with *P* < 0.01 and absolute Z-score of ±1.5 were considered statistically significant.

---

immune cells signatures with AML risk score in two independent bulk RNA-seq datasets (i.e., Fornerod et al., AML dataset[56] and TARGET AML-2 dataset). The samples from bulk RNA-seq datasets were divided into different risk groups (i.e., high, medium, and low) based on leukemic stem cell 6 (LSC6) scores. LSC6 score, calculated using expression of six genes (*DNMT3B, GPR56, CD34, SOCS2, SPINK2, FAM30A*), depicted significant prognostic ability and estimation of EFS for pediatric AML[57]. We assessed various immune cell types in bulk RNA-seq datasets by employing a single sample gene set enrichment analysis (ssGSEA) approach. The immune cell types investigated included M1 Macrophages (*S100A8, S100A9, S100A12, TYROBP, VCAN, CD68, MNDA, CYBB, STAT1*), naive T-cells (*CCR7, LEF1, TCF7, SELL*), effector T-cells (*CCL5, NKG7, GNLY, GZMA, GZMK, NFACT1*), exhausted T-cells (*HAVCR2, LAG3, PDCD1, NFATC1, TIGIT, TOX*), and regulatory T-cells (*CCL5, KLRB1, KLRD1, GZMH, CD69, CD44*). In Fornerod et al., AML dataset, M1 Macrophages depicted a significant (*P* = 1.2e−05) negative correlation with the LSC6 score indicating lower enrichment in high-risk samples (Supplementary Fig. 27a). In addition, the exhausted T-cell enrichment score depicted a significant (*P* = 1.9e−05) positive correlation with the LSC6 score (Supplementary Fig. 27a). We also evaluated the association between categorized scores (i.e., low-,

medium-, and high-risk) and enrichment of these cell types. Low-risk AML samples were significantly enriched in M1 Macrophages as compared to medium-risk samples (*P* < 0.0001). Exhausted T-cells were significantly enriched in the medium-risk samples as compared to low-risk samples (*P* < 0.05) (Supplementary Fig. 27b). The high-risk samples were excluded from the Fornerod et al., AML dataset analysis due to low numbers (*n* = 3) as compared to the other two categories (medium-risk, *n* = 27; low-risk, *n* = 102).

A similar analysis on the TARGET AML-2 cohort also depicted a negative correlation between the LSC6 score and M1 Macrophages (*P* = 7.3e−11) as well as naive T-cells (*P* = 9.8e−11). Exhausted T-cells depicted a significant (*P* = 5.7e−09) positive correlation with the LSC6 risk score (Supplementary Fig. 28a). Furthermore, a comparison of ssGSEA scores for M1 Macrophages, naive T cell, and exhausted T cells in the low-, medium-, and high-risk categories in the TARGET AML-2 cohort showed that the low-risk AML samples were significantly more enriched in the M1 Macrophages (*P* < 0.001) and naive T-cells (*P* < 0.05) compared to the high-risk samples. On the other hand, the high-risk AML samples were significantly enriched in exhausted T cells compared to the low-risk samples (*P* < 0.001) (Supplementary Fig. 28b).

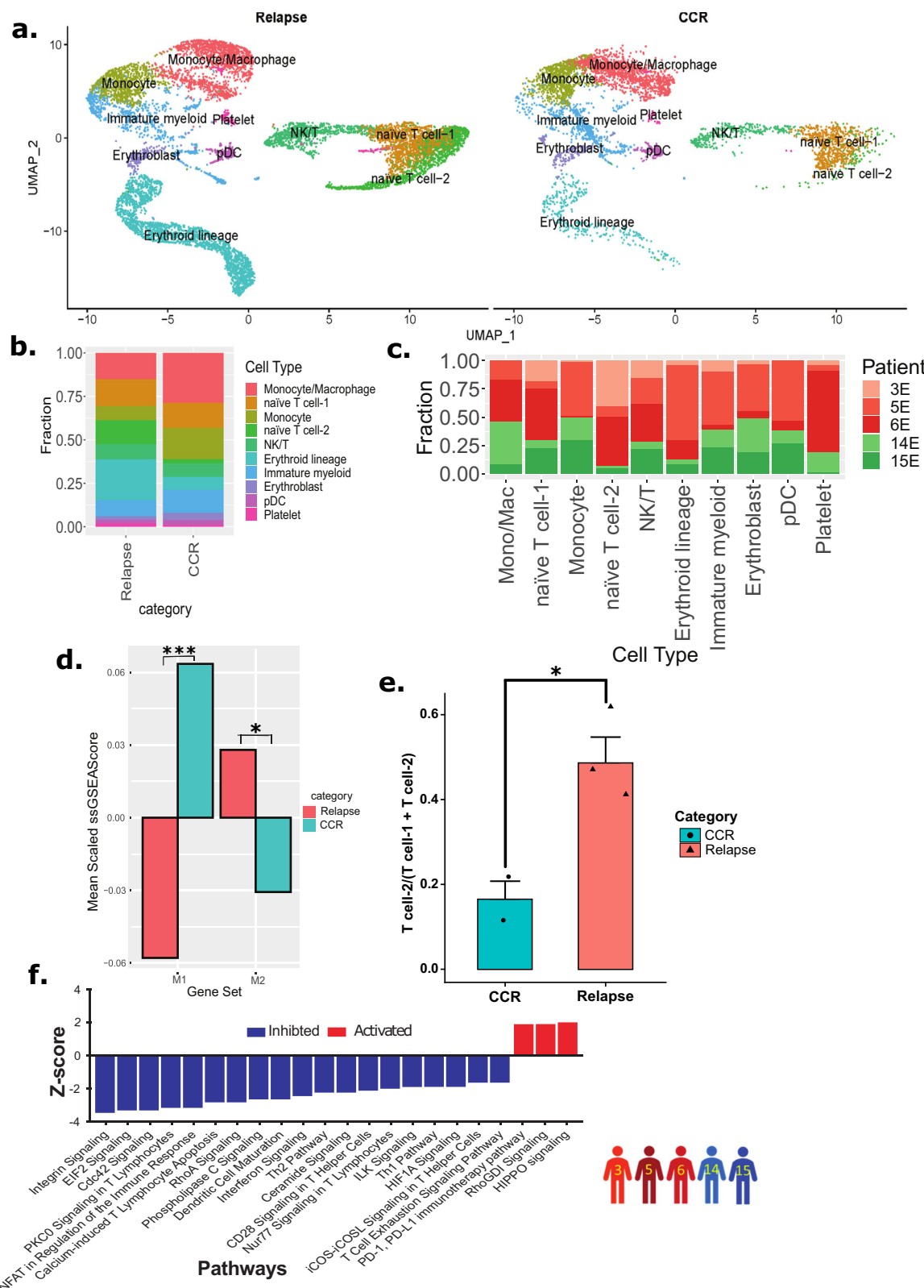

These findings support our single-cell results of M1 Macrophages and naïve T-cells enrichment in the better prognosis CCR samples and exhausted T cells enrichment in poor prognosis relapse samples.

## Discussion

ScRNA-seq analysis, which has revolutionized cancer research, can provide deep insights into AML tumor and microenvironment heterogeneity. While scRNA-seq studies have provided critical insights into adult TME, more studies are required specifically for pediatric AML. Van Galen et al. reported changes in blasts as well as differentiated cells in the TME of adult AML patients using snRNA-seq and genotyping assays[4]. Another study using scRNA-seq analysis of pediatric leukemias identified dysfunctional innate and adaptive immunity in the AML TME[7]. There are substantial differences in

**Fig. 7 | EOI non-blast cells analysis exhibits different patterns in samples from those with relapse and those with CCR. a** UMAP plot of annotated non-blast cells clusters split based on relapse- and CCR-association status. **b** Bar plot showing the proportion of cell types in relapse- and CCR-associated samples at EOI. The Y-axis represents the fraction of each cell type. **c** Bar plot showing sample/patient fraction (Y-axis) in major cell types (X-axis). **d** Bar plot of mean scaled ssGSEA scores of mono/mac cells from those with relapse and those with CCR (CCR: $n = 2203$ cells from $n = 2$ biologically independent samples, relapse: $n = 2416$ cells from $n = 3$ biologically independent samples) show that M1 markers are significantly more enriched in patients with CCR (two-sided t-test, $P = 3.8e{-}05$, ***$P < 0.001$) and M2 markers are more enriched in those with relapse (two-sided t-test, $P = 0.046$,

$*P < 0.05$). Source data are provided as a Source data file. **e** The bar plot shows that the ratio of the naive T cell-2 cluster to total T-cells (i.e., naive T cell-2 and naive T cell-1) is significantly higher ($P = 0.0232$) in samples with relapse ($n = 3$) compared to samples with CCR ($n = 2$) (two-sided t-test, $*P < 0.05$). Bars represent the mean values, with error bars representing the mean + SEM values. Source data are provided as a Source data file. **f** Bar plot of significantly activated (Z-score >1.5) or inhibited (Z-score < −1.5) pathways in T cell-2 cluster (relapsed-enriched, $n = 1538$ cells) compared to T cell-1 cluster ($n = 2272$ cells). The significantly (One-tailed Fisher's exact test, $P < 0.05$) activated and inhibited pathways are shown with red and blue colors, respectively.

pediatric and adult AML arising from cytogenetics and molecular abnormalities[58,59]. Using different ages of leukemic cells, mice models established to represent infant, childhood, young adult, and middle age, displayed differential interaction between age-defined leukemic blasts and stroma[60]. A specific transcriptional landscape that distinguishes pediatric disease biology was also revealed[60]. Ontogeny-dependent susceptibility for transformation by fusion genes has been shown to determine the clinical phenotypes, which can be reversed by targeting the fusions to overcome the differentiation blockage[61]. It was found in an inducible transgenic mice model that rapid mega-karyoblast AML was induced upon expression of *ETO2-GLIS2* in fetal HSCs compared to shift to myeloid transformation with very slow leukemogenesis upon expression of the fusion gene in adult HSCs[61]. Given the differences between adult and pediatric AML it is critical to conduct research specifically with pediatric AML patient samples to provide an accurate map of disease pathology at the cellular and molecular levels.

Our analysis of BM samples from pediatric AML patients comprising of various genetic subtypes (Supplementary Table 1) revealed the AML-blast cells are heterogeneous and patient-specific (Fig. 3a). Supervised analysis on blast cells using a machine learning approach identified a 7-gene blast progenitor signature that is overexpressed in the AML-blast populations studied. The 7-gene signature included genes coding for growth factors (*CLEC11A*), myeloid differentiation factor (*AZU1*), cell development (*NREP*), and transcription factor (*PRAME*). TRH has been shown to have high expression in AML[62] while the function associated with ARMH1 is not known yet. The functional variability of the 7-gene signature capable of distinguishing AML-blasts from non-blast cells is suggestive of multiple synergistic pathways being modulated in AML-blasts. Significant association of the 7-gene signature overexpression with poor OS in some AML patients can be attributed to the associated regulation of myriad cell functions that promote tumor growth, proliferation, and migration[3,13,18–21]. This 7-gene signature identified blast cells on a validation set of additional pediatric AML samples as well as on an external pediatric AML dataset strengthening its utility as a pediatric AML diagnostics and prognostics biomarker. Comparative gene expression analysis with healthy BM and HSCs also depicted that 7-gene signature is specifically overexpressed in the AML-blasts, strengthening its association with leukemogenesis. Analysis of longitudinal samples (Dx, EOI, and Rel) showed that blast cells identified at Dx reduced significantly at EOI and reappeared in Rel samples. The blast cells in the samples collected at relapse and disease diagnosis depicted significant similarity indicating expansion of certain therapy-resistant rare Dx blast cells clones that might be responsible for disease relapse. We postulate that after further validation on larger datasets, the 7-gene signature can be used in conjugation with current immunophenotyping-based AML diagnosis and prognosis with better accuracy.

Further scRNA-seq data analysis revealed distinct gene expression profiles at Dx of blasts from patients that are destined to relapse vs. patients that remain in CCR. The blast cells of relapse-associated samples even at the time of disease diagnosis depicted upregulation of multiple genes related to cytoskeleton formation (*FLNA, RFLNB/*

*FAM101B*) associated with poor clinical outcome. Cytoskeleton proteins affect various tumor cell characteristics like cellular proliferation and migration to promote drug resistance, invasiveness, and metastasis[63]. Also, actin cytoskeleton remodeling can change physical and functional interactions at the immunological synapse[64]. Microtubule targeting or disrupting agents have been tested for a variety of therapy-resistant cancers[65]. RFLNB belonging to the refilin family, are short lived F-actin regulators, that control cell fate specification and differentiation[66]. RFLNB promoter binds transcription factor PKNOX1/PREP1 that regulates embryonic development and determines cell fate[67]. RFLNB and PKNOX1 are also known to modulate responsiveness to TGFβ and promote epithelial–mesenchymal transition[68]. RFLNB targets FLNA to organize a perinuclear network and stabilize nuclear shape[69]. FLNA is known to interact with integrin, ITGB7 that interacts with VCAM1[70]. Therefore, overexpression of *FLNA, RFLNB* in relapse-associated blasts cells may modulate the cell structure and subsequent interactions with immune cells. These changes may influence cell junction organization, cell motility, and activate nuclear transcription programs that may contribute to AML relapse. The genes over-expressed in the relapse-associated blasts cells were found to correlate with the upregulation of multiple pathways including RhoGDI, eNOS, and protein kinase A signaling. Rho GDP-dissociation inhibitor (RhoGDI) pathway activation has been shown to enhance chemoresistance in cancer cell lines and animal models[71]. Studies also show that endothelial NOS (eNOS) can affect tumor processes such as resistance, angiogenesis, invasion, and metastasis. An increase in eNOS expression is found to have associations with poor relapse-free and overall survival as well[72]. In contrast, Dx blasts from samples with CCR showed significant upregulation of genes like *MPO* and *TRH* that are associated with better survival outcomes. Previously *TRH* expression has been shown to be increased in t(8;21) AML subtype and this high expression is associated with better clinical outcomes[73]. TRH has been implicated to have a role in the immune system[74], but its role in AML-blasts is not clear. MPO is the key enzyme of myeloid lineage commitment and is associated with a favorable prognosis in AML[55]. These gene-level differences identified in relapse- and CCR-associated blasts and subsequent differential regulation of pathways and upstream key transcriptional regulators may contribute to differences in hematopoietic niches that result in AML relapse/CCR. We observed CNAs in multiple chromosomes indicative of AML-associated chromosomal aberrations. Future in-depth CNA analysis along with correlation to transcriptome changes in pediatric AMP patients might provide a better understanding of the role of copy number alterations in disease onset, progression as well as post-therapy response.

In addition, the investigation of paired Dx and EOI samples led to the identification and characterization of residual blast cells after treatment and provided the opportunity to characterize them. The residual blast cells depicted overexpression of multiple genes associated with tumor growth (*SELENOP*), metabolism, and stemness (*HOPX, FAM30A*). The higher expression of *HOPX* is associated with higher platelet counts, lower white blood cell counts, lower lactate dehydrogenase levels, lower complete remission rates, and poor overall survival[53]. *SELENOP* encodes for a selenoprotein that mediates

microenvironmental changes that influence tumor growth and prognosis across multiple solid tumors such as prostate and breast cancers[54]. Similarly, the *FAM30A* gene has been recently shown to be associated with leukemia stemness and high-risk pediatric AML[57]. The pathways analysis of post-therapy residual blast cells showed upregulation of multiple pathways including fatty acid oxidation, and PPARα network. Fatty acid oxidation has been linked to enrichment and survival of leukemic stem cells resulting in disease progression and relapse[75]. Furthermore, the resistance of AML to conventional and metabolically oriented therapies (Venetoclax) has also shown an association with upregulation of fatty acid oxidation in the leukemic stem cells (LSCs)[76,77]. Therefore, our preliminary data support targeting fatty acid metabolism in AML LSCs for achieving CCR, though more validation is required.

While AML is mainly characterized by enrichment of immature myeloblasts, immune cells in the TME play an integral role in the disease progression and outcome[78]. Mature myeloid lineage cells have been implicated in altering T cell phenotypes and generating a immunosuppressive microenvironment in AML[79,80]. In addition, dysregulation of crosstalk between tumor and stromal cells has been shown as pivotal toward tumor proliferation or suppression. Variations in immune contexture have been shown to be associated with genomic and microenvironment factors in hematological malignancies[28]. To explore the AML microenvironment at baseline i.e., at disease diagnosis, we performed focused analysis on immune cells in Dx samples from patients with CCR and relapse outcomes. Overall, T cells were more abundant in Dx BM microenvironment of patients with relapse while monocyte/macrophages were enriched in patients with CCR. CCR-associated samples depicted naive T cells enrichment compared to relapse-associated samples, which may contribute to effective resolution of the blast cells post-therapy. Samples from those with relapse with enriched exhausted T cells at Dx may have an immunosuppressive BM microenvironment resulting in less effective posttherapy blast cells clearance. Also, pathway analysis showed enhanced Th1 pathways and costimulatory ICOS/ICOS-L T helper pathway indicative of a more inflammatory T-cell pattern in samples from those with relapse. ICOSL expression in the AML cells has been shown to result in ICOS+ Tregs expansion that promotes immune evasion as well as IL-10 secreted by ICOS+ Tregs support AML cells proliferation[81]. Further analysis of samples collected at EOI suggested a similar pattern in immune cells enrichment at EOI and Dx i.e., there were more T cells in samples that relapse later and more inflammatory monocyte/macrophages in samples with sustained remission. The transcriptionally distinct naive T cell cluster enriched in EOI samples from those with relapse (T cell-2) may be influenced by the type of residual blasts present in these samples. The analysis of innate immune cells depicted enrichment of inflammatory M1 macrophages in Dx and EOI samples of patients with CCR compared to samples of patients with relapse, indicating that the presence of these adaptive immune cells may contribute to remission as well maintain it. Leukemia-associated macrophages (LAM) have been associated with survival and drug resistance of AML cells[80]. It has been shown that targeting IRF7/SAPK/JNK pathway can promote M1 characteristics in LAM resulting in prolonged survival of leukemic mice[82]. Conversely, relapse-associated EOI samples had significantly fewer M1 macrophages that may not clear up the residual blast cells that drive disease relapse. These findings indicate a potential role of innate immune cells in achieving sustained remission and exploration for therapeutic intervention.

In conclusion, common transcriptome/pathways present in the various AML genetic sub-types enabled the development of a 7-gene signature for diagnosis and prognosis of pediatric AML. At the same time, distinct differences were identified in AML-associated blasts as well as non-blast cells that distinguished samples with relapse from samples with CCR. We uncovered the presence of patient-specific blasts at Dx, post-therapy residual blasts at EOI, and the presence/

activation of different arms of the immune system (monocyte/macrophage and T cells) in pediatric AML samples from those with relapse and those with CCR at both Dx and EOI. The identification of upregulation of less studied genes like *ARMH1* in AML-blasts is important for diagnostic purposes and may have therapeutic potential. Using scRNA-seq data analysis we were able to identify AML-blast-specific gene signatures, identify and characterize AML-associated blasts and delineate immune cells make-up in the BM microenvironment of children with different clinical outcomes.

## Methods
### Clinical samples
Samples were obtained as part of a precision medicine study of Aflac Cancer and Blood Disorder Center from Children's Healthcare of Atlanta Pediatric Biorepository that was approved by the institutional review board at Emory University, Atlanta, GA. Signed consent was provided by parents or legal guardians for pediatric patients to permit the use of biological material in accordance with a protocol that was approved by the institutional review board. BM samples were obtained at the initial presentation of AML (Dx), at the time of therapy remission; post-therapy (EOI), and at the time of relapse (Rel) as part of routine hematopathology evaluation. The study was performed on 20 patients, out of which 14 patients remained in CCR, and six patients relapsed. To identify the AML-blasts associated gene signature, we analyzed scRNA-seq data of Dx and EOI samples from four patients (patients 3, 5, 6, 14) and the TARGET AML RNA-seq dataset 1 (TARGET AML-1) (https://www.cancer.gov/ccg/research/ genome-sequencing/target). The 7-gene signature established using Dx, EOI matched samples from patients 3, 5, 6, 14 was validated using paired Dx, EOI single-cell profiling data from patients 16–20. In addition, we also validated the diagnostic performance of the 7-gene signature in a publicly available dataset containing eight AML and four healthy human BM samples[7]. We also performed comparative analysis on normal HSC by downloading Human Cell Atlas HSC scRNA-seq dataset from https://data.humancellatlas.org/explore/projects/cc95ff89-2e68-4a08-a234-480eca21ce79. We further validated the 7-gene signature expression in another set of samples from TARGET AML resource, i.e., TARGET AML-2; normal BM ($n = 324$), ≤30% blasts ($n = 169$), 30–60% blasts ($n = 342$), ≥60% blasts ($n = 826$) (https://www.cancer.gov/ccg/research/genome-sequencing/ target). Samples that did not have associated BM blast percentages ($n = 61$) were excluded from this analysis. The focused single-cell analysis at Dx were performed on samples from 14 patients: six with relapse (1D–6D) and eight with CCR (7D–14D). For EOI, the analysis was performed on five patients' samples: three with relapse (patients 3, 5, 6) and two with CCR (patients 14, 15). The independent validation of immune cells signature was performed in two publicly available RNA-seq datasets: 1. Fornerod et al., AML dataset[56] ($n = 132$), 2. TARGET AML-2 dataset ($n = 1398$).

### Single-cell RNA sequencing libraries preparation and sequencing
We performed scRNA-seq of viably thawed BM samples processed using drop-seq approach, which captures single cells along with uniquely barcoded primer beads together in oil droplets enabling large-scale parallel single-cell transcriptome studies. The single-cell suspensions were processed using a 10x Genomics workflow that generates digitally barcoded stable and uniform single-cell droplets for cell lysis and library preparation using a 10x Chromium Controller. The libraries were prepared using the 10x Genomics Single Cell 3'v3, Chromium Next GEM single cell 5'v1 reagent kits. Sequencing was performed using the massively parallel sequencing on the Novaseq 6000 platform. We captured the expression of ~1000–2000 genes per cell.

## Single-cell RNA sequencing analysis

Raw scRNA-seq data was demultiplexed, aligned to the human reference genome (hg38), and processed for single-cell gene counting using the Cell Ranger Software from 10x Genomics Inc (v3.0.2). The single-cell count data was normalized using the SCTransform function in Seurat v3.0 Bioconductor package[83] that uses regularized negative binomial models for normalizing sparse single-cell data. The normalized expression profiles of the samples were merged, and batch corrected (if required) based on shared embedding using harmony Bioconductor package[84]. Further quality control, preprocessing, unsupervised, and supervised analysis of the data was performed using various R and Bioconductor packages. The quality filtering on scRNA-seq data was performed by multiple filtering parameters including filtering out cells with >25% of mitochondrial genes and lower genes expression capture (<200 genes), and genes only uniquely expressed in <3 cells in the dataset.

The unsupervised analysis using principal component analysis (PCA) was performed on variable genes to identify principal components with significant variation. The PCs with significant variations were used as input for UMAP analysis, to determine the overall relationship among the cells. The cells with similar transcriptome profiles clustered together and subsequently annotated to different cell types such as T cells (*CD3*+), B cells (*CD19*+, *CD79A*+, *MS4A1*+), macrophages (*CD68*+), monocytes (*CD14*+), and other immune cells based on expression of specific well-established genes. The putative blasts cell clusters were annotated using top expressed genes. The comparative analysis of single-cell landscape of AML blasts and non-blasts during separate phases of disease (Dx, EOI, Rel) was performed using split UMAP plots, determining heterogeneity (based on clusters of cells) and enrichment of cell types. To understand differences in the TME between relapse- and CCR-associated samples, we performed a comparative analysis of the abundance of various canonical cell types (T cells, B cells, monocytes) as well as the blast cells. Further, to determine changes in cellular states or activation in relapsed and remission samples, comparative analysis of specific cell types marker genes was performed using dot plot function[83]. In an unbiased approach, cell types and subtypes signatures were generated by comparing the expression profile of target cell type with the rest of the cells using the non-parametric Wilcoxon rank test ($P < 0.01$) and fold change >1.2.

## Identifying blast-enriched genes using external pediatric TARGET AML-1 dataset

The preliminary comparative analysis of AML Dx enriched clusters and EOI-enriched clusters (mainly immune cells) identified a lengthy list of 232 AML-enriched genes that are overexpressed in AML blast cells (Fold Change >1.2 and $P < 0.01$). To reduce the false positives and identify robust AML blast-associated genes, we evaluated their pattern on the external pediatric AML dataset from TARGET initiative (TARGET AML-1 dataset). The study contains bulk RNA sequencing data generated from BM samples of more than 300 AML subjects that include infants (<3 years old), children (3–14 years old), and adolescents/young adults (15–39 years old). The raw TARGET AML-1 dataset was normalized using VOOM algorithm[85]. To explore the expression pattern of single-cell profiling identified AML-blast-associated genes in the TARGET AML-1 dataset, we binned the Dx datasets into separate groups based on percentages of blast cells (>60%, 30–60%, <30%). We hypothesized that genes that are specifically expressed by blast cells will depict a progressive downregulation pattern from high to low blast % and EOI samples. The significance of progressive downregulation from high to low blast%/EOI groups was determined using unpaired Student t-test (P). Genes with fold change >1.8 (high blast % vs. low blast % samples) and $P < 0.04$ were considered significantly associated with blast cells. The resultant genes from this analysis were further evaluated for expression in normal BM microenvironment cell types in

single-cell data from paired Dx and EOI patients. Genes with high expression in normal BM microenvironment cells were filtered out. The workflow for generation of blast-associated signatures is shown in Supplementary Fig. 11. All analysis was performed using different Bioconductor and R packages including LIMMA, and ggplot[86].

## Generation of AML blast-associated 7-gene signature using support vector machine

To identify the key genes associated with AML blasts we developed a classifier using a support vector machine (SVM) model[16,17] in the MetaboAnalyst/R package[87]. During the development of the classifier, a set of 20 genes that are overexpressed in blast cells at the single-cell level were selected as input features along with class labels (AML Blast, Non-Blast cells). The classifier was trained using normalized and preprocessed single-cell data (from paired Dx, EOI patients' samples: 3, 5, 6, 14). The classifier was trained using a linear SVM kernel along with a recursive feature elimination approach for developing the best model with the most important features in discriminating the pediatric AML-blast cells from non-blast cells. The performance of the classifier during training was determined using an out-of-bag (OOB) cross-validation approach using the MetaboAnalyst package[87]. The SVM classifier identifies the most "important" features by considering their relative contribution to the classification task, as indicated by the error rates. During the training, multiple SVM models with varying numbers of genes from 5 to 15 were generated by removing features with the least importance. The features were ranked based on their inclusion in the SVM models for classifying AML blast vs non-blast cells. The performance of classifiers was measured using threshold-independent receiver operating characteristic (ROC) analysis. The analysis assisted in identifying a minimal set of genes that can distinguish blast cells from other cells with high Area Under Curve (AUC) from ROC analysis. We selected the seven most important genes for the final model as they depicted performance comparable to models developed based on higher number of genes.

## Validating 7-gene signature in healthy BMs, normal HSCs, additional AML samples, and external AML datasets

AML-blast-specific expression of the 7-gene signature was validated by checking expression in healthy BM and normal HSCs. Healthy BM cells and HSCs were integrated with AML paired Dx, EOI samples (patients' samples: 3, 5, 6, 14) and normalized using the SCTransform function in Seurat v3.0. Batch effect corrections were made using Harmony[82]. The expression of the genes was visualized in our AML, healthy BM, and normal HSC clusters by generating violin plots.

For validating the 7-gene signature, we generated another single-cell dataset from AML BM samples of patients 16–20 (methods described above). In the validation set, blast cell clusters were identified (Supplementary Fig. 8), confirmed by expression of undifferentiated cells markers (*MPO +, CD34 +, AZU +*) and SingleR automatic annotation[22] (Supplementary Fig. 8). From the blast cells, the 7-gene signature expression was extracted and normalized to conduct an SVM classifier performance to differentiate blast/non-blast cells. The performance of classifiers was measured using threshold-independent ROC analysis. In addition, we also validated the performance of 7-gene signature in classifying AML blast and non-blast cells using an external dataset of eight AML samples[7], four healthy BM samples[7] and normal HSCs (https://data.humancellatlas.org/explore/projects/cc95ff892e68 4a08a234480eca21ce79). After preprocessing, normalization and unsupervised analysis, blast and non-blast cells were annotated based on expression of canonical normal cell markers from lymphoid, myeloid, and erythroid lineages. The blast cells were identified based on: (1) classification as myeloid progenitor cell types by SingleR[22], (2) patient or AML-specific clusters that overexpress established blast markers (*MPO, CD34, AZU1*) (Supplementary Fig. 9). After blast labeling, a 7-gene matrix of blast and non-blast cells was created for SVM

classification of blast/non-blast cells. The performance of classifiers was measured using threshold-independent ROC analysis.

Additional validation was performed using an external bulk RNA-Seq dataset from the TARGET AML cohort. We developed the TARGET AML-2 dataset by filtering out samples used during the development of the 7-gene signature. The TARGET AML-2 dataset contains 1398 AML primary BM samples and 324 normal BM samples. We downloaded FPKM values from the GDC Data Portal (https://www.cancer.gov/ccg/research/genome-sequencing/target) using Bioconductor/R packages including TCGAutils and tidyverse. The AML Dx samples were binned into separate groups based on the BM blast cell percentages: <30% ($n = 169$), ≥30%–<60% ($n = 342$), and ≥60% ($n = 826$). AML samples ($n = 61$) without information about blast cells percentages were excluded from the analysis. The Z-score of enrichment of the 7-gene signature for each sample in the dataset was calculated using Gene Set Variation Analysis (GSVA) and comparative analysis across groups performed. In addition, the variation in individual gene expression (log[FPKM + 1]) for the 7-gene signature was also evaluated by generating boxplots. The Wilcoxon rank sum test was used to perform comparisons between groups to determine significant differences in expression profiles.

### Survival analysis of AML-blast genes
To determine the association of blast genes with clinical outcomes in AML, we performed survival analysis on the bulk RNA-Seq TARGET AML-1 dataset from the GDC Data Portal (https://portal.gdc.cancer.gov/) using the web-based platform, Survival Genie (bhasinlab.bmi.emory.edu/SurvivalGenie/)[26]. The primary pediatric BM ($n = 145$) were collected at Dx with available clinical survival data and normalized FPKM expression counts. We examined the association of both the combined and individual effects of blast genes on clinical outcomes in the TARGET AML-1 cohort. For gene analysis, we used the median (50th percentile) cut-off or an optimal cut point (cutp) estimated based on martingale residuals[88] using the 'survMisc' package in Bioconductor/R to separate the patients into high and low gene expression groups. For combined analysis, we used the non-parametric method, ssGSEA, as implemented in the GSVA package[89] to obtain a gene-set enrichment score to obtain a gene-set enrichment score. Tumor samples were categorized into high and low gene expression groups using either the median or a cutp of the ssGSEA score as cut-off. OS or EFS statistical analysis was performed using the 'survival' R package. Kaplan–Meier survival curves were used to estimate the OS/EFS with survfit function and a log-rank test was done to compute differences in OS between the high and low gene expression groups. Univariate analysis with Cox proportional hazards regression was performed on the dataset using the coxph function. The results were considered significant if the P values from log-rank and Wald test were below .05.

### Single-cell copy number alteration analysis
Malignant phenotype of the AML blast cells is associated with somatic copy number alterations resulting in gene amplifications and gene deletions. In this study to predict the copy number alterations in blast cells identified in single-cell data, we used InferCNV algorithm, v1.3.3[23,90]. The algorithm determines dysregulations of genes across chromosomal positions in the tumor cells and normal cells to identify regions in blast cells that are over- or less-abundant as compared to normal cells. We performed analysis on initial dataset of paired DX and EOI samples as well as on DX only samples also. The analysis was performed on the blast cells identified based on overexpression of 7-gene signature. CNA profile of healthy BM and HSCs were used as a reference, the reference sample's expression is subtracted from the observational AML cells to retain the relative expression intensities across each chromosome which represents the final CNA profile of the AML-blasts consisting of chromosomal deletion/duplication events.

We have also performed analysis on individual patients by using blast and normal immune cells as observational and reference cells, respectively. Gene expression intensities were represented in a heatmap where genes in the scRNA dataset were sorted by genomic position and were further ordered within each chromosome. Using the InferCNV output, we calculated a CNA chromosome count, which represents the number of chromosomes predicted to have a copy number alteration, for each cell. This value can range from 0 to 21 (chromosome 6 is excluded from this count due to the association with differential HLA gene expression) and was plotted on a UMAP to show the distribution of cells with predicted alterations.

### Pathway and systems biology analysis
To precisely characterize the AML blast cells and understand the molecular mechanism of disease progression, we performed pathways enrichment and systems biology analysis. Pathways and systems biology analysis was performed using the Ingenuity Pathway Analysis software package (IPA 9.0) (Qiagen). A detailed description of IPA analysis is available at the Ingenuity Systems' website (https://www.ingenuity.com). Furthermore, systems biology analysis was performed using upstream regulators enrichment approach to identify the upstream transcriptional regulators that can explain observed transcriptome changes. Regulatory analysis helps in identifying significantly activated or inhibited transcriptional regulators based on upregulation or downregulation of its target genes. The significance of transcriptional regulators activation/inhibition was determined using one-tailed Fisher's Exact test. The regulators with $P < 0.01$ and absolute Z-score of ±1.5 were considered statistically significant.

### Gene set enrichment analysis
In addition to pathways enrichment analysis to identify subtle concordant changes in the gene sets, we performed gene set enrichment analysis. Single sample Gene Set Enrichment analysis (ssGSEA) score represents the degree to which the genes in a particular gene set are coordinately up- or down-regulated within a cell cluster, which could serve as an indicator of gene set activity. To interrogate the pathways/gene sets that are differentially enriched in one cell type vs. other, we performed statistical analysis using student t-test or LIMMA[91]. The gene sets enrichment analysis was performed on pathways and gene sets were obtained from Molecular Signatures Database v7.5[46] or customized set like our AML-blast 7-gene signature. Here ssGSEA was performed using escape package[92] from the Bioconductor library and pathways/gene sets with $P < 0.01$ being considered statistically significant.

### Validation of outcomes associated immune cell gene signatures in bulk RNA-seq datasets
We assessed enrichment of immune cells/phenotypes derived from scRNA-seq analysis into two independent bulk RNA-seq datasets: Fornerod et al., AML dataset with 132 samples and TARGET AML dataset 2 (https://www.cancer.gov/ccg/research/genome-sequencing/target) with 1398 samples. The Fornerod et al., AML dataset includes LSC6 risk score[57] information for each sample and corresponding LSC6 risk categories [low-risk ($n = 102$), medium-risk ($n = 27$), high-risk ($n = 3$)]. We calculated the LSC6 score for the TARGET AML-2 samples and stratified them into risk categories based on the same thresholds used in the Fornerod et al. AML dataset. The LSC6 score for the TARGET AML-2 samples was calculated using the formula from Elsayed et al.[57]. The dataset contained 175 low-risk (LSC6 score <0.95), 467 medium-risk (LSC6 score ≥0.95 and <1.9), and 756 high-risk (LSC6 score ≥1.9) samples. We tested the enrichment of M1 Macrophage (*S100A8, S100A9, S100A12, TYROBP, VCAN, CD68, MNDA, CYBB, STAT1*), naive T-cell (*CCR7, LEF1, TCF7, SELL*), effector T-cell (*CCL5, NKG7, GNLY, GZMA, GZMK, NFACT1*), exhausted T-cell (*HAVCR2, LAG3, PDCD1, NFATC1, TIGIT, TOX*), and regulatory T-cell (*CCL5, KLRB1, KLRD1, GZMH,*

*CD69, CD44*) in the bulk RNA-seq datasets by calculating ssGSEA score for each immune cell type. The Pearson's correlation was calculated between the LSC6 and ssGSEA scores for each immune cell type, and the corresponding *P*-values to determine associations. We also evaluated the association between categorized scores (i.e., low-, medium-, and high-risk) and ssGSEA immune cell enrichment scores by performing Wilcoxon signed-rank test.

## Reporting summary
Further information on research design is available in the Nature Portfolio Reporting Summary linked to this article.

## Data availability
The raw scRNA-seq data generated in this study have been deposited in the Gene Expression Omnibus database under accession code GSE235923. Due to IRB guidelines and patient/legal consent constraints, raw sequencing files in the FASTQ format are not publicly shared. However, this data can be provided upon request, subject to appropriate IRB approval and data transfer agreement. An interactive data resource and analytical tool developed based on this AML single-cell data is available online at https://bhasinlab.bmi.emory.edu/PediatricSCAtlas/. Publicly available datasets utilized in this study are available via the National Cancer Institute GDC Portal (https://portal.gdc.cancer.gov/projects/TARGET-AML) and the St. Jude Genomics Platform (https://platform.stjude.cloud/data/publications)[56]. Source data are provided with this paper.

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

## Acknowledgements

The study was supported by funding from Cure Childhood Cancer Foundation (Aflac Precision Medicine Program, D.K.G.) and Emory startup funds (M.B.).

## Author contributions

H.S.S., S.I.P., D.D., S.R., M.P., R.J.S., S.M.C., D.S.W., C.C.P., and D.K.G. collected BM samples and clinical data. B.E.T., S.S.B., and M.B. designed the experiments. B.E.T. and S.S.B. conducted the experiments. U.K., G.B.U., P.P., H.M., B.E.T., D.S., S.S.B., B.D., D.K.G., and M.B. analyzed data. B.E.T., U.K., G.B.U., H.M., P.P., D.S., S.S.B., B.D., R.J.S., S.I.P., C.C.P., and D.K.G. interpreted results and wrote manuscript. M.B. supervised the study.

## Competing interests

M.B. serves on the board of Canomiks Inc. as chief scientific advisor and has equity in it. D.K.G. and D.D. hold equity in Meryx Inc. S.S.B. serves as CEO of Anxomics LLC and has equity in it. The remaining authors declare no other competing interests.
