## [Peer Review File · Nature Communications]

Reviewers' comments:

Reviewer #1 (Remarks to the Author): Expert in leukaemia immune microenvironment and genomics

In this manuscript, Ulukaya and Thomas et al., have undertaken an analysis of pediatric AML cases and tried to i) build a 7-gene signature to correlate prognosis in pediatric AML ii) analyze the differences between blasts in patients with CCR and relapse, and iii) analyze differences between immune subsets in patients with CCR and relapse.

The aim i) is geared towards understanding the differences between AML blasts and healthy cells in the bone marrow environment, which is of major interest in the fields of hematology and tumor biology. In non-pediatric AML, this has already been addressed for example by Van Galen et al., (Cell 2019). It is a bit worrying that the authors derive their gene signature by comparing AML blast cells to non-AML-blast cells (including immune cells, stromal cells, and erythrocyte contamination cells) which is a different task than comparing AML blast cells to healthy hematopoietic stem cells, latter of which could be considered of higher interest.

In their analysis pipeline set in aim i), there is a major concern of leakage of training data into the validation data. In other words, the authors have used the same data set to derive the signature and then validate the signature with the same data set. The authors first derive their 7-gene signature by comparing AML blast-cells to non-AML-blast cells to derive 232 genes. These genes were then compared to a larger TARGET bulk-RNA-seq data set, in which the patients were divided into three bins based on the blast percentages after which the 244 genes were filtered to feature only genes associated with patients with the highest blast percentage (>60%). After that, it is unclear what has been done to later define 20 genes and then 7 genes. Nevertheless, the 7-gene signature score is then validated i) in the same TARGET data set and is of course associated with an inferior outcome and ii) in the same scRNA-seq data.

The aim iii) is currently presented on an abstract level, where e.g., the received clusters are unnamed making following the arguments cumbersome. No orthogonal omic data or validation cohorts are used, even though multiple AML cohorts profiled with scRNA-seq are already available (including Van Galen et al., Dufva and Pölönen et al., and Petti et al. publications).

Based on above and detailed comments below, the statistical rigidity of the analysis described in the manuscript is unclear. Many different selection criteria should be explained better so that the reader would not worry about “cherry-picking” the major findings featured in the paper.

Major criticism:

- 1) The MZB1+ blast cluster overexpresses several B-cell markers, including CD79A, and JCHAIN (in Supplementary Fig 2c). Given that in Fig. 1a the authors have not identified a pre-B cell or plasma cell cluster that could be identified from pediatric bone-marrow, and in Fig. 1d the clusters seem to rather increase in the EOI-time point, and in the text, it is mentioned not to be patient-specific, have the authors considered that this population could be normal plasma cells and not blast cells at all?
- 2) The use of patient samples is confusing. E.g., in Fig. 1c, only 8 samples are shown with a bit ambiguous annotation (D possibly reflects Dx and E reflects EOI?). In supplementary figure 1, the study design shows 13 DX patients and the text described 14 Dx samples and 5 EOI samples.
- 3) According to tables S1A and S1B, all end-of-induction samples are MRD negative. How is the analysis in “Characterization of treatment-resistant residual cell population at EOI” possible if there are no malignant cells left at this timepoint? What method was used to assess MRD? Does this indicate that the 7-gene signature may also detect normal myeloblasts or monocyte/macrophages?
- 4) In the text, in multiple points, the word “enrichment” is used, but the used method for counting of enrichment is not stated (Mann-Whitney, Fisher's, O/E, OR?) leaving the reader unaware of the magnitude of the enrichment.
- 5) Related to data shown in Fig. 1f, the true goal introduced in the Introduction in terms of finding biomarkers associated with AML blast cells is to find the genes expressed preferentially by AML cells in comparison to primitive, healthy hematopoietic stem cells (HSCs), not related to immune infiltrate, stroma, or erythrocytes as the authors have here done. It is unclear whether the now discovered genes are also abundantly expressed by healthy HSCs. The 7-gene signature should be tested in healthy bone marrow scRNA-seq datasets e.g. Human Cell Atlas and similar to evaluate if the signature is enriched also in normal myeloblasts/progenitors.
- 6) Fig. 2a; there does not seem to be any difference between the high blast (>60%) and the 30-60% blast categories, even though the 20 genes come from 44 genes that were upregulated in this bin. The utilized test is not mentioned.
- 7) It is unclear how it is ended from the 44 genes to 20 genes. The authors state that these 20 genes were upregulated in the AML blast cells in the scRNAseq data; however the original 232 DEGs were already defined like this? The authors should make a diagram out of their selection process for the genes which would explain what selection criteria were used and where.
- 8) Fig. 2d, these genes were DEGs from patients with high blast percentages. Is this overall survival significance retained when the number of blasts is taken into consideration?
- 9) Fig. 2e-f, it is unclear whether these patients were included in the original analysis seen in Fig. 1 and in the cells which were used to derive the 7 gene score. It seems that patients 5 and 6 were used in both analyses. This raises critical concern whether there might be spilling of the training and testing data sets and in that manner the validity of the 7 gene signature is questionable.
- 10) Fig. 2g, how the authors derived the features from SVM is not defined in the text.

- 11) Fig. 2h is validating the gene set score in the same data set from where it was derived, which is unacceptable as it is a circular argument. The authors ought to validate their gene set in a separate cohort or split the TARGET cohort into bins.
- 12) Fig 3a-b, it seems that the same samples that were used to derive the 7-gene signature were again subjected to validation in the same cohort from which it was derived.
- 13) Fig 3a-b, the clusters are patient-specific. Hence it is assumed that CCR and Relapse samples group differently.
- 14) Fig 3c, it is unclear why some relapse and some CCR associated clusters were included or not included.
- 15) In “Comparative analysis of Dx relapse- and CCR-associated samples”, the authors should assess how genes enriched in certain genetic subtypes may explain differences in relapsing vs. CCR patients. E.g. relapsing group includes 3 MLL and CCR group 3 RUNX1 and 3 NPM1 cases known to influence prognosis. The association of identified genes to prognosis in the TARGET data should be tested in multivariable analyses which consider known prognostic markers such as driver mutations/translocations to find out if these genes reflect the genetic subtypes or provide independent prognostic information. Also, it would be useful to examine the expression of the identified genes in the scRNA-seq data stratified by genetic subtype to find out which genes are subtype-enriched, and which more generally expressed.
- 16) In “Inflammatory monocytes/macrophages enriched in CCR-associated samples”, how have the authors made sure that they are analyzing only non-AML monocytes? Given that cluster 7 ‘overexpressed many of the 20-gene signature genes including PRAME, CITED4, CLEC11A, KCNE5, and MFSD10, unlike clusters 1 and 2, indicating the immaturity of these cells’, is it possible that some of these are leukemic monocytes?
- 17) Fig 4a-c is supposed to be an analysis of non-AML cells. Why was the MPO+ blast cell cluster included?
- 18) The overall point of Fig. 4 is unclear and could be merged with Fig. 5 and supplementary figures.
- 19) In figure 4, percentage values are shown. Was the number of cells from each sample similar? If not, wouldn't that impact the percentage values?
- 20) Fig 5a-b the clusters are not named, and the markers are buried in the Supplementary Figure 8b, making following the story extremely cumbersome for the reader and leaving the results on an abstract level.
- 21) Fig 5a, it is unclear why analysis on T-cells was performed on another algorithm, Monocle, rather than with the algorithm used in the other analyses, Seurat.

Minor criticism:

1) "The unsupervised analysis on preprocessed and normalized scRNA-seq data using Uniform Manifold Approximation and Projection (UMAP) approach (4) identified 14 single cell clusters"

The UMAP does not identify clusters, but clustering does. Please amend.

2) The clustering identified in Fig. 1a seems to under cluster different immune subset populations, as T-cells are not separated into CD4+ or CD8+ lineages, or even between NK and T-cells.

3) Fig. 1e is repeating the data from Fig. 1b and adds little info to it. MPO and CD34 are said to be clinically validated blast cell markers. This probably refers to flow cytometry and staining analysis, but are there any clinically validated blast cell markers from the transcriptomic data?

4) The labels in Fig 1c are unclear

5) Fig 1f label, the term "expression" is ambiguous and the labeling "pseudocolor" is also uncommon. The reason to highlight some genes is not mentioned.

6) Fig. 2a The x-axis labels are unreadable. The highlighted genes (n=6) are not explained in the labels

7) Fig. 2e the numbers are not explained.

8) "including CLEC11A (7), PRAME (8), and AZU1 (9) have been previously associated with AML" is a vague statement, please amend.

9) Suppl Fig 5a-b, it is unclear how the module score was defined.

Reviewer #2 (Remarks to the Author): Expert in paediatric AML genomics

This is an important study that will have impact on the AML field, highlighting features of paediatric AML not previously known. The paper is a computational study, deep diving into the molecular features of AML from single cell transcriptomics data. The power lies with the longitudinal samples from diagnosis, end of induction treatment and relapse patients. It is continuously stated (as a negative point in most cases), that AML is heterogenous. This study uses samples from patients from all different cytogenetic backgrounds and in fact treatment schedules. The single cell approach was able to pull out important signatures, some with prognostic power, from these samples. This I believe is important to highlight and a step forward in combatting those negative comments that "AML is heterogenous". With this in mind, I think the authors should make a much better effort is being more transparent in the clinical features of the samples used – highlighting and interpreting the data in the context of the cytogenetics of the samples and the treatment they received when appropriate and the results are indicating it– perhaps doing additionally computational analysis to take these questions on board.

Specific comments

1. The samples used in this study are from many different cytogenetic subgroups – how was this considered in the data analysis – for example, 2,4,5 and 6 relapsed with 4 and 5 both having MLLr and 6 having 7q. CCR EOI from 3, 5,6, 14 and 15, with 14 having t(8;21) and 15 the only patient in the study with FLT3-ITD and NPM1. What risk category were each of these patients in (the risk category should be added to table S1A). How was the difference in treatments received by the patients considered in the data analysis?
2. It would be much clearer if a schematic/table of the samples used for each sc-RNA study for each figure were being analysed.
3. On page 4, it is not indicated which patient samples were used for the scRNA-seq. It is stated that 3 from relapsed patients (4 relapsed in the study so which 3 were single cell profiled) and 1 from a patient with CCR (5 had CCR in the study so which of the 5 was single cell profiled). Figure 1 C indicates samples 3D,3E, 5D, 5E, 6E,6D, 14E and 14D (4 paired Dx and EOI samples). It should be indicated on table S1B that it was these samples that were single cell profiled by scRNA-seq. 14 CCR is an t(8;21) with only 40% blasts in the BM compared to 85-91% in the relapsed samples (1 with MLLr, 1 with 7q, and 1 with AML,NOS(not otherwise specified)).
4. The scRNA-seq was performed on the entire bone marrow samples, the advantage of which was that the analysis could determine what other cells from the stroma and immune cells were in the leukemic milieu. The downside of this is that the leukemic stem cell signatures may have been missed. From the blast percentage in table S1, the numbers varied considerably across relapsed and CCR samples. It would be important to know and specify the samples used in the single cell analysis to compare the blast percentage and this may in fact impact on the cells present in each sample and the signatures/transcripts present at the higher levels. This point should be considered.
5. In figure 1F, the color bar for EOI-enriched and Dx-enriched appear to be DX=blast populations as indicated in figure 1D and EOI=non blast populations – but in the text on the bottom of page 4 it describes the evaluation of 44 genes in the blast cell clusters – This is confusing. The legend states it is comparing the genes in blasts compared to other differentiated cell types which makes sense.
6. On page 5, it is discussed that figure 2B depicts uniform expression of 2 genes in the 20 gene signature that better distinguish blasts than those used clinically and shown in figure 1E/S4. However, in S4 the data is expression from combined Dx and EOI AML samples, whereas it is unclear if the data from 2B is from only Dx samples. The data showing separately for Dx and EOI samples, or just Dx samples for the clinically used markers Cd33 and NCAM1 would better support this statement.
7. Supporting the statement that the majority of the blast genes exhibit association with overall survival in figure 2D, table S5 is provided which is not easily discernible to ascertain what “the majority” means. A figure with the OS curves, as shown in 2D would be easier to interpret.
8. The 20-gene signature is then looked at in Dx, EOI and Rel of 2 patients – Whilst it is clear that the scRNA 20-gene blast cell profile is diminished in EOI in figure 2E, which 2 patients is this data from. It was explained that single cell sequencing was carried out on 4 patients, with only 1 other those relapsing so it is very unclear which 2 patients this data is from, what their cytogenetics are, and their blast % in the BM. Figure 2F indicates that samples 5 and 6 are used here. From deduction then, and looking carefully at table S1B, there is only 2 patients in the study where Dx, EOI and relapse samples are

used. 1 of these is an MLLr, the other is a 7q, the WHO classification is different and the treatment both received was different and their outcomes also different (deceased (sample 5) V alive (sample 6)). Looking at the analysis then in figure 2F, one can see that the blast and immune cell populations are very different between these 2 patients. I think this is relevant information that should be discussed/revealed better in the text. Also, for confirmation is this data generated in 2F using the 20-gene signature – as this should be clearly indicated in the text and legend.

9. The study then looked at the 14 Dx samples derived in figure 2G-H. The UMAP projection depicts CCR and relapse clusters but it is not clear how this data was generated? Understanding that 6 patients relapsed and 8 patients achieved CCR, how were the clusters derived from the Dx samples only? It is how the data is explained is the problem – the “CCR-Associated clusters”, are these clusters found only in the patients that achieved CCR? So they are clusters seen at diagnosis associated with CCR-patients/samples and clusters associated with patients that relapsed? And was it done using the 7 gene score? This needs to be more clearly explained if this is the case.

10. For figure 3B, are the presence of the different clusters impacted by the blast percentage in the patients rather than the clusters are not present in the patient? – in other words, a cluster may be present but undetectable because of a dominance of another cluster found in the dominant blast? So the inter-patient variability may be due to the constraining factors of the study, mainly being the different cytogenetics? Indeed there is a dominance of NPM1 positive samples in the CCR patients as shown in table S1A, and this is a gene pulled out unsurprisingly then from a CCR-associated cluster in figure 3C.

11. In figure 5B, there appears to be specific T cell clusters enriched in specific samples, again possible linked to the cytogenetics of the samples. This should be considered.

12. The data in figure 5E is not explained very well – how was this data generated?

13. In figure 5F, why were clusters 1 and 2 and 11 left out of the analysis of relapse-associated clusters, and why was 4 and 9 left out of the CCR-associated cluster gene analysis? Is this because there was a dominant cluster in a specific samples - and what were the clinical features of those samples – this should be transparent in the text. An the without a valid reason, these clusters should not be excluded.

14. When the term “non-AML cells” is used, how confident can it be that these cells are non-AML. What is your definition of non-AML? They are in fact cells part of the AML TME.

15. In the results discussing the genes enriched in cluster 7, it is clear that in S9B the analysis is of the 20 gene signature but what list of genes was interrogated in S9C – was this a manual list of immature markers?

16. Figure 6C it is stated that M2 markers MRC1, ARG1 and MMP9 (M2 markers) are mostly overexpressed in the relapse-associated samples – From the figure this is very difficult to see. The percent expression is low and the (dots are extremely small to make out with the naked eye). Importantly, CD163 appears to be overexpressed in the CCR-associated clusters, a marker linked with poor AML survival and M2 macrophages. What do the authors think of this – is a potential explanation that in fact the cytogenetics is important, these are paediatric compared to the data in the literature being from adult? This may in fact be highlighting some very relevant and unique immune properties for paediatric AML.

17. It should indicate in the text regarding sc-RNA seq for figure 7 which patients samples were used – it states 3 relapse and 2 CCR but which patients were these? Was the cluster analysis done using these samples all grouped together (figure S10, 7B)– and so the interpretation is that the clusters 6 and 9, and residual blasts remain regardless of relapse or CCR? This should be clear if so. And if that is the case, are these blast clusters important and figure S10 necessary? The survival curves in figure S11 are more important in the main figure rather than figure 7C I think.

18. A comment or additional discussion in the text highlighting how this scRNA-seq analysis places paediatric AML distinct from adult AML is warranted, and would highlight greatly the impact of this study in the AML field. Referring to functional studies such as Chaudhury et al, Nature Comms,(9), 2018 and Lopez et al, Cancer Discovery 2019 highlight such importance in line with this.

Minor comments

19. The age of the patients is missing from table S1 which is important to add.

20. It would be easier to ingest the data if table S1A and table S1B were merged.

21. The colours of the samples in figure 1C are very similar and the font of the population names is blurry/shadow. Can more divisive colors be used for C and D.

22. Figure 2A there is no ref/green color legend and the labels along the x axis are illegible.

23. Figure S7E is illegible – the string network is too squashed and gene names can't be read.

24. The legend in figure 6A has 4 colors but only 3 labels – the colour matched to labels need to be indicated. State in the legend what "Other" is.

25. The quality of figure 6B is not good. Font is over stretched.

26. In the text for a lot of the figures, a number of genes are mentioned. It would help that for each figure where a gene is highlighted in the text, that a box or color highlighting that gene along the axis labels is placed.

27. The legend for figure 7B and 7C need inverting.

Reviewer #3 (Remarks to the Author): Expert in single-cell RNA-seq and leukaemia genomics

In this manuscript, matched samples from pediatric AML cases from multiple time points (including diagnosis, end-of-induction, and either remission or relapse) were analyzed using single-cell RNA-sequencing. The authors present an AML blast gene signature and describe differences in the microenvironment at time of diagnosis between patients that ultimately relapse or reach remission.

The general approach presented here is substantively flawed. A key step in analyzing tumor samples by scRNA-seq is identifying the tumor cells. For solid tumors, this is generally done by identifying copy number alterations (CNAs) in the tumor cells. In AML, it is generally done by identifying SNVs and/or CNAs in individual cells. This was not done in this paper. Mutation/CNA identification is a critical step in any analysis, particularly in AML, because tumor cells often partially differentiate such that they express markers of more differentiated cell types. This manuscript simply assumed that AML cells are the cells that express typical markers of stemness/HSPCs, such as CD34, or a mixture of markers (which might arise from doublets). This is an invalid assumption; many tumor cells have been missed, and it is the basis of most of the analyses in this manuscript.

Aside from the above concern, there are problems with circularity in the definition of the AML blast signature. The putative blasts identified by the authors may well be leukemic blasts, but the analyses intended to identify a blast signature simply identify genes that are correlated with the markers used to identify blasts in the first place. A better experiment would be to enrich normal bone marrow samples for progenitor cells, such as by sorting for CD34+ cells, then to compare (by scRNAseq) leukemic blasts to the normal HSPCs. (Again, this would require genetic confirmation that the blasts are indeed leukemic).

The observations about macrophage differences in the pre-relapse vs pre-remission samples are affected by the same problem above. However, the T cell observations are interesting and potentially correct (because most T-cells in an AML sample are indeed real T-cells).

This project could be reworked by (1) performing exome sequencing on every sample, (2) identifying mutant cells in the scRNA-seq data to confidently and unambiguously identify tumor cell clusters, and (3) analyzing normal bone marrow samples by scRNAseq.

Reviewer #1 (Remarks to the Author): Expert in leukemia immune microenvironment and genomics

In this manuscript, Thomas et al., have undertaken an analysis of pediatric AML cases and tried to i) build a 7-gene signature to correlate prognosis in pediatric AML ii) analyze the differences between blasts in patients with CCR and relapse, and iii) analyze differences between immune subsets in patients with CCR and relapse.

The aim i) is geared towards understanding the differences between AML blasts and healthy cells in the bone marrow environment, which is of major interest in the fields of hematology and tumor biology. In non-pediatric AML, this has already been addressed for example by Van Galen et al., (Cell 2019). It is a bit worrying that the authors derive their gene signature by comparing AML blast cells to non-AML-blast cells (including immune cells, stromal cells, and erythrocyte contamination cells) which is a different task than comparing AML blast cells to healthy hematopoietic stem cells, latter of which could be considered of higher interest.

Response: In the original manuscript, we compared tentatively identified blasts cells at diagnosis (Dx) (on basis of cells that disappear at end of induction (EOI) compared to Dx) to non-blast cells at EOI to identify differentially expressed genes in the former. To evaluate the association of our signature to AML blasts, in the revised manuscript we analyzed expression in the normal HSCs

dataset from the Human Cell Atlas (HCA) initiative (<https://data.humancellatlas.org/explore/projects/cc95ff89-2e68-4a08-a234-480eca21ce79>) as well as in healthy BM control samples (Bailur et al., 2020). We observed higher expression of the genes of the 7-gene signature in the AML blast cells compared to HSCs and healthy BM controls, demonstrating the blast specificity of our signature. We have included this data as a supplementary figure in the revised manuscript (Figure S7).

In their analysis pipeline set in aim i), there is a major concern of leakage of training data into the validation data. In other words, the authors have used the same data set to derive the signature and then validate the signature with the same data set. The authors first derive their 7-gene signature by comparing AML blast-cells to non-AML-blast cells to derive 232 genes. These genes were then compared to a larger TARGET bulk-RNA-seq data set, in which the patients were divided into three bins based on the blast percentages after which the 244 genes were filtered to feature only genes associated with patients with the highest blast percentage (>60%). After that, it is unclear what has been done to later define 20 genes and then 7 genes. Nevertheless, the 7-gene signature score is then validated i) in the same TARGET data set and is of course associated with an inferior outcome and ii) in the same scRNA-seq data.

Response: To elucidate a shared gene signature that could be used to identify heterogeneous AML blast cells, we used four paired Dx and EOI samples. The initial set of 232 differentially expressed genes in AML blast cells from scRNA-seq data was then pared down to 44 genes by investigating gene expression in TARGET AML Dx samples with different blast percentages and EOI samples. To derive a more concise signature, we rechecked the expression of these 44 genes in the original scRNA-seq dataset (Dx blast cells) and selected the top 20 highly expressed blast-specific genes. The final 7-gene signature was then selected using the AI support vector machine (SVM) tool (explained in the methods section, page 17; lines 3-15) that can distinguish between AML blast and non-blast cells during cross-validation. We have clarified the signature selection process by including a flow chart showing each selection step sequentially (Figure S9).

We agree that training data and validation data should not overlap. To address this in the revised manuscript, we have tested the 7-genes signature's blast-specificity and ability to identify blast cells in a) a newly generated AML scRNA-seq dataset (patients 16-20) (Fig. 2H), and b) an external publicly available AML dataset (Bailur et al., 2020; Fig. 2I). The cells were labeled as blast or non-blast in these datasets based on expression of canonical markers, comparison with healthy BM, comparison with EOI AML samples, and patient specificity in clusters. Once these "true" labels were assigned, training and testing sets were created and SVM models were trained to predict a cell as blast or non-blast based on the expression of the 7-genes signature. These SVMs were then used to predict the labels of the testing sets and this resulted in 78.4% AUC (Fig. 2H) and 80.9% AUC (Fig. 2I) values for the additional processed and external AML datasets respectively. **The aim iii) is currently presented on an abstract level, where e.g., the received clusters are unnamed making following the arguments cumbersome. No orthogonal omic data or validation cohorts are used, even though multiple AML cohorts profiled with scRNA-seq are already available (including Van Galen et al., Dufva and Pölönen et al., and Petti et al. publications).**

Response: We have labelled the immune clusters in the revised manuscript (Fig. 4C). We have included new validation samples (new set of AML samples processed in the lab and external

publicly available datasets) in our revised manuscript. We did not compare with the already published adult datasets as transcriptome profiles of children differ significantly from adults. We did use gene markers from Van Galen *et al.*, publications for cell annotation. We validated our AML blast signature in independently generated single-cell datasets, as well as showed that the signature displayed minimal expression in the healthy BM dataset. We have referenced the findings of Van Galen *et al.*, (page 3: lines 24-27, page 12: lines 3-5) and Dufva *et al.*, (page 7: lines 17, 18; page 14: lines 13-15) publications in our revised manuscript. Petti *et al.*, used integrated whole genome sequencing and scRNAseq data to identify mutations in AML blast cells. We used inferCNV tool to identify chromosomal alterations in the AML blasts.

Based on above and detailed comments below, the statistical rigidity of the analysis described in the manuscript is unclear. Many different selection criteria should be explained better so that the reader would not worry about “cherry-picking” the major findings featured in the paper. Major criticism:

1) The MZB1+ blast cluster overexpresses several B-cell markers, including CD79A, and JCHAIN (in Supplementary Fig 2c). Given that in Fig. 1a the authors have not identified a pre-B cell or plasma cell cluster that could be identified from pediatric bone-marrow, and in Fig. 1d the clusters seem to rather increase in the EOI-time point, and in the text, it is mentioned not to be patient-specific, have the authors considered that this population could be normal plasma cells and not blast cells at all?

Response: Thanks for pointing this out. Upon re-evaluation, we deemed the cluster originally designated as MZB+ blast to be a mix of plasma cells and pre-B cells, based on canonical markers expression (*EGFL7*, *MSI2*, *JCHAIN*, *MZB1*), so we have renamed the cluster as pre-B/plasma in the revised manuscript (**Fig. 1A, B**).

2) The use of patient samples is confusing. E.g., in Fig. 1c, only 8 samples are shown with a bit ambiguous annotation (D possibly reflects Dx and E reflects EOI?). In supplementary figure 1, the study design shows 13 DX patients and the text described 14 Dx samples and 5 EOI samples.

Response: We have made the changes in the manuscript to clearly indicate what patients were included in the different analyses (clinical sample information: page 4, lines 2-11). We have also amended Figure S1 to show the patients in different colors and with numbers for clarity. The D and E abbreviations denote samples collected at time of disease diagnosis and end of induction, respectively. Similarly, R indicates the bone marrow samples collected at the time of disease relapse. We have added this information along with other clinical information in the supplementary table S1A. The original table S1A and table S1B have been combined in accordance with a reviewer’s suggestion. We have included a new supplementary table S1B with some of clinical features like risk stratification, MRD status etc., of the patients. In the revised manuscript, each figure and legend indicate patients used in the analysis.

According to tables S1A and S1B, all end-of-induction samples are MRD negative. How is the analysis in “Characterization of treatment-resistant residual cell population at EOI “possible if there are no malignant cells left at this timepoint? What method was used to

assess MRD? Does this indicate that the 7-gene signature may also detect normal

myeloblasts or monocyte/macrophages?

Response: MRD status was clinically determined at EOI and provided as clinical information in the AFLAC biorepository record. Two samples were MRD+ (patient samples 1 and 17, Table S1B). The residual blasts cells in the EOI samples were identified as cells that clustered in the same cluster as the Dx AML blasts, as seen in the split UMAP and bar plots (Azu+ and MPO+ blast clusters in **Fig 2F** and clusters 4, 6, 8 in **Fig. 6A**). These EOI blast cells (~6% of total cells) (from patients 3, 5, 6, 14, 15) might have escaped detection using MRD assay but have transcriptome profiles highly similar to blast cells identified in Dx samples, therefore we labeled them as residual therapy-resistant blast cells.

In the text, in multiple points, the word “enrichment” is used, but the used method for counting of enrichment is not stated (Mann-Whitney, Fisher's, O/E, OR?) leaving the reader unaware of the magnitude of the enrichment.

Response: Most of the times “enrichment” is used to signify an increase in a particular cell or subtype of cells in the sample set based on percentages. We have amended the manuscript to address the above issue and used enrichment to describe a significant change in pathways using single-sample Gene Set Enrichment Analysis (ssGSEA) approach. The ssGSEA enrichment score represents the degree to which the genes in a particular gene set/pathway are coordinately up- or down-regulated within a sample. The statistical details for the method have been described in the original publication about Gene Set Enrichment Analysis by Subramanian et al., 2005 (PMID: 16199517). In the revised manuscript we added details and added the appropriate citation (page 9: line 27, page 19, lines 4-15).

4) Related to data shown in Fig. 1f, the true goal introduced in the Introduction in terms of finding biomarkers associated with AML blast cells is to find the genes expressed preferentially by AML cells in comparison to primitive, healthy hematopoietic stem cells (HSCs), not related to immune infiltrate, stroma, or erythrocytes as the authors have here done. It is unclear whether the now discovered genes are also abundantly expressed by healthy HSCs.

Response: In the revised manuscript, we have included the expression of the 7-gene signature in normal HSCs as well as healthy BM from external datasets. The genes in the 7-gene blast signature are expressed at lower levels in HSCs and healthy BM samples compared to AML blast cells (Figure S7C).

Fig. 2a; there does not seem to be any difference between the high blast (>60%) and the 30-60% blast categories, even though the 20 genes come from 44 genes that were upregulated in this bin. The utilized test is not mentioned. It is unclear how it is ended from the 44 genes to 20 genes. The authors state that these 20 genes were upregulated in the AML blast cells in the scRNAseq data; however the original 232 DEGs were already defined like this? The authors should make a diagram out of their selection process for the genes which would explain what selection criteria were used and where.

Response: The selected genes are expressed highly in AML blast cells, so not much difference is seen between >60% blasts and 30-60% blasts. The differences are more obvious when compared to <30% blasts and EOI samples (**Fig. 2A, C**). We have mentioned this in the main text (page 5, lines 18-21). The comparison of high blast samples (>60% blast) with low blast (<30%) and EOI was performed using unpaired Student t-test (page 16, lines 38-41). We initially identified 232 genes from scRNAseq data, which were trimmed to 44 genes on basis of expression and survival data from TARGET AML dataset. We then went back to scRNAseq data and narrowed it down to 20 genes based on blast-specific expression and minimal expression in the immune microenvironment cells. To clearly depict the process of identification and validation of 7-gene signature, we have included a flow chart in the revised manuscript (Figure S9).

Fig. 2d, these genes were DEGs from patients with high blast percentages. Is this overall survival significance retained when the number of blasts is taken into consideration?

Response: The overall survival shown in **Fig. 2D** did not take into consideration blast percentages. The survival analysis was done using the Survival genie platform (<https://bbisr.shinyapps.winship.emory.edu/SurvivalGenie/>) with the TARGET AML dataset. No selection criteria were applied to select high blast cases for survival analysis.

Fig. 2e-f, it is unclear whether these patients were included in the original analysis seen in Fig. 1 and in the cells which were used to derive the 7 gene score. It seems that patients 5 and 6 were used in both analyses. This raises critical concern whether there might be spilling of the training and testing data sets and in that manner the validity of the 7 gene signature is questionable. Fig. 2g, how the authors derived the features from SVM is not defined in the text.

Response: **Fig. 2e-f** (**Fig. 2F, G** in the revised manuscript) is the one showing blast cells in Dx and Rel samples, with less blast cells in EOI samples. This analysis was performed on patients 5 and 6 with matched Dx -* EOI -* Rel samples. The identification of the 7-gene signature was performed using four patients (3, 5, 6, 14). The primary purpose of these figures (**Fig 2F, G**) is to show how the blast clusters (*AZU+*, *MPO+* blasts) identified as blast clusters by the 7-gene signature are dominant in Dx and Rel samples and reduced in EOI samples. We are not validating the signature here and have made changes in the manuscript to reflect this (page 5, lines 38-43). We also explained in the methods section how SVM was used to identify the 7-gene signature (page 17, lines 3-15) and for its independent validation purposes (page 17, lines 16-38).

Fig. 2h is validating the gene set score in the same data set from where it was derived, which is unacceptable as it is a circular argument. The authors ought to validate their gene set in a separate cohort or split the TARGET cohort into bins.

Response: The purpose of original **Fig. 2h** was to demonstrate that the high expression of AML blast signature genes is associated with poor survival in select pediatric AML patients to highlight the role of the gene signature in clinical outcome. In the revised manuscript, we have removed the figure and validated the blast association of 7-gene signature using additional AML samples as well as an external AML dataset and healthy BM sample scRNA-seq datasets. We have briefly described the association of 7-gene signature with poor clinical outcome in the pediatric AML in the result section (page 6, lines 34-38).

Fig 3a-b, it seems that the same samples that were used to derive the 7-gene signature were again subjected to validation in the same cohort from which it was derived.

Response: In the revised manuscript we have validated the gene signature in external AML scRNA-seq dataset of 8 samples as well as 10 new samples processed and analyzed in our lab (Fig. 2H, I). Fig. 3 is not depicting validation of the signature but rather characterizing blast cells at Dx by studying the transcriptome and associated pathways in the cell clusters identified as AML blast cells using the 7-gene signature. The four samples used for developing the gene signature were included in the characterization of TME at Dx and EOI.

Fig 3a-b, the clusters are patient-specific. Hence it is assumed that CCR and Relapse samples group differently.

Response: The visual analysis of the UMAP clusters showed that all the relapse-associated AML blasts clustered next to each other (except cluster 9) as did the CCR-associated blasts, indicating transcriptional differences between blast cells based on outcome (i.e., relapse and remission). Evaluating variation within relapsed and remission clusters indicated that most of these clusters are patient specific, highlighting the inter-patient heterogeneity of the blast cells.

3) Fig 3c, it is unclear why some relapse and some CCR associated clusters were included or not included.

For comparative analysis of relapse and CCR, we selected clusters that have cells from multiple patients and cluster together to avoid noise in the differential analysis due to large variation among clusters. In the revised manuscript, we explained this selection (page 7, lines 11-12).

4) In “Comparative analysis of Dx relapse- and CCR-associated samples”, the authors should assess how genes enriched in certain genetic subtypes may explain differences in relapsing vs. CCR patients. E.g. relapsing group includes 3 MLL and CCR group 3 RUNX1 and 3 NPM1 cases known to influence prognosis. The association of identified genes to prognosis in the TARGET data should be tested in multivariable analyses which consider known prognostic markers such as driver mutations/ translocations to find out if these genes reflect the genetic subtypes or provide independent prognostic information. Also, it would be useful to examine the expression of the identified genes in the scRNA-seq data stratified by genetic subtype to find out which genes are subtype-enriched, and which more generally expressed.

Response: In the revised manuscript, we have performed the analysis to identify the difference in gene expression based on genetic subtype (Figure S13) and expanded results section to include this analysis (page 8, lines 1-38).

5) In “Inflammatory monocytes/macrophages enriched in CCR-associated samples”, how have the authors made sure that they are analyzing only non-AML monocytes? Given that cluster 7 ‘overexpressed many of the 20-gene signature genes including PRAME, CITED4, CLEC11A, KCNE5, and MFSD10, unlike clusters 1 and 2, indicating the immaturity of these cells’, is it possible that some of these are leukemic monocytes?

Response: To further refine the annotation in the revised manuscript, we performed re-clustering the non-blast cells from the Dx samples (**Fig. 4A, B**). This cluster (cluster 6, formerly cluster 7) mainly contained cells from the samples with relapse (Figure S17A) and showed expression of progenitor cell markers (*MEIS1*, *PROM1*) (Figure S17C). We checked for the expression of leukemia monocyte markers (*ITGAM*, *CD33*, *ANPEP*) and observed very low expression in the monocyte/macrophage clusters 1, 2 and 6 (page 10: lines 11-12) (Figure S17B).

Fig 4a-c is supposed to be an analysis of non-AML cells. Why was the MPO+ blast cell cluster included?

Response: In the original manuscript, this cluster (cluster 3) displayed expression of genes like *CD34* and *EGR1*, minimal expression of 7-gene signature and high expression of MPO, therefore we annotated it as MPO+ blast cells. In the revised manuscript, we performed automatic cellular annotation using SingleR algorithm and this cluster was determined to be a mix of GMP and CMP. Based on the SingleR annotation and manual marker expression (*LYZ*, *MNDA*), we inferred that this cluster as immature myeloid cells (not a blast cluster) and renamed it in the revised manuscript (**Fig. 4A, B**).

The overall point of Fig. 4 is unclear and could be merged with Fig. 5 and supplementary figures.

Response: **Fig. 4** in the original manuscript represented the overall landscape of the non-blast cells from the 14 Dx samples, while **Fig. 5** and **6** showed detailed analysis on T-lymphocytes and monocyte/macrophages clusters. As per suggestion, in the revised manuscript we have combined **Figures 4** and **5** from the original manuscript.

In figure 4, percentage values are shown. Was the number of cells from each sample similar? If not, wouldn't that impact the percentage values?

Response: The number of cells from each sample were not similar, therefore we agree with reviewer that it will impact percentage values. It is difficult to profile same number of cells from each patient, therefore we removed this figure from revised manuscript.

Fig 5a, it is unclear why analysis on T-cells was performed on another algorithm, Monocle, rather than with the algorithm used in the other analyses, Seurat.

Response: To highlight the difference between T-cell subclusters we performed analysis using Monocle algorithm with which we observed better resolution between the relapse and CCR samples.

Minor criticism:

1) "The unsupervised analysis on preprocessed and normalized scRNA-seq data using Uniform Manifold Approximation and Projection (UMAP) approach (4) identified 14 single cell clusters"

Response: We have amended the manuscript by changing the word 'identified' to 'revealed' (page 4, line 19).

2)The clustering identified in Fig. 1a seems to under cluster different immune subset populations, as T-cells are not separated into CD4+ or CD8+ lineages, or even between NK and T-cells.

Response: In the **Fig. 1A**, to observe overall clustering patterns of cells, we performed clustering at resolution value of 0.5. This clustering segregated cells from myeloid, lymphoid, and erythroid lineages clearly. Clustering at a higher resolution resulted in formation of smaller subclusters for blast cells, therefore we elected to proceed with 0.5 resolution for this initial analysis. To understand the association of different types and subtypes of T cells, we have performed focused analysis on T cells (**Fig. 4C**) that differentiates clusters for CD4+, CD8+ and NK/T cells.

3)Fig. 1e is repeating the data from Fig. 1b and adds little info to it. MPO and CD34 are said to be clinically validated blast cell markers. This probably refers to flow cytometry and staining analysis, but are there any clinically validated blast cell markers from the transcriptomic data?

Response: **Fig 1E** in the original manuscript highlighted RNA level expression of two clinical markers routinely used for AML blast identification in FACS and immunohistochemistry assays. As their performance at protein level is not validated, we have moved the feature plots to Figure S5 in the revised manuscript.

4)The labels in Fig 1c are unclear

Response: Thanks. We edited **Fig. 1C** labels in the revised manuscript.

5)Fig 1f label, the term "expression" is ambiguous and the labeling "pseudocolor" is also uncommon. The reason to highlight some genes is not mentioned.

Response: 'Expression' represents scaled normalized expression of each gene. We have mentioned this in the revised figure legend (Page 20, lines 16-17). We have replaced the term pseudo color with color. We had highlighted the genes that were discussed in the results section of the manuscript.

6)Fig. 2a the x-axis labels are unreadable. The highlighted genes (n=6) are not explained in the labels

Response: The X-axis represents the patient IDs that are not fully readable due to >150 patients used for generating the heatmap. In the revised manuscript in the **Fig 2A** legend, we added axis information (Page 20, line 23). The highlighted genes were mentioned in the main text. We have removed the highlights in the revised figure.

7)Fig. 2e the numbers are not explained.

Response: The original **Fig. 2E** is the split UMAP for the Dx, EOI, and Rel samples of patients 5 and 6 (**Fig. 2F** in the revised manuscript). UMAP is only used for visualization, and the numbers on the axis are not used for inference or clustering.

8)"including CLEC11A (7), PRAME (8), and AZU1 (9) have been previously associated with AML" is a vague statement, please amend.

Response: We have amended to add more context (page 5, lines 30-33).

9)Suppl Fig 5a-b, it is unclear how the module score was defined.

In the Revised manuscript, Figure S5 is now Figure S6. We have explained the calculation of the module score in the Figure S6 legend.

10) Fig 5a-b the clusters are not named, and the markers are buried in the Supplementary Figure 8b, making following the story extremely cumbersome for the reader and leaving the results on an abstract level.

Response: In the revised manuscript, we have combined original figures 4 and 5. We have listed the cluster annotation in the revised figure (**Fig. 4C**).

Reviewer #2 (Remarks to the Author): Expert in pediatric AML genomics

This is an important study that will have impact on the AML field, highlighting features of pediatric AML not previously known. The paper is a computational study, deep diving into the molecular features of AML from single cell transcriptomics data. The power lies with the longitudinal samples from diagnosis, end of induction treatment and relapse patients. It is continuously stated (as a negative point in most cases), that AML is heterogenous. This study uses samples from patients from all different cytogenetic backgrounds and in fact treatment schedules. The single cell approach was able to pull out important signatures, some with prognostic power, from these samples. This I believe is important to highlight and a step forward in combatting those negative comments that “AML is heterogenous”. With this in mind, I think the authors should make a much better effort is being more transparent in the clinical features of the samples used – highlighting and interpreting the data in the context of the cytogenetics of the samples and the treatment they received when appropriate and the results are indicating it– perhaps doing additionally computational analysis to take these questions on board.

Response: Besides identifying common transcriptome signatures in relapse and CCR-associated Dx blasts, we followed your suggestion to perform additional analysis to look at subtypes' specific differences. We have included figures showing DEGs in Dx blasts of different AML subtypes (MLL+ (patients 1, 4, 5), RUNX+ (patients 11, 12, 14, 20), 7q (patient 6, 16), NPM+ (patient 10) triple mutation (patient 13), CEBPA+ (patient 8), INV(16) (patient 9), no mutations (patients 2, 3, 7), AMKL (patient 17), APL (patient 18), FLT3/ITD (patient 19), and 7q+RUNX (patient 20) used in the study (Figure S13). We also performed CNA analysis of the Dx blasts to identify chromosomal alterations associated with various subtypes (Figure S14). In this paper we did not focus on the treatment type as most of the patients received similar treatments (Table S1B).

Specific comments

1. The samples used in this study are from many different cytogenetic subgroups – how was this considered in the data analysis – for example, 2,4,5 and 6 relapsed with 4 and 5 both having MLLr and 6 having 7q. CCR EOI from 3, 5,6, 14 and 15, with 14 having t(8;21) and 15 the only patient in the study with FLT3-ITD and NPM1. What risk category were each of these patients in (the risk category should be added to table S1A). How was the difference in treatments received by the patients considered in the data analysis?

Response: We have now added the risk categories in Table S1B. Most of the patients profiled in the study were treated with similar therapy, and therefore, we have not performed therapy-focused analysis. Additionally, in this part of the analysis we are exploring blast and immune cells characteristics at the time of disease diagnosis to correlate them with patient outcomes, therefore performing therapy-focused analysis is not applicable for this part of the study.

2. It would be much clearer is a schematic/table of the samples used for each sc-RNA study for each figure were being analyzed.

Response: Thanks for this suggestion. In the revised manuscript, for each figure and legend, we have added patient keys. Additionally, in each result section, we have also mentioned patient samples used in the analysis to make it clear for the readers.

3. On page 4, it is not indicated which patient samples were used for the scRNA-seq. It is stated that 3 from relapsed patients (4 relapsed in the study so which 3 were single cell profiled) and 1 from a patient with CCR (5 had CCR in the study so which of the 5 was single cell profiled). Figure 1 C indicates samples 3D,3E, 5D, 5E, 6E,6D, 14E and 14D (4 paired Dx and EOI samples). It should be indicated on table S1B that it was these samples that were single cell profiled by scRNA-seq. 14 CCR is an t(8;21) with only 40% blasts in the BM compared to 85-91% in the relapsed samples (1 with MLLr, 1 with 7q, and 1 with AML,NOS(not otherwise specified)).

Response: All sample data presented in the manuscript has been profiled using scRNA-seq. The footnote for revised supplementary Table S1 provides the specific samples used for each analysis. In the revised figures, we have included patient icons to show the patients used as well as included patient information in the figure legends.

4. The scRNA-seq was performed on the entire bone marrow samples, the advantage of which was that the analysis could determine what other cells from the stroma and immune cells were in the leukemic milieu. The downside of this is that the leukemic stem cells signatures may have been missed. From the blast percentage in table S1, the numbers varied considerably across relapsed and CCR samples. It would be important to know and specify the samples used in the single cell analysis to compare the blast percentage and this may in fact impact on the cells present in each sample and the signatures/ transcripts present at the higher levels. This point should be considered.

Response: Our aim was to develop a blast-associated signature that could distinguish AML blasts from non-blast cells to characterize them separately. We have specified the samples used in each assay in the revised manuscript (in the main script, legends, and figures). Single cell assays reflect the cell populations in the bone marrow, and we agree with the reviewer that higher blast percentages will shrink the immune populations and this is observed in Dx vs EOI comparisons. However, individual barcoding and deep sequencing has been performed to avoid transcript drop-out and under-representation of cell types. As we are analyzing specific cell types in scRNA-seq data, the transcripts captured reflect expression levels in specific cell types irrespective of other cells captured in the assay.

5. In figure 1F, the color bar for EOI-enriched and Dx-enriched appear to be DX=blast populations as indicated in figure 1D and EOI=non blast populations – but in the text on the bottom of page 4 it describes the evaluation of 44 genes in the blast cell clusters – This is confusing. The legend states it is comparing the genes in blasts compared to other differentiated cell types which makes sense.

Response: Fig. 1F is Fig. 1E in the revised manuscript. We agree and have modified the text in the revised manuscript text to reflect the above suggested change. (Page 5, lines 5-7).

6. On page 5, it is discussed that figure 2B depicts uniform expression of 2 genes in the 20 gene signature that better distinguish blasts that those used clinically and shown in figure 1E/S4. However, in S4 the data is expression from combined Dx and EOI AML samples, whereas it is unclear if the data from 2B is from only Dx samples. The data showing separately for Dx and EOI samples, or just Dx samples for the clinically used markers Cd33 and NCAM1 would better support this statement.

Response: Fig 2B is showing integrated analysis of Dx and EOI samples (patients 3, 5, 6, 14). We have followed your suggestion and replaced Fig. 2B with split feature plots showing expression

in Dx and EOI samples. We have indicated in the figure legend and inserted the patient icons in the figure to clarify the samples used in the analysis. In the revised manuscript Figure S4 is Figure S5 and we show expression as split feature plots of Dx and EOI samples.

7. Supporting the statement that the majority of the blast genes exhibit association with overall survival in figure 2D, table S5 is provided which is not easily discernible to ascertain what “the majority” means. A figure with the OS curves, as shown in 2D would be easier to interpret.

Response: 14 out of 20 putative blast AML-associated genes show significant association with survival outcome (page 5 line 21-23). The significance of gene expression and clinical outcome is shown with asterisk (* $P < .05$, ** $P < .01$, *** $P < .001$), explained in the footnote of the table (Table S5). Due the number of genes ($n=20$), we have presented the data as a table (Table S5) instead of individual OS curves.

8. The 20-gene signature is then looked at in Dx, EOI and Rel of 2 patients – Whilst it is clear that the scRNA 20-gene blast cell profile is diminished in EOI in figure 2E, which 2 patients is this data from. It was explained that single cell sequencing was carried out on 4 patients, with only 1 other those relapsing so it is very unclear which 2 patients this data is from, what their cytogenetics are, and their blast % in the BM. Figure 2F indicates that samples 5 and 6 are used here. From deduction then, and looking carefully at table S1B, there is only 2 patients in the study where Dx, EOI and relapse samples are used. 1 of these is an MLL, the other is a 7q, the WHO classification is different and the treatment both received was different and their outcomes also different (deceased (sample 5) V alive (sample 6)). Looking at the analysis then in figure 2F, one can see that the blast and immune cell populations are very different between these 2 patients. I think this is relevant information that should be discussed/revealed better in the text. Also, for confirmation is this data generated in 2F using the 20-gene signature – as this should be clearly indicated in the text and legend.

Response: Fig. 2E (now Fig. 2F) is samples 5 and 6. We have specified this in the text and the figure. We have mentioned the differences in the immune cells and blasts as well in the revised text (page 5, line 38-44; page 6, lines 1-3). We used the 7-gene signature in the revised manuscript to identify the blast cells (page 5, lines 38-41) (Figure S6A, B).

9. The study then looked at the 14 Dx samples derived in figure 2G-H. The UMAP projection depicts CCR and relapse clusters but it is not clear how this data was generated? Understanding that 6 patients relapsed and 8 patients achieved CCR, how were the clusters derived from the Dx samples only? It is how the data is explained is the problem – the “CCR-Associated clusters”, are these clusters found only in the patients that achieved CCR? So they are clusters seen at diagnosis associated with CCR-patients/samples and clusters associated with patients that relapsed? And was it done using the 7 gene score? This needs to be more clearly explained if this is the case.

Response: The focus of the study is to characterize the association of baseline tumor and microenvironment with outcomes. We specifically looked at Dx samples for characterizing TME of Dx samples (patients 1-14). The clinical outcome is known for each sample and was then referred to when we mention CCR- or relapse-associated. In the original manuscript, we analyzed 14 samples collected at the time of disease diagnosis, eight of these samples are from patients who achieved complete clinical remission after induction therapy and six are from patients with relapse

and the terms ‘CCR-associated’ or ‘relapse-associated’ reflect these outcomes respectively. In the revised manuscript, we have given a clearer explanation of the samples used in the study. We included five additional patients with Dx and EOI samples in the revised manuscript for validation of the 7-gene signature. Table S1A, B includes details of the samples used in the study.

For the Dx study, when we state clusters are CCR-associated, we mean that majority of the cells in a particular cluster are from patients who achieve CCR. We used the 7-gene signature to identify AML blasts (Figure S11). We then subsetted out the AML blasts and non-blast cells and analyzed them separately (page 7: lines 2-4).

10. For figure 3B, are the presence of the different clusters impacted by the blast percentage in the patients rather than the clusters are not present in the patient ?– in other words, a cluster may be present but undetectable because of a dominance of another cluster found in the dominant blast? So the inter-patient variability may be due to the constraining factors of the study, mainly being the different cytogenetics? Indeed, there is a dominance of NPM1 positive samples in the CCR patients as shown in table S1A, and this is a gene pulled out unsurprisingly then from a CCR-associated cluster in figure 3C.

Response: We can detect sample specific clusters even if they are present at a fraction of the dominant sample in a cluster. We have performed chromosomal number alterations (CNA) using the InferCNV tool in the revised manuscript and show genetic subtype specific DEGs as well as shared modulation of gene expression.

11. In figure 5B, there appears to be specific T cell clusters enriched in specific samples, again possible linked to the cytogenetics of the samples. This should be considered.

Response: Of the 11 clusters formed upon sub-clustering of the T cells, we do see patient specific T cell clusters (Fig. 4D in the revised manuscript; page 9: lines 18-19). This might be due to transcriptome differences in the T cells response induced by the nature of the blasts. But overall, there is a pattern of the type of T cells present at Dx in relapse- and CCR-associated samples which we discussed (page 9: lines 20-43; page 10: lines 1-5).

12. The data in figure 5E is not explained very well – how was this data generated?

Response: As this figure (original manuscript Fig. 5E) was not significant to this paper, we have removed it.

13. In figure 5F, why were clusters 1 and 2 and 11 left out of the analysis of relapse-associated clusters, and why was 4 and 9 left out of the CCR-associated cluster gene analysis? Is this because there was a dominant cluster in a specific sample - and what were the clinical features of those samples – this should be transparent in the text. An the without a valid reason, these clusters should not be excluded.

Response: This is Fig. 4F in the revised manuscript. We selected relapse- (clusters 3,5,6) and CCR- (clusters 7,8,10) associated clusters that were not sample specific for DEGs analysis. We have mentioned this in the revised manuscript main text (page 9: lines 35-36), and also the figure legend (page 21: lines 27-29).

14. When the term “non-AML cells” is used, how confident can it be that these cells are non-AML. What is your definition of non-AML? They are in fact cells part of the AML TME.

Response: Non-AML cells refer to non-blast cells that are well-differentiated cells or normal immature cells. In the revised manuscript we have replaced the term “non-AML” with “non-blast”.

15. In the results discussing the genes enriched in cluster 7, it is clear that in S9B the analysis is of the 20 gene signature but what list of genes was interrogated in S9C – was this a manual list of immature markers?

Response: This figure is Figure S17C in the revised manuscript. The genes in Figure S17C are a set of canonical marker genes used by our group to annotate cells. We have mentioned this in the main text (page 10, lines 12-15) as well as amended the legend for Figure S17C.

16. Figure 6C it is stated that M2 markers MRC1, ARG1 and MMP9 (M2 markers) are mostly overexpressed in the relapse-associated samples – From the figure this is very difficult to see. The percent expression is low and the (dots are extremely small to make out with the naked eye). Importantly, CD163 appears to be overexpressed in the CCR-associated clusters, a marker linked with poor AML survival and M2 macrophages. What do the authors think of this – is a potential explanation that in fact the cytogenetics is important, these are pediatric compared to the data in the literature being from adult? This may in fact be highlighting some very relevant and unique immune properties for pediatric AML.

Response: Thanks. We agree that M2 marker genes were not very visible in the dot plot. Upon re-analysis of the data, we observed that levels of M2 macrophage markers as well as differences between CCR- and relapse-associated samples are not significant (CD163 was weakly expressing compared to M1 markers with expression seen only in <25% of CCR-associated samples). We instead present focused ssGSEA of M1 macrophages genes (*S100A8*, *S100A9*, *S100A12*, *TYROBP*, *VCAN*, *CD68*, *MNDA*, *CYBB*, *STAT1*) to demonstrate increased expression in Dx CCR-associated samples compared to relapse-associated samples (**Fig. 5B**, page 10: lines 15-18). We agree that pediatric AML is different from adult AML and studies like this study is important to identify the differences.

17. It should indicate in the text regarding sc-RNA seq for figure 7 which patients samples were used – it states 3 relapse and 2 CCR but which patients were these? Was the cluster analysis done using these samples all grouped together (figure S10, 7B)– and so the interpretation is that the clusters 6 and 9, and residual blasts remain regardless of relapse or CCR? This should be clear if so. And if that is the case, are these blast clusters important and figure S10 necessary? The survival curves in figure S11 are more important in the main figure rather than figure 7C I think.

Response: In the revised manuscript this is **Fig. 6**. We mentioned which samples are used in the main text of the revised manuscript (page 10: lines 37-39) as well as in the figure legend for **Fig. 6**.

In the **Fig. 7** of the revised manuscript, we are focusing on the non-blast cells, so we do not discuss clusters 6 and 9. Figure S10 is Figure S18 in the revised manuscript. This figure is important, as it shows the blast cells in the Dx, EOI integrated object. We have removed Fig. 7C (module score). We agree that the survival curves are important but due to the paucity of space we have kept it as a supplementary figure (Figure S19).

18. A comment or additional discussion in the text highlighting how this scRNA-seq analysis places pediatric AML distinct from adult AML is warranted and would highlight greatly the impact of this study in the AML field. Referring to functional studies such as Chaudhury et al, Nature Comms,(9), 2018 and Lopez et al, Cancer Discovery 2019 highlight such importance in line with this.

Response: We agree. In pediatric leukemic blasts the mutation rates are much lower compared to adult leukemic blasts. Also, the BM niche and immune microenvironment will be different in pediatric AML patients compared to adult AML patients. We have mentioned this and referenced article from Chaudhury et al., 2018 in the 'discussion' section of the manuscript (page 12, lines 219).

Minor comments

1. **The age of the patients is missing from table S1 which is important to add.**

Response: We have added the age in the revised table S1A.

2. **It would be easier to ingest the data if table S1A and table S1B were merged.**

Response: We merged the tables in the revised manuscript but added a new table (Table S1B) to give addition information (as putting everything in one table would have made it difficult to read).

3. **The colours of the samples in figure 1C are very similar and the font of the population names is blurry/shadow. Can more divisive colors be used for C and D.**

Response: The Figure 1C has been replaced.

4. **Figure 2A there is no ref/green color legend and the labels along the x axis are illegible.**

Response: Thanks. We have inserted figure color bar on side. X axis is patient number, we have included this information in the figure legend (page 20: line 23).

5. **Figure S7E is illegible – the string network is too squashed and gene names can't be read.**

Response: Corrected to make legible (Figure S12 in revised manuscript).

6. **The legend in figure 6A has 4 colors but only 3 labels – the colour matched to labels need to be indicated. State in the legend what "Other" is**

Response: This is Fig. 5A in the revised manuscript. We have shown only the monocytes/macrophages clusters and relabeled the clusters clearly.

7. **The quality of figure 6B is not good. Font is over stretched. Response:**

This is Fig. 5C in the revised manuscript. We have inserted a new figure.

8. **In the text for a lot of the figures, a number of genes are mentioned. It would help that for each figure where a gene is highlighted in the text, that a box or color highlighting that gene along the axis labels is placed.**

Response: We have highlighted the genes mentioned in the text and also added a note to the figure legend.

9. **The legend for figure 7B and 7C need inverting.**

Response: Thanks. This is **Fig. 6 B, C** in the revised manuscript. We have fixed it.

Reviewer #3 (Remarks to the Author): Expert in single-cell RNA-seq and leukemia genomics In this manuscript, matched samples from pediatric AML cases from multiple time points (including diagnosis, end-of-induction, and either remission or relapse) were analyzed using single-cell RNA-sequencing. The authors present an AML blast gene signature and describe differences in the microenvironment at time of diagnosis between patients that ultimately relapse or reach remission.

The general approach presented here is substantively flawed. A key step in analyzing tumor samples by scRNA-seq is identifying the tumor cells. For solid tumors, this is generally done by identifying copy number alterations (CNAs) in the tumor cells. In AML, it is generally done by identifying SNVs and/or CNAs in individual cells. This was not done in this paper. Mutation/CNA identification is a critical step in any analysis, particularly in AML, because tumor cells often partially differentiate such that they express markers of more differentiated cell types. This manuscript simply assumed that AML cells are the cells that express typical markers of stemness/HSPCs, such as CD34, or a mixture of markers (which might arise from doublets). This is an invalid assumption; many tumor cells have been missed, and it is the basis of most of the analyses in this manuscript.

Response: Mutation/CNV analysis is difficult because of the sparsity of data in scRNA-seq analysis. Also, as it is difficult to get pediatric HSC and healthy bone marrow samples as controls, we wanted to try a different approach. This involved leveraging the power of UMAP clustering of matched Dx and EOI samples. We took matched Dx and EOI samples to identify cell clusters that are enriched in Dx samples and reduced in post-therapy EOI samples. Cell clusters showing significant reduction post-therapy were presumed to be blast cells that are reduced by induction treatment. We also looked at clinical markers like *MPO* and *CD34* when putatively classifying the clusters significantly reduced in post-therapy samples as blasts. Following the development of the signature for distinguishing AML blasts from non-blast cells, it was validated using an external scRNA-seq AML dataset as well as internal additional AML scRNA-seq datasets for identifying blast and non-blast cells. The signature was also validated by checking expression in normal healthy bone marrow (young adult) and normal hematopoietic stem cells.

In the revised manuscript we have performed additional CNA analysis using inferCNV tool for detecting additions or deletions in the AML blasts. The algorithm determines dysregulations of genes across chromosomal positions in the tumor cells and normal cells to identify regions in blast cells that are over- or less- abundant as compared to normal cells. We performed initial CNV analysis with three AML samples compared to normal HSC and healthy BM (Figure S10) and a second CNA analysis using the initial 14 Dx samples and normal HSC and healthy BM samples (Figure S14).

Aside from the above concern, there are problems with circularity in the definition of the AML blast signature. The putative blasts identified by the authors may well be leukemic blasts, but the analyses intended to identify a blast signature simply identify genes that are correlated with the markers used to identify blasts in the first place. A better experiment would be to enrich normal bone marrow samples for progenitor cells, such as by sorting for CD34+ cells, then to compare (by scRNAseq) leukemic blasts to the normal HSPCs. (Again, this would require genetic confirmation that the blasts are indeed leukemic).

Response: We validated our AML blast signature using additional datasets to address the issue of circularity. In the revised manuscript, additional AML samples were processed and analyzed, and a publicly available AML dataset was also used for validating the 7-gene signature. Support vector machine (SVM) is a powerful supervised machine learning technique for identifying multigene biomarker panels from complex datasets. We first used SVM to identify the 7-gene signature that could differentiate blasts from non-blasts with an AUC of 0.968 (**Fig. 2E**). For validating the 7-gene signature in the new AML dataset, the 7-gene signature expression was extracted from the blast cells and normalized to conduct an SVM classification of blast/non-blast cells based on cross-validation. Expression matrix of 7-gene signature in the Dx, EOI cells were classified using MetaboAnalyst for SVM classification. Similarly, we also validated the 7-gene signature in an external AML dataset by training a multi-variate SVM model that uses all seven genes to generate an ROC curve of the model based on cross-validation performance. We obtained AUC values of 0.81 and 0.78 for the external and internal AML samples respectively (**Fig. 2H, I**).

In the revised manuscript we demonstrated that the 7-gene signature shows minimum expression in external healthy bone marrow samples and normal hematopoietic stem cells datasets (**Figure S7**). Also, using InferCNV tool, we demonstrate chromosomal alterations in the identified blasts compared to normal HSCs and healthy bone marrow cells (**Figure S10, S14**).

The observations about macrophage differences in the pre-relapse vs pre-remission samples are affected by the same problem above. However, the T cell observations are interesting and potentially correct (because most T-cells in an AML sample are indeed real T-cells).

Response: We compared AML samples to healthy BM and normal HSC. The monocyte and macrophage clusters from AML samples cluster with the healthy BM monocytes and macrophages (**Figure S7A, B**) indicating transcriptome similarity. We also checked for leukemic myeloid genes in monocyte/macrophage clusters (**Figure S17B**) to demonstrate their non-leukemic nature.

This project could be reworked by (1) performing exome sequencing on every sample, (2) identifying mutant cells in the scRNA-seq data to confidently and unambiguously identify tumor cell clusters, and (3) analyzing normal bone marrow samples by scRNAseq.

Response: The purpose of this project was to identify transcriptome differences in the tumor microenvironment of Dx and EOI AML samples. Exome sequencing is beyond the scope of the project. In the revised manuscript we have validated the developed 7-gene AML blast signature by checking gene expression in young adult healthy BM and adult normal HSC scRNA-Seq datasets. We also validated the signature using additional AML scRNA-seq datasets (both external and internal).

REVIEWER COMMENTS

Reviewer #1 (Remarks to the Author):

Authors have made revisions based on the comments by the reviewers. New data sets have been included which has improved the manuscript, but there are still some problems related to data analysis which heavily impact the interpretation of the whole manuscript. Major concerns are listed below.

1) Gene signature related to AML blasts. It is still a bit difficult to understand why authors made their gene signature comparing DX and EOI samples and not AML blasts and healthy bone marrow stem cells? EOI samples include a wide variety of different cell types. Why not new signature was made with additional datasets including healthy HSCs? In the revised manuscript authors do validate their signature in the normal BM cells, but this is not convincing (please see below). This figure should be in the main manuscript and not in the supplement as it is clearly relevant for the findings. The presentation of this data needs further modifications. The way the violin plots are separated do not follow the cluster annotation authors have shown in SFig 7A. The current groups are large, and the cell numbers between different groups differ quite remarkably. The way the authors are currently showing the data, it is not possible to understand how many cells are positive for given gene - the correct way to do this would be to show also individual cells. Probably there are lot of "outlier" cells that are not seen in the violins. Also, the P-values are not shown or are cut. The figure needs to be reformatted to:

- a. show the same groups as in SFig7A. It seems that HSCs have some of these "blast" clusters. What is the expression of these genes in those HSC cells?
- b. include all the cells as e.g., jitter
- c. show the P-values between the different groups

2) Train/test data leakage issues. Although authors say that in the revised manuscript, there are no data leakage issues, this is questionable. Authors still use their 7-gene signature to analyze the TARGET cohort (which they have used in the development of 7 gene signature), which will inevitably end up as being statistically significant. Additional validation with external data sets does not overcome this issue.

3) The use of SVM tool to select the final 7-gene signature

- a. It should be explained in the methods how the SVM was applied to get from 20 genes to 7 genes. Were different subsets from 20 genes tried to get the 7 genes, or was some metric assessed from an initial 20-gene classifier?

4) Annotations of blast and non-blast cells. Authors say in the response letter and in the manuscript that “The cells were labeled as blast or non-blast in these datasets based on expression of canonical markers, comparison with healthy BM, comparison with EOI AML samples, and patient specificity in clusters.” However, no UMAPs and cluster annotations are provided. This “true labeling” should be shown in the manuscript. In addition, authors should comment are the happy with their classifier and how well this clustering is performing with different types of AML?

5) Clustering and naming of the clusters. In the CCR/relapse associated data, authors do not use orthogonal omic data or validation cohorts for their immune cell clusters. For example the high number of T cell clusters is not in-line with the number of clusters from the previous publications, and over clustering maybe an issue.

6) One sentence summary should be modified to make it more clear for the readers

Reviewer #2 (Remarks to the Author):

I thank the authors for taking on board the suggestions made and substantially addressing the critiques. I believe the changes to the figures, with the patients and samples indicated schematically (patient keys), is a wonderfully clear and clever way to show clarity for each figure. This is highly appreciated in the revised manuscript, and helps to alleviate many concerns and confusions brought up on the first draft. I believe the authors analysis and inclusion of results based on the genetics of the samples in figure S13 is an important addition to the paper. The inclusion of the CNV analysis is a smart approach to identify blasts from non blasts and I think the authors have addressed the critiques with this analysis really satisfyingly well. Tis is a really important addition to the revised paper. The validation data provided on independent samples for the 7 gene signature is powerful and supports the authors original conclusion and hypothesis. I am satisfied that the authors do not include therapy-focused analysis according to their response. Some very minor spelling corrections required on proof-reading, but this reviewer is more than satisfied with this revised manuscript, and congratulate the authors on great work.

Reviewer #3 (Remarks to the Author):

This version of the manuscript is significantly improved, and addresses many of my previous concerns. I have some additional comments (mostly minor):

- p. 4 Lines 36-37: It is unclear exactly what comparison(s) are being done. Are DEGs found on a patient-specific basis and then combined to select the intersection? union? More clarity would help.
- Fig. 1F appears not to be referenced in the text (?)
- The description on p. 6 lines 15-16 could be more clear
- The CNV analysis performed here compares putative AML blasts to normal cells from different patients. It is more standard to compare putative tumor cells to putative normal cells from the same patient to account for any germline differences.
- A more standard word in place of “interactant” (interactor? Interacting protein?) might be preferable
- The T-cell and macrophage analyses yielded interesting results, but the power to detect signatures across multiple patients was low, which is concerning since this is central to the study’s conclusions. For instance, on p. 9 (Fig. 4D), clusters 3 and 6 are mostly from the same two patients. It is difficult to generalize these signatures across patients. Likewise, on p. 10 (Fig. S17), clusters 1 and 2 are mostly from one patient each, and Cluster 6 is from three patients. Again, this provides little power to generalize these signatures across multiple patients. This analysis and the analogous T-cell analogous in Fig. 4 would be strengthened by showing that these signatures exist in more patients. If the appropriate bulk data sets exist, this could be done using deconvolution (e.g. using Cibersortx) of bulk data to generalize across larger numbers of patients.
- p. 10 lines 11-12: It should be demonstrated that these putative non-leukemic cells do not contain CNAs, rather than just relying on marker genes for the assertion that they are nonmalignant.
- p. 11 lines 13-15: Writing could be more clear.

Below, we have enclosed pointwise responses to address the concerns and suggestions. The major revised parts have been highlighted in **red** color for your convenience of re-reviewing.

Reviewer #1 (Remarks to the Author):

The authors have made revisions based on the comments by the reviewers. New data sets have been included which has improved the manuscript, but there are still some problems related to data analysis which heavily impact the interpretation of the whole manuscript. Major concerns are listed below.

Response: We would like to thank the reviewer for acknowledging that the revised manuscript is significantly improved with the addition of new data. We are also thankful to the reviewer for insightful comments. We have performed additional analysis, revised figures, and text to further improve the manuscript.

1) Gene signature related to AML blasts. It is still a bit difficult to understand why authors made their gene signature comparing DX and EOI samples and not AML blasts and healthy bone marrow stem cells? EOI samples include a wide variety of different cell types. Why not new signature was made with additional datasets including healthy HSCs? In the revised manuscript authors do validate their signature in the normal BM cells, but this is not convincing (please see below). This figure should be in the main manuscript and not in the supplement as it is clearly relevant for the findings. The presentation of this data needs further modifications.

Response 1.1a: Thanks for the comment about the analysis approach. As it is difficult to obtain age-matched pediatric healthy BM samples, we attempted to develop an AML-blast signature by identifying the genes differentially expressed in Dx-enriched clusters (identified putatively as AML-blasts) compared to EOI-enriched clusters (immune cell clusters). The genes were further filtered using pediatric AML dataset 1 from the TARGET initiative (TARGET AML-1) to identify genes overexpressed in high-blast (>60%) vs low-blast (<30%) samples. We further integrated the data with young adult healthy BM cells and normal adult HSCs (**Fig. 2H**) and evaluated the module score expression profile of the 7-gene signature. The 7-gene signature depicted elevated expression in blast cells clusters as compared to normal BM and HSC clusters (**Fig. 2I**). As per the reviewer's suggestions, we have included this figure showing the expression of the 7-gene signature as module score in AML (four patients: 3,5,6,14), normal HSCs, and healthy BM datasets (**Fig. 2I**) (page 6, lines 9-18). We also performed validation of this signature's association with AML-blasts in additionally generated pediatric AML single-cell RNA sequencing (scRNA-seq) data (**Figure S8**, samples 16-20), publicly available pediatric AML scRNA-seq data (**Figure S9**, Bailur et al., 2020), and additional bulk RNA-seq TARGET AML samples that were not used in the development of the 7-gene signature (**Figure S10**, TARGET

AML-2). Through rigorous validation across multiple datasets and observing reduced expression levels in healthy BM and normal HSCs datasets, we have compelling evidence to suggest an association between the 7-gene signature and pediatric AML-blast cells (page 6, lines 19-42; page 7, lines 1-7).

1.1b: The way the violin plots are separated do not follow the cluster annotation authors have shown in SFig 7A. The current groups are large, and the cell numbers between different groups differ quite remarkably. The way the authors are currently showing the data, it is not possible to understand how many cells are positive for given gene - the correct way to do this would be to show also individual cells. Probably there are lot of "outlier" cells that are not seen in the violins. Also, the P-values are not shown or are cut. The figure needs to be reformatted to: a. show the same groups as in SFig7A. It seems that HSCs have some of these "blast" clusters. What is the expression of these genes in those HSC cells? b. include all the cells as e.g., jitter. c. show the P-values between the different groups.

Response 1.1b: It will be cumbersome and crowded to show different cell types represented in Figure S7a, b, so we have kept the original groups format for Figure S7c. We have shown the expression of the 7-gene signature in three datasets (AML, healthy BM, normal HSCs) in Fig. 2I, and updated Figure S7B, which now depicts the proportion of different cell types in each dataset. The HSCs have a weak expression of the 7-gene signature (Fig. 2I), but it is significantly lower for most of the genes when compared to AML blasts (Figure S7C). We have modified Figure S7C to show the distribution of cells expressing specific genes from the 7-gene signature along with the *P*-value calculated by performing a statistical comparison using Wilcoxon signed-rank test between AML-blasts (*AZU*⁺, *CD38*⁺, *MPO*⁺) and normal HSCs.

2) Train/test data leakage issues. Although the authors say that in the revised manuscript, there are no data leakage issues, this is questionable. Authors still use their 7-gene signature to analyze the TARGET cohort (which they have used in the development of 7 gene signature), which will inevitably end up as being statistically significant. Additional validation with external data sets does not overcome this issue.

Response 1.2: During the development of the 7-gene signature the Dx, EOI TARGET dataset (TARGET-AML-1) was used to evaluate the expression of differentially expressed genes identified in the blasts of Dx scRNA-seq data. We agree that the TARGET-AML-1 dataset was used during the development of the 7-gene signature as well as determining the survival association of the genes. To address potential train/test data leakage issues in the revised manuscript, we implemented a rigorous validation process for our 7-gene signature in AML blasts enrichment. We performed independent validation using multiple distinct sources of data: additional scRNA-seq pediatric AML data generated

specifically for this study (Figure S8, samples 16-20), and publicly available pediatric AML scRNA-seq data from Bailur et al., 2020 (Figure S9). By incorporating these independent datasets, we ensured that our findings were not biased or influenced by the original training data. This approach allowed us to evaluate the generalizability and robustness of our 7-gene signature beyond the initial dataset. These datasets were never used in the development of the 7-gene signature. Additionally, the enrichment of 7-gene signature also depicted a correlation with clinically calculated blast percentages in TARGET AML samples (TARGET AML-2, n=1,661) that were not included in developing the 7-gene signature (Figure S10) (page 6, lines 37-42; page 7, lines 1-8). The TARGET AML-2 data was split into four groups, namely, normal BM (n=324), <30% blasts (n=169), ≥30% and <60% blasts (n=342), and ≥60% blasts (n=826). Most of the individual genes from the 7-gene signature exhibited higher expression in AML samples as compared to healthy BM samples. The genes also depicted a pattern of increased expression with increasing blast percentages supporting the association with pediatric AML-blast cells. These independent validations further strengthen our finding for the association of the 7-gene signature with pediatric AML-blast cells. We have edited the workflow figure to include additional validation of the AML-blast signature (Figure S11 in the revised manuscript).

3) The use of the SVM tool to select the final 7-gene signature. a. It should be explained in the methods how the SVM was applied to get from 20 genes to 7 genes. Were different subsets from 20 genes tried to get the 7 genes, or was some metric assessed from an initial 20-gene classifier?

Response 1.3: An SVM classifier was utilized to train a model using the expression levels of 20 genes that are found to be over-expressed by blast cells. The goal was to distinguish between blast and non-blast cells based on the expression patterns of these twenty genes. This training was performed using single-cell data obtained from paired Dx and EOI samples of patients 3, 5, 6, and 14, along with blast/non-blast labels. To assess the performance of the classifier during training, we employed out-of-bag (OOB) cross-validation, using the MetaboAnalyst package. This approach measures the classifier's accuracy by evaluating its performance on unseen data points. The SVM classifier identifies the most "important" features by considering their relative contribution to the classification task, as indicated by the error rates. The features incorporated into the best model, which are considered the most "important," were determined by examining the frequency with which they were included in the best-performing model. Consequently, the 20 genes were ranked based on their inclusion in the top models. Through this analysis, we identified the most significant genes that resulted in the 7-gene signature. Although additional predictors using 5 to 15 genes were also developed, they were

ultimately excluded from validation since the 7-gene signature demonstrated comparable performance during cross-validation.

4) Annotations of blast and non-blast cells. Authors say in the response letter and in the manuscript that “The cells were labeled as blast or non-blast in these datasets based on expression of canonical markers, comparison with healthy BM, comparison with EOI AML samples, and patient specificity in clusters.” However, no UMAPs and cluster annotations are provided. This “true labeling” should be shown in the manuscript. In addition, authors should comment are happy with their classifier and how well this clustering is performing with different types of AML?

Response 1.4: We have included split UMAPs (Dx and EOI AML, healthy BM, normal HSC), dot plots for canonical cell annotation markers, and bar plots showing sample and cluster proportions to better explain blast cell identification in the validation AML scRNA-seq data generated in our lab and in the external AML scRNA-seq dataset (**Figures S8, S9**). We also have edited the main text to reflect these additions (page 6, lines 19-36). The 7-gene signature achieved an AUC of 0.78 in distinguishing AML-blasts and non-blasts on validation AML scRNA-seq samples (RUNX+, 7/add(7q)/del(7q), APL and AMKL) processed in our lab. On an additional external validation set of pediatric AML samples (Bailur et al., 2020: inv(16)/t(16;16), 11q23 (KMT2A), t(8;21)(q22;q22), Monosomy 7, FLT3-ITD, mutated NPM1) the 7-gene signature achieved an AUC of 0.81 in distinguishing blast cells from non-blast cells. Given the heterogeneity in AML subtypes tested, we are quite satisfied with the performance of the 7-gene classifier in distinguishing AML-blast clusters from non-blast clusters (**Fig. 2J, K** in the revised manuscript).

5) Clustering and naming of the clusters. In the CCR/relapse-associated data, authors do not use orthogonal omic data or validation cohorts for their immune cell clusters. For example, the high number of T cell clusters is not in-line with the number of clusters from the previous publications, and clustering may be an issue.

Response 1.5: The patient-specific T cell subclusters, identified through the subsetting of the T cell cluster, can be attributed to the immune response triggered by individual blasts (page 10, lines 9-11). The revised manuscript demonstrates that these subclusters predominantly exhibit distinct overexpression of top differentially expressed genes (**Figure S20**). To evaluate the quality of the T cell subcluster clustering, we calculated the silhouette score, which achieved a mean score of 0.1 based on 10 principal components, indicating the stability of the clusters. Heterogeneity in marker expression among the clusters was observed when analyzing the expression of canonical T-cell markers in the subclusters (**Figure S19B**), which is expected given the presence of multiple T-cell subtypes. Furthermore, in the analysis of T cell subsets, we compared the profiles of multiple CCR and relapse-

enriched clusters instead of individual clusters, thereby mitigating the impact of clustering on the analysis. For instance, in **Fig. 4E**, we calculated the ssGSEA scores for naïve and exhausted T cell gene sets in CCR- and relapse-associated T cells. Additionally, in **Fig. 4F**, we compared the relapse-enriched subclusters (**3, 5, and 6**) to the remission subclusters (**7, 8, and 10**) to account for sample-specific differences and identify dysregulations associated with T cells in remission and relapse samples. It is important to note that the findings from this study are preliminary and require further validation in a larger BM microenvironment study focusing on profiling of normal adaptive and innate immune cells.

To associate immune signatures with outcome, we incorporated two independent omic data cohorts: a bulk RNA-seq dataset from pediatric AML samples (Fornerod et al., 2021) and bulk RNA-seq dataset 2 from TARGET AML pediatric AML samples (TARGET AML-2), excluding samples used in our initial analysis depicted in **Fig. 2** (page 12, lines 35-42; page 13, lines 1-25; page 21, lines 27-42; page 22, lines 1-7). In the Fornerod et al. AML dataset, comprising $n=132$ samples, risk categories were determined based on the 6-gene leukemic stem cell (LSC6) score developed by Elsayed et al., 2020. For the TARGET AML-2 cohort ($n=1,398$), we employed the same LSC6 risk score categorization method as Fornerod et al. (page 21, lines 37-39). The enrichment of **M1 Macrophages** (*S100A8*, *S100A9*, *S100A12*, *TYROBP*, *VCAN*, *CD68*, *MNDA*, *CYBB*, *STAT1*), **naïve T cells** (*CCR7*, *LEF1*, *TCF7*, *SELL*), **effector T cells** (*CCL5*, *NKG7*, *GNLY*, *GZMA*, *GZMK*, *NFACT1*), **exhausted T cells** (*HAVCR2*, *LAG3*, *PDCD1*, *NFATC1*, *TIGIT*, *TOX*), and **regulatory T cells** (*CCL5*, *KLRB1*, *KLRD1*, *GZMH*, *CD69*, *CD44*) in the bulk RNA-seq datasets were assessed by calculating single-sample gene set enrichment analysis (ssGSEA) scores for each immune cell type. In the Fornerod et al. AML dataset, M1 Macrophages exhibited a significant ($P=1.2e-05$) negative correlation with the LSC6 score, indicating lower enrichment in high-risk samples (**Figure S27A**). Additionally, exhausted T cells enrichment scores displayed a significant ($P=1.9e-05$) positive correlation with the LSC6 score (**Figure S27A**). We also examined the association between categorized scores (i.e., low-, medium-, and high-risk) and the enrichment of these cell types. Low-risk AML samples showed significantly higher enrichment of M1 Macrophages compared to medium-risk samples (Wilcoxon test, $P<.0001$). Exhausted T cells were significantly more enriched in medium-risk samples compared to low-risk samples (Wilcoxon test, $P<.05$) (**Figure S27B**). Due to the low number of high-risk samples ($n=3$) relative to the other two categories, it was excluded from the Fornerod et al., AML dataset analysis. A similar analysis conducted on the TARGET AML-2 cohort also revealed a negative correlation between the LSC6 score and the enrichment of M1 Macrophages ($P=7.3e-11$) as well as naïve T cells ($P=9.8e-11$). Furthermore, exhausted T cells exhibited a significant ($P=5.7e-09$) positive correlation with the LSC6 risk score (**Figure S28A**). Moreover, when comparing ssGSEA scores for M1 Macrophages, naïve T cells, and exhausted T cells among the low-, medium-, and high-risk categories in the TARGET-AML-2 cohort, we observed that low-risk AML samples were significantly more

enriched in M1 Macrophages ($P < .001$) and naïve T cells ($P < .05$) compared to high-risk samples. Conversely, high-risk AML samples exhibited significantly higher enrichment of exhausted T cells ($P < .001$) compared to low-risk AML samples (**Figure S28B**). These findings support our single-cell results, which showed the enrichment of M1-Macrophages and exhausted T cells in the CCR and relapse samples, respectively.

6) One sentence summary should be modified to make it more clear for the readers

Response 1.6: We have modified the sentence to: **Single-cell transcriptome profiling maps the blast cells and bone marrow microenvironment of pediatric AML patients with continuous clinical remission and relapse** (title page).

Reviewer #2 (Remarks to the Author):

I thank the authors for taking on board the suggestions made and substantially addressing the critiques. I believe the changes to the figures, with the patients and samples indicated schematically (patient keys), is a wonderfully clear and clever way to show clarity for each figure. This is highly appreciated in the revised manuscript, and helps to alleviate many concerns and confusions brought up on the first draft. I believe the authors analysis and inclusion of results based on the genetics of the samples in figure S13 is an important addition to the paper. The inclusion of the CNV analysis is a smart approach to identify blasts from non blasts and I think the authors have addressed the critiques with this analysis really satisfyingly well. This is a really important addition to the revised paper. The validation data provided on independent samples for the 7 gene signature is powerful and supports the authors original conclusion and hypothesis. I am satisfied that the authors do not include therapy-focused analysis according to their response. Some very minor spelling corrections required on proofreading, but this reviewer is more than satisfied with this revised manuscript and congratulate the authors on great work.

Response: Thank you for appreciating our efforts in performing independent validations and revising the manuscript. As suggested, we have carefully reviewed the manuscript and corrected any spelling mistakes.

Reviewer #3 (Remarks to the Author):

This version of the manuscript is significantly improved and addresses many of my previous concerns. I have some additional comments (mostly minor):

- p. 4 Lines 36-37: It is unclear exactly what comparison(s) are being done. Are DEGs found on a patient-specific basis and then combined to select the intersection? union? More clarity would help.

Response: The DEGs were identified by comparing Dx-enriched clusters (identified putatively as AML-blasts) compared to EOI-enriched clusters (immune cell clusters) from all patients that yielded 232 significantly differentially expressed genes that are not patient-specific. To clarify the analysis, we have updated the text in the revised manuscript specifying groups used for DEGs (page 4, lines 3641).

- **Fig. 1F appears not to be referenced in the text (?)**

Response: Thanks, **Figure 1** has five panels from A to E with no 1F. We have carefully reviewed the citations of the figures in the manuscript to ensure their accuracy.

- **The description on p. 6 lines 15-16 could be more clear**

Response: To clarify, we have edited the text and figures associated with this part of the manuscript. We have updated the supplementary figure (**Figure S8**) and added an additional supplementary figure (**Figure S9**). We have also modified the main text associated with the figure (**Fig. 2 H, I**) in the revised manuscript (page 6, lines 19-33).

- **The CNV analysis performed here compares putative AML blasts to normal cells from different patients. It is more standard to compare putative tumor cells to putative normal cells from the same patient to account for any germline differences.**

Response: Thank you for the feedback. We appreciate your suggestion. As per your recommendation, we have conducted further analysis on the copy number variations (CNVs) of AML blasts for each patient. In this analysis, we utilized matched non-blast cells from the same patient as a reference control. The results of this additional analysis have been incorporated into the revised manuscript as **Figure S17**, located on page 9, lines 26-27. We have compared the outcomes of this analysis with the CNV analysis that employed healthy bone marrow (BM) and normal hematopoietic stem cells (HSCs) as reference controls. The comparison of these two analyses revealed similar patterns of copy number alterations, as depicted in **Figure S16**.

- **A more standard word in place of “interactant” (interactor? Interacting protein?) might be preferable**

Response: We have edited the manuscript and replaced the word “interactant” in the revised manuscript with “interacting protein” or “interactor” (page 8; line 17).

- **The T-cell and macrophage analyses yielded interesting results, but the power to detect signatures across multiple patients was low, which is concerning since this is central to the study’s conclusions. For instance, on p. 9 (Fig. 4D), clusters 3 and 6 are mostly from the same two patients. It is difficult to generalize these signatures across patients. Likewise, on p. 10 (Fig. S17), clusters 1 and 2 are mostly from one patient each, and Cluster 6 is from three**

patients. Again, this provides little power to generalize these signatures across multiple patients. This analysis and the analogous T-cell analogous in Fig. 4 would be strengthened by showing that these signatures exist in more patients. If the appropriate bulk data sets exist, this could be done using deconvolution (e.g. using Cibersortx) of bulk data to generalize across larger numbers of patients.

Response: To validate the association of different immune cell enrichment from scRNA-seq analysis with clinical outcomes, we performed additional analysis on two bulk RNA-seq pediatric AML datasets (i.e., Fornerod et al., AML dataset and TARGET AML-2 dataset) (page 12, lines 34-42; page 13, lines 1-25; page 21, lines 4-26).

The Fornerod et al., AML dataset (n=132) was divided into three groups, namely high-risk, medium-risk, and low-risk groups based on 6-gene leukemic stem cell (LSC6) score. The LSC6 score has demonstrated remarkable prognostic value in predicting outcomes in independent pediatric AML datasets (Elsayed et al, 2020) (page 12, lines 39-41).

The TARGET AML-2 dataset contains 1,398 AML samples that were recently released on the Genomic Data Commons and not used in our initial analysis depicted in **Fig. 2**, i.e., TARGET AML-1. For the TARGET AML-2 dataset (n=1,398), we calculated the LSC6 risk score using the algorithm adopted by Fornerod *et al.* (explained in the methodology section; page 21, lines 33-39). We assessed the enrichment of M1 Macrophage (*S100A8*, *S100A9*, *S100A12*, *TYROBP*, *VCAN*, *CD68*, *MNDA*, *CYBB*, *STAT1*), naïve T cells (*CCR7*, *LEF1*, *TCF7*, *SELL*), effector T cells (*CCL5*, *NKG7*, *GNLY*, *GZMA*, *GZMK*, *NFACT1*), exhausted T cells (*HAVCR2*, *LAG3*, *PDCD1*, *NFATC1*, *TIGIT*, *TOX*), and regulatory T cells (*CCL5*, *KLRB1*, *KLRD1*, *GZMH*, *CD69*, *CD44*) in the bulk RNA-seq datasets by calculating single sample gene set enrichment analysis (ssGSEA) score for each immune cell type. In the Fornerod *et al.* AML dataset, we demonstrate a significantly ($P<.05$) higher exhausted T cell population within high-risk score samples and significant ($P<.0001$) enrichment of M1 Macrophages within low-risk score samples compared to medium-risk score patients (**Figure 27B**). We performed a similar analysis on the TARGET AML-2 dataset and observed significantly high naïve T cells ($P<.05$) and M1 macrophages ($P<.0001$) in the low-risk score samples compared to high-risk score samples. The high-risk compared to low-risk samples depicted significantly high enrichment of exhausted T cells ($P<.001$) (**Figure S28B**). Further details of the analysis are described in response 1.5 mentioned earlier.

- p. 10 lines 11-12: It should be demonstrated that these putative non-leukemic cells do not contain CNAs, rather than just relying on marker genes for the assertion that they are nonmalignant.

Response: To establish the non-leukemic nature of cluster 6, we conducted supplementary copy number alteration (CNA) analysis employing the inferCNV tool. The findings revealed no prominent amplifications or deletions in cluster 6 when compared to healthy BM monocytes (Figure S22) (page 11, lines 5-7).

• **p. 11 lines 13-15: Writing could be more clear.**

Response: We have edited the text to provide a more concise and precise analysis and result explanation in the revised manuscript (page 12, lines 2-5).

REVIEWER COMMENTS

Reviewer #1 (Remarks to the Author):

Authors have made new analysis based on the reviewer comments and added new validation sets which increase the robustness of the findings.

Reviewer #3 (Remarks to the Author):

I appreciate that the authors included several independent data sets, which they used to assess the performance of their 7-gene signature to distinguish blasts from non-blasts in pediatric AML. These assessments renewed some of my original concerns about the basic approach, which I feel could have been done more cleanly and less ambiguously if these additional data sets had been used to (re)derive the gene signature. Specific comments:

p. 4 lines 36-40: Thank you for clarifying the methodology. It appears the authors did not check to make sure that the DEGs are consistent within each patient.

p. 6 lines 7-18 and Fig. S7: I appreciate the inclusion of the healthy BM samples, but the results of this comparison (both the module score and the overlapping blast populations in AML and normal BM in the UMAPs) suggest that the 7-gene signature might just mark early myeloid blast-like cells – not necessarily malignant blasts. S7C is also concerning because it appears to be an apples-to-oranges comparison: why does it compare AML blasts to the entire HSC data set, which contains many mature cells, rather than to the early myeloid blast-like populations in the HSC data set?

p. 6 lines 31-33: Still labeling blasts based on gene expression. Are these blast clusters copy-number-altered? Are other cell types copy number-altered? (Not all AML cells look like blasts in scRNA-seq data.)

p. 6 lines 37-40: Some of these genes appear to go in the wrong direction. Are these 7 signature genes the most highly correlated with blast percentage in these data sets? A regression analysis of expression vs actual blast percentage would be a clean way to do this, without calculating all possible p-value pairs.

p. 7 lines 10-21: I think this is figure S16, not S12? S12 appears to be cut off. Regardless, in Fig. S16, it looks like the blasts in nine patients do not actually contain CNVs (other than Chromosome 6 CNVs, which should be excluded, as they often indicate differential HLA gene expression rather than true CNVs), which goes back to my original concern about identifying malignant cells. In Fig. S17 it again looks like several patients' blasts do not have CNVs. Since many AML tumors do not have CNVs, I originally suggested exome sequencing and single-cell variant calling for malignant cell identification (cb_sniffer and Vartrix were developed for this purpose -- in the context of malignant cell identification in AML), which I think might have given cleaner results. It would also be helpful to see where the CN-altered/mutation-containing cells lie on the UMAPs. As presented, it is hard to draw connections between the CNV analyses and the UMAPs.

Figure S27. This substantially strengthens the conclusions in this part of the manuscript. (NB: I don't see where deconvolution was used (e.g. CIBERSORTx or similar), as ssGSEA is not considered deconvolution.)

Reviewer #1 (Remarks to the Author):

Authors have made new analysis based on the reviewer comments and added new validation sets which increase the robustness of the findings.

We would like to thank the reviewer for valuable comments and suggestions during the previous revision of the manuscript. These inputs have significantly contributed to enhancing the robustness of our reported findings.

Reviewer #3 (Remarks to the Author):

I appreciate that the authors included several independent data sets, which they used to assess the performance of their 7-gene signature to distinguish blasts from non-blasts in pediatric AML. These assessments renewed some of my original concerns about the basic approach, which I feel could have been done more cleanly and less ambiguously if these additional data sets had been used to (re)derive the gene signature. Specific comments:

p. 4 lines 36-40: Thank you for clarifying the methodology. It appears the authors did not check to make sure that the DEGs are consistent within each patient.

Response: Thank you for your valuable feedback. This section of the manuscript (page 4, lines 36-41) pertains to identifying differentially expressed genes (DEGs) between diagnosis (Dx)-enriched AML-blast clusters and end of induction (EOI) non-blast clusters from four AML patients, for whom we had both Dx and EOI samples (patients 3, 5, 6, 14). The DEGs were selected based on higher average fold change and multiple test-corrected P-values derived using Wilcoxon Rank Test. The Wilcoxon Rank Test is a non-parametric statistical model for P-value calculation based on mean and the variance of genes between groups (i.e., Dx-enriched AML-blast cells, and EOI non-blast cells) to identify significantly altered genes. It is important to note that the expression of individual genes may vary among patients, likely due to the inherent heterogeneity of AML. However, the primary focus of our DEG analysis remained on creating a robust gene signature that could consistently discriminate AML blasts from non-blast cells across all four patients.

Addressing the point raised by the reviewer, we have examined the expression patterns of individual genes within the 7-gene signature in the four AML patients. This analysis indicates that there is variability in the expression of these genes (Figure S6A), which can be attributed to the heterogeneity of AML. The expression levels of most genes are significantly higher in Dx-AML blast cells across multiple patients. In contrast, these genes either exhibited no expression or only minimal expression in non-blast EOI-enriched cells from all patients. We have added a new supplementary figure showing the expression of the genes from the 7-gene signature separated based on individual patients (Figure S6A) and updated the text in the manuscript accordingly (page 5, lines 33-36).

p. 6 lines 7-18 and Fig. S7: I appreciate the inclusion of the healthy BM samples, but the results of this comparison (both the module score and the overlapping blast populations in AML and normal BM in the UMAPs) suggest that the 7-gene signature might just mark early myeloid blast-like cells – not necessarily malignant blasts. S7C is also concerning because it appears to be an apples-to-oranges comparison: why does it compare AML blasts to the entire HSC data set, which contains many mature cells, rather than to the early myeloid blast-like populations in the HSC data set?

Response: Thank you for this comment. We concur with the reviewer's perspective that comparing AML blasts with early myeloid blast-like populations would provide valuable insights and enhance the significance of our 7-gene signature. To evaluate the expression of 7-gene signature in early myeloid, mature myeloid, and AML blast populations, we subsetted the normal HSC and healthy BM scRNA-seq data into common myeloid progenitor (CMP) (cluster 3) and mature monocyte/macrophage (clusters 1, 2, 12) based on the expression of established markers (*CD14*, *CD68*) and SingleR labels. We revised Figure S7 to show the expression of the 7-gene signature as well as the individual genes in different clusters as a dot plot as well as the proportion of the three sample types (AML, healthy BM, normal HSC) in each cluster (Figure S7B). The module score of the 7-gene signature was significantly higher in AML-blasts compared to normal HSC and healthy BM CMP cells (revised Figure S7C). The evaluation of the expression of individual genes also revealed significantly higher expression of most of the

genes from the 7-gene signature in AML-blasts as compared to HSC CMP cells, except *NREP* and *C1QBP*. These results support our finding of higher expression of the 7-gene signature in pediatric AML-blast cells as compared to normal cells, including progenitor cells. We have updated the text (page 6, lines 15-27) in the revised manuscript and modified Figure S7 and its legend.

p. 6 lines 31-33: Still labeling blasts based on gene expression. Are these blast clusters copy-number-altered? Are other cell types copy number-altered? (Not all AML cells look like blasts in scRNA-seq data.)

Response: Thank you for this comment. We had used multiple factors when annotating clusters as “AML-blast cells”: (i) Substantial proportion of cells in the clusters should be from AML samples (Figure S9A, C), (ii) lack of expression for canonical normal cell marker (Figure S9B), (iii) expression of known AML-blasts markers, i.e., *CD38*, *MPO*, and *AZU1* (Figure S9B) and (iv) SingleR annotation of cell clusters as myeloid progenitors (GMP, CMP) (Figure S9C). To address the reviewer’s critique, we performed additional copy number alteration (CNA) analysis using the inferCNV tool. The AML clusters were set as “observations” and the normal HSC cells were set as “references”. The tool predicts CNAs (loss or duplication) per chromosome from each cell. We calculated the total CNAs containing chromosomes per cell and plotted them as a feature plot (Figure S9D) to show the distribution of CNAs per cell on a UMAP. While blast clusters 11, 13, and 14 had high CNA chromosome counts, some non-blast clusters such as 2 (Mono/Mac), 12 (GMP, Mono/Mac), and 9 (Mono/Mac) have CNA counts as well. These non-blast clusters with high numbers of chromosomes with CNAs also have larger proportions (>30%) of healthy BM and healthy HSC cells, expressed mature monocyte and macrophage markers (*CD14*, *CD68*), lack expression of AML blast markers (*MPO*, *CD38*, *AZU1*), and/or have high proportions of cells predicted as monocytes by SingleR. Therefore, taking all the factors into account, we believe these high CNA non-blast clusters contain non-malignant normal cells. We have revised Figure S9C to show the proportions of the sample types (AML, normal HSC, healthy BM) in each cluster. Overall, blast clusters have a higher distribution of CNA chromosome counts than non-blast clusters (Figure S9E). The additional CNA analysis is incorporated into the edited manuscript (page 6, lines 40-43; page 7, lines 1-7) and updated the legend of Figure S9.

p. 6 lines 37-40: Some of these genes appear to go in the wrong direction. Are these 7 signature genes the most highly correlated with blast percentage in these data sets? A regression analysis of expression vs actual blast percentage would be a clean way to do this, without calculating all possible p-value pairs.

Response: Thanks for the suggestion. This part pertains to the external validation of the 7-gene signature in the external bulk RNA-seq TARGET AML-2 dataset (page 7, lines 12-26). As suggested, we performed the regression analysis of the Z-score of 7-gene signature and expression of individual genes of the 7-gene signature vs. actual blast percentage in TARGET AML-2 dataset. To perform this analysis, we selected primary AML BM samples with greater than 5% diagnosis BM blast percentage (n=1,320). The Z-score of the 7-gene signature indicating combined expression depicted a significantly positive association ($R=0.16$, $P=4.7e-09$) with blast

percentages. Further, expression of four of the seven genes in the 7-gene signature were found to be significantly positively correlated ($R > 0$, $P < 0.01$) with BM blast percentages in TARGET AML-2 dataset (Figure S10). *PRAME* seems to have weak association with blast percentages as elevated expression is observed in select AML samples irrespective of blast percentages (Figure S10C). *AZU1* and *TRH* have significant negative correlation ($R < 0$, $P < 0.01$) with BM blast percentages. This is most likely attributable to the heterogeneity in AML samples, and overall expression profile being obtained from bulk RNA-seq data comprised of both non-malignant and malignant BM cells, rather than solely measuring the expression from blast cells as in the scRNA-seq data. As stated above, the 7-gene signature was significantly positively correlated with BM blast percentage in this validation TARGET AML-2 dataset. While developing the AML blast-specific signature, these genes (from the 7-gene signature) were among the 44 genes identified to be over-expressed in high blast % samples (>60%) versus low blast % samples (<30%) and EOI samples in the TARGET AML-1 dataset (Table S2). While these were not the seven top differentially expressed genes among the 44 genes (Table S2), they were found to be the most important features of the best performing SVM trained to classify AML cells as blast or non-blast (page 19, lines 21-40). We have updated Figure S10 and its legend and revised the manuscript text (page 7, lines 1523).

p. 7 lines 10-21: I think this is figure S16, not S12? S12 appears to be cut off. Regardless, in Fig. S16, it looks like the blasts in nine patients do not actually contain CNVs (other than Chromosome 6 CNVs, which should be excluded, as they often indicate differential HLA gene expression rather than true CNVs), which goes back to my original concern about identifying malignant cells. In Fig. S17 it again looks like several patients' blasts do not have CNVs. Since many AML tumors do not have CNVs, I originally suggested exome sequencing and single-cell variant calling for malignant cell identification (cb_sniffer and Vartrix were developed for this purpose -- in the context of malignant cell identification in AML), which I think might have given cleaner results. It would also be helpful to see where the CN-altered/mutation-containing cells lie on the UMAPs. As presented, it is hard to draw connections between the CNV analyses and the UMAPs.

Response: The text on page 7, lines 10-21 (page 7, lines 27-38 in the revised manuscript) are associated with Figure S12 from the four paired Dx, EOI patient samples that were used for the generation of the 7-gene signature. We have corrected the cut-off part of this figure. Figures S16 and S17, referred to on page 9, lines 41-43 in the revised manuscript, are for the Dx AML-blasts CNA analysis from patients 1-14. The 7-gene signature was used to select the blast clusters, as described on page 8, lines 5-7 (Figure S13). Additionally, we performed CNA analysis to support this blast cell annotation. The patient-wise CNA analysis shows that there was heterogeneity in predicted CNAs among patients, and we agree with the reviewer that not all AML blasts contain CNAs. The CNAs distribution in various UMAP clusters has been plotted by CNA chromosome count metric per cell. We have excluded chromosome 6 CNAs from analysis due to their false positive probability from HLA gene expression (Figures S16B, C) and updated that in the main text (page 10, lines 2-6), methods section (page 21, lines 29-33) and Figure S16 legend. We have updated the CNA analysis representation showing the

CNA feature map (Figure S16C) for blast cells from each patient (Figure S16B). In the feature map, we have added T-cells that have been used as a reference for CNA analysis and depicted no/minimal CNAs (Figure S16C). Most of the blast cell clusters from AML patients (Figure S16B) have higher CNA count (Figure S16C, colored in yellow and red) supporting malignant phenotype as compared to normal T cells.

Figure S27. This substantially strengthens the conclusions in this part of the manuscript. (NB: I don't see where deconvolution was used (e.g. CIBERSORTx or similar), as ssGSEA is not considered deconvolution.)

Response: Thanks for pointing out this oversight. We have corrected the figure titles for Figures S27 and S28 and edited the manuscript (page 13, line 10) to reflect the correct terminology for the enrichment analysis.

REVIEWERS' COMMENTS

Reviewer #3 (Remarks to the Author):

Most of my concerns have been addressed; thank you

Reviewer #3 (Remarks to the Author):

Most of my concerns have been addressed; thank you.

We are grateful to reviewer 3 for the critiques, and suggestions to improve the manuscript.